# Spurious Correlations in High Dimensional Regression: The Roles of Regularization, Simplicity Bias and Over-Parameterization

**Simone Bombari** [1]   **Marco Mondelli** [1]

## Abstract

Learning models have been shown to rely on spurious correlations between non-predictive features and the associated labels in the training data, with negative implications on robustness, bias and fairness. In this work, we provide a statistical characterization of this phenomenon for high-dimensional regression, when the data contains a predictive *core* feature $x$ and a *spurious* feature $y$. Specifically, we quantify the amount of spurious correlations $\mathcal{C}$ learned via linear regression, in terms of the data covariance and the strength $\lambda$ of the ridge regularization. As a consequence, we first capture the simplicity of $y$ through the spectrum of its covariance, and its correlation with $x$ through the Schur complement of the full data covariance. Next, we prove a trade-off between $\mathcal{C}$ and the in-distribution test loss $\mathcal{L}$, by showing that the value of $\lambda$ that minimizes $\mathcal{L}$ lies in an interval where $\mathcal{C}$ is increasing. Finally, we investigate the effects of over-parameterization via the random features model, by showing its equivalence to regularized linear regression. Our theoretical results are supported by numerical experiments on Gaussian, Color-MNIST, and CIFAR-10 datasets.

## 1. Introduction

Machine learning systems have been shown to learn from patterns that are statistically correlated with the intended task, despite not being causally predictive (Geirhos et al., 2020; Xiao et al., 2021). As a concrete example, a blue background in a picture might be positively correlated with the presence of a boat in the foreground, and while not being a predictive feature per se, a trained deep learning model could use this information to bias its prediction. In the literature, this statistical (but non causal) connection is referred to as a *spurious correlation* between a feature and the learning task. A recent and extensive line of research

has investigated the extent to which deep learning models manifest this behavior (Geirhos et al., 2019; Xiao et al., 2021) and has proposed different mitigation approaches (Sagawa et al., 2020a; Liu et al., 2021), given its implications to robustness, bias, and fairness (Zliobaite, 2015; Zhou et al., 2021). The phenomenon, also referred to as shortcut learning, is often attributed to the relative "simplicity" of spurious features (Geirhos et al., 2020; Shah et al., 2020; Hermann & Lampinen, 2020) and to the implicit bias of over-parameterized models toward learning simpler patterns (Belkin et al., 2019; Rahaman et al., 2019; Kalimeris et al., 2019). Consequently, the *core features* that are informative about the task (e.g., the boat in the foreground) may be neglected, as *spurious features* (e.g., the blue background) provide an easier shortcut to minimize the loss function.

Prior work has attempted to formalize the *simplicity bias* relying on boolean functions (Qiu et al., 2024), model-specific biases (Morwani et al., 2023), one-dimensional features (Shah et al., 2020) and their pairwise interactions (Pezeshki et al., 2021). However, when considering high-dimensional natural data (e.g., the boat and its background in Figure 1), it remains unclear, based on these notions, what exactly makes the features easy or difficult to learn, and to what extent a trained model relies on spurious correlations. Furthermore, while Sagawa et al. (2020b) show that over-parameterization can exacerbate spurious correlations when re-weighting the objective on minority groups (e.g., boats with a green background), its effect on models trained via empirical risk minimization (ERM) is less understood. This is a critical point when additional group membership annotations are too expensive to obtain, and ERM is a key part of training (Liu et al., 2021; Ahmed et al., 2021).

Our work tackles these issues: we provide a rigorous characterization of the statistical mechanisms behind learning spurious correlations in high-dimensional data, focusing on the solution obtained via ERM. Formally, we model the input sample $z$ as composed by two distinct features, *i.e.*, $z = [x, y]$, where $x \in \mathbb{R}^d$ is the core feature and $y \in \mathbb{R}^d$ the spurious one. The first panel of Figure 1 provides an illustration with a boat in the foreground ($x$) and its blue background ($y$). Then, we quantify spurious correlations via the covariance $\mathcal{C}$ (see (3)) between the label $g$ ("boat") and the model output given $\tilde{z} = [\tilde{x}, y]$ as input. Here, $\tilde{x}$ (a

[1]Institute of Science and Technology, Austria. Correspondence to: Simone Bombari <simone.bombari@ista.ac.at>.

*Proceedings of the 42$^{nd}$ International Conference on Machine Learning*, Vancouver, Canada. PMLR 267, 2025. Copyright 2025 by the author(s).

truck in the foreground) is a new core feature independent of everything else, see the second panel of Figure 1. Now, if $\mathcal{C}$ is positive, it means that the model is biased towards $g$ only because of $y$, since $\tilde{x}$ is independent from $x$ and $g$. More precisely, we provide a sharp, non-asymptotic characterization of $\mathcal{C}$ for linear regression (Theorem 4.3). Armed with such a characterization, we then:

• Interpret $\mathcal{C}$ via upper bounds on its magnitude (Proposition 5.1). This highlights the role of the regularization strength and of the data covariance via *(i)* its Schur complement with respect to the covariance of the core feature $x$, and *(ii)* the covariance of the spurious feature $y$. Specifically, we link the smallest eigenvalue of the Schur complement to the strength of the correlation between $y$ and $x$, and the largest eigenvalue of the spurious covariance to the simplicity of $y$.

• Prove a trade-off between $\mathcal{C}$ and the test loss (Proposition 5.3), which implies that spurious correlations can be beneficial to performance when learning in-distribution. Specifically, we show that the optimal regularization minimizing the test loss lies in an interval where $\mathcal{C}$ is positive and monotonically increasing.

• Investigate the role of over-parameterization via a *random features* (RF) model. Specifically, we show that the RF model is equivalent to linear regression with an effective regularization that depends on the over-parameterization (Theorem 6.4). This allows to leverage the earlier analysis on regularized linear regression to quantify spurious correlations in over-parameterized, non-linear models.

Throughout the paper, the theoretical results are supported by numerical experiments on synthetic Gaussian data, Color-MNIST, and CIFAR-10, which validates our analysis even in settings not strictly following the modeling choices. Our code is publicly available at the GitHub repository `simone-bombari/spurious-correlations`.

## 2. Related Work

**Spurious correlations.** Learning from spurious correlations in a training dataset is rather common (Geirhos et al., 2019; Arjovsky et al., 2020; Geirhos et al., 2020; Sagawa et al., 2020a; Xiao et al., 2021; Singla & Feizi, 2022) and it has unwanted consequences, e.g., lack of robustness towards domain shift, prediction bias and compromised algorithmic fairness (Zliobaite, 2015; Geirhos et al., 2019; Zhou et al., 2021; Veitch et al., 2021; Seo et al., 2022). Thus, multiple mitigation approaches have been proposed, with (Sagawa et al., 2020a; Zhang et al., 2021) or without (Liu et al., 2021; Ahmed et al., 2021) available annotations. Specifically, Tiwari & Shenoy (2023) exploit the difference in the features learned at different layers of a deep neural network; Izmailov et al. (2022); Kirichenko et al. (2023) re-train the last layer of the ERM solution to adapt the features to the

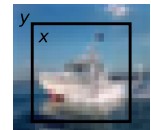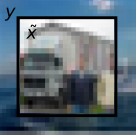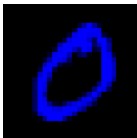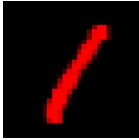

Figure 1. *Left two panels:* pictorial representation of the core (spurious) feature $x$ ($y$) and an independent core feature $\tilde{x}$, taken from an image of a boat and a truck in the CIFAR-10 dataset. *Right two panels:* examples from a binary Color-MNIST dataset, where the labels correspond to the number shapes, and the zeros (ones) are colored in blue (red) with probability $(1 + \alpha)/2$.

distribution shift; and Chang et al. (2021a); Plumb et al. (2022) mitigate the problem via data augmentation.

**Simplicity bias.** Recent work has shown that deep learning models have a bias towards learning from "easier" patterns (Belkin et al., 2019; Rahaman et al., 2019; Kalimeris et al., 2019). In shortcut learning, this property is formalized in different ways across the literature. The difficulty of a feature is defined in terms of the minimum complexity of a network that learns it by Hermann & Lampinen (2020) and in terms of the smallest amount of linear segments that separate different classes by Shah et al. (2020). Moayeri et al. (2022) connect the simplicity to the position and size of the features in an image. Morwani et al. (2023) define the simplicity bias in 1-hidden layer neural networks via the rank of a projection operator that does not alter them substantially, and they focus on a dataset generated via an independent features model learned via the NTK. The NTK is also used to analyze gradient starvation (Pezeshki et al., 2021) and feature availability (Hermann et al., 2024), regarded as explanations of the simplicity bias. Qiu et al. (2024) focus on parity functions and staircases, analyzing the learning dynamics of features having different complexity.

**High-dimensional regression.** The test loss of linear regression when the input dimension $d$ scales proportionally with the sample size $n$ has been characterized precisely both in-distribution (Hastie et al., 2022; Cheng & Montanari, 2024) and under covariate shift (Yang et al., 2023; Mallinar et al., 2024; Song et al., 2024). Furthermore, Montanari et al. (2019); Chang et al. (2021b); Han & Xu (2023) have studied the distribution of the ERM solution via the convex Gaussian min-max Theorem (Thrampoulidis et al., 2015). Specifically, our work builds on the non-asymptotic characterization provided by Han & Xu (2023).

In contrast with linear regression where the number of parameters equals the input dimension, random features models (Rahimi & Recht, 2007) capture the effects of over-parameterization, as the number of parameters is independent of $d$ and $n$. Mei & Montanari (2022) have characterized the test loss of random features, showing that it displays a double descent (Belkin et al., 2019). Furthermore, the RF model has been used to understand a wide family of phe-

nomena such as feature learning (Ba et al., 2022; Damian et al., 2022; Moniri et al., 2024), robustness under adversarial attacks (Dohmatob & Bietti, 2022; Bombari et al., 2023; Hassani & Javanmard, 2024), and distribution shift (Tripuraneni et al., 2021; Lee et al., 2023). This model has been also considered in the setting where the data has two dependent components (Loureiro et al., 2021), although it was used to study the training and generalization error when only partial information is available. The equivalence between an over-parameterized RF model and a regularized linear one has also been studied in detail (Goldt et al., 2022; 2020; Hu & Lu, 2023; Montanari & Saeed, 2022). However, existing rigorous results show the equivalence at the level of training and test error. In contrast, we are interested in the covariance defined in (3) and, for this reason, we prove an equivalence at the level of the predictor (Theorem 6.4).

## 3. Preliminaries

**Notation.** Given a vector $v$, we denote by $\|v\|_2$ its Euclidean norm. Given a matrix $A$, we denote by $\mathrm{tr}(A)$ and $\|A\|_{\mathrm{op}}$ its trace and operator (spectral) norm. Given a symmetric matrix $A$, we denote by $\lambda_{\min}(A)$ ($\lambda_{\max}(A)$) its smallest (largest) eigenvalue. We denote by $\lambda$ (without a sub-script) the $\ell_2$ regularization term in ridge regression. All complexity notations $\Omega(\cdot)$, $\mathcal{O}(\cdot)$, $\omega(\cdot)$, $o(\cdot)$ and $\Theta(\cdot)$ are understood for large data size $n$, input dimension $d$, and number of parameters $p$. We indicate with $C, c > 0$ numerical constants, independent of $n$, $d$, $p$, whose value may change from line to line.

**Setting.** We consider supervised learning with $n$ training samples $\{(z_1, g_1), \ldots, (z_n, g_n)\}$ and labels defined by a (not necessarily deterministic) function of the inputs $g_i = f^*(z_i)$, where $z_i \in \mathbb{R}^{2d}$ denotes the $i$-th training input and $g_i \in \mathbb{R}$ the corresponding label. Input samples are composed by two distinct parts (or *features*), i.e., $z_i^\top = [x_i^\top, y_i^\top]$, with $x_i, y_i \in \mathbb{R}^d$, and they are sampled i.i.d. from the distribution $\mathcal{P}_{XY}$. We further denote with $\mathcal{P}_X$ ($\mathcal{P}_Y$) the marginal distribution of the $x_i$-s ($y_i$-s). The features $x$ and $y$ have the same dimension $d$ to ease the presentation. We opted to focus on the setting $z = [x, y]$ due to its connection with the motivating example of images with background, but the analysis could be extended to other cases, such as $z = x + y$, briefly discussed in Appendix D.

We focus on the setting where the labels $g_i$ depend only on $x_i$, i.e., $g_i = f^*(z_i) = f_x^*(x_i)$ for some (not necessarily deterministic) function $f_x^*$. Hence, $y_i$ is independent from $g_i$, after conditioning on $x_i$. We highlight that the independence between $y_i$ and $g_i$ is conditional on $x_i$, and that the covariance between $y_i$ and $x_i$ is in general non-zero. We refer to $y_i$ as the *spurious feature* of the $i$-th sample, and to $x_i$ as its *core feature*. As an example, $x_i$ may represent the main object in an image and $y_i$ the (not necessarily independent)

background, see Figure 1.

In this setup, the training data is used to learn $f^*(z)$ through a parametric model $f(\theta, z)$ via regularized empirical risk minimization (ERM). Specifically, we perform the following optimization in parameter space:

$$\hat{\theta} = \arg\min_\theta \left( \frac{1}{n} \sum_{i=1}^n \ell\left(f(\theta, z_i), g_i\right) + \lambda \|\theta\|_2^2 \right), \quad (1)$$

for some regularization term $\lambda \geq 0$, where $\ell$ is a loss function[1]. We define the test loss associated to the model $f(\hat{\theta}, \cdot)$ as

$$\mathcal{L}(\hat{\theta}) = \mathbb{E}_{z \sim \mathcal{P}_{XY}, \, g = f^*(z)} \left[ \ell\left(f(\hat{\theta}, z), g\right) \right]. \quad (2)$$

We note that this test loss is in distribution.

**Spurious correlations.** We express the extent to which a model $f(\hat{\theta}, \cdot)$ learns spurious correlations between the spurious feature $y$ and the label $g$ as

$$\mathcal{C}(\hat{\theta}) = \mathrm{Cov}\left( f\left(\hat{\theta}, [\tilde{x}^\top, y^\top]^\top\right), g \right), \quad (3)$$

where the covariance is computed on the probability space of $[x^\top, y^\top]^\top \sim \mathcal{P}_{XY}$, $g = f_x^*(x)$ and of the independent core feature $\tilde{x} \sim \mathcal{P}_X$. In words, $\mathcal{C}(\hat{\theta})$ expresses how the output of the model $f(\hat{\theta}, \cdot)$ evaluated on an out-of-distribution sample $[\tilde{x}^\top, y^\top]^\top$ (where the two features are sampled independently from the marginal distributions $\mathcal{P}_X$ and $\mathcal{P}_Y$) correlates to the label associated to the in-distribution sample $g = f^*(z) = f_x^*(x)$. We highlight that, if the model $f(\hat{\theta}, \cdot)$ *does not rely* on the spurious feature $y$, then $\mathcal{C}(\hat{\theta}) = 0$ as $x$ and $\tilde{x}$ are independent.

We note that (3) formalizes to the regression setting the definition in the survey (Ye et al., 2024) and the fairness metric in (Zliobaite, 2015) (when interpreting $y$ as the protected variable). Furthermore, in the context of classification, it can be connected to the worst group accuracy (Sagawa et al., 2020a;b). In fact, our definition (3) is also related to the *out-of-distribution* test loss: assuming $\mathbb{E}[f(\hat{\theta}, [\tilde{x}^\top, y])^2] = \mathbb{E}[f_x^*(\tilde{x})^2] = 1$ and $\mathbb{E}[f(\hat{\theta}, [\tilde{x}^\top, y])] = \mathbb{E}[f_x^*(\tilde{x})] = 0$ for simplicity and considering a quadratic loss, we have

$$\mathbb{E}_{\tilde{x}, y}\left[ \left( f(\hat{\theta}, [\tilde{x}^\top, y]) - f_x^*(\tilde{x}) \right)^2 \right] \geq 2 - 2\sqrt{1 - \mathcal{C}(\hat{\theta})^2}. \quad (4)$$

This implies that an increase in $\mathcal{C}(\hat{\theta})$ hurts the performance of the model when core and spurious features are sampled independently (and, thus, the model is tested out-of-distribution). The proof of (4) is in Appendix E.

---

[1]In general, existence and uniqueness of $\hat{\theta}$ depend on the choice of the model $f(\theta, z)$, the loss function $\ell$ and the regularization term $\lambda$. For the purposes of our work, we will precisely define $\hat{\theta}$ for linear regression (Section 4) and for random features (Section 6).

# 4. Precise Analysis for Linear Regression

To study $\mathcal{C}(\cdot)$ as defined in (3), we focus on a high-dimensional *linear regression* model, *i.e.*,

$$f_{\text{LR}}(\theta, z) = z^\top \theta, \qquad (5)$$

where $\theta \in \mathbb{R}^{2d}$. The data also follows a linear model, *i.e.*,

$$g_i = z_i^\top \theta^* + \epsilon_i = x_i^\top \theta_x^* + \epsilon_i, \qquad (6)$$

where $\theta^* \in \mathbb{R}^{2d}$, $\theta_x^* \in \mathbb{R}^d$, and $\epsilon_i$ is label noise. The second equality in (6) implies that $\theta^* = [\theta_x^{*\top}, \mathbf{0}_d^\top]^\top$, where $\theta_x^*, \mathbf{0}_d \in \mathbb{R}^d$ and each entry of $\mathbf{0}_d$ is 0. We set $\|\theta^*\|_2 = \|\theta_x^*\|_2 = 1$ and let the $\epsilon_i$-s be i.i.d. (and independent from the $z_i$-s), mean-0, sub-Gaussian, with variance $\sigma^2 > 0$.

We introduce the shorthands $Z = [z_1^\top, \ldots, z_n^\top]^\top \in \mathbb{R}^{n \times 2d}$, $G = [g_1, \ldots, g_n]^\top \in \mathbb{R}^n$, and $\mathcal{E} = [\epsilon_1, \ldots, \epsilon_n]^\top \in \mathbb{R}^n$ to indicate the data matrix, the labels, and the noise vector respectively. Then, using a quadratic loss, (1) reads

$$\hat{\theta}_{\text{LR}}(\lambda) = \arg\min_\theta \left( \frac{1}{n} \|Z\theta - G\|_2^2 + \lambda \|\theta\|_2^2 \right), \qquad (7)$$

which admits the unique solution

$$\hat{\theta}_{\text{LR}}(\lambda) = \left( Z^\top Z + n\lambda I \right)^{-1} Z^\top G, \qquad (8)$$

for $\lambda > 0$ and, if $Z^\top Z$ is invertible, also for $\lambda = 0$.

**Assumption 4.1** (Data distribution). $\{z_i\}_{i=1}^n$ are $n$ i.i.d. samples from the multivariate, mean-0, Gaussian distribution $\mathcal{P}_{XY}$, such that its covariance $\Sigma := \mathbb{E}\left[zz^\top\right] \in \mathbb{R}^{2d \times 2d}$ is invertible, with $\lambda_{\max}(\Sigma) = \mathcal{O}(1)$, $\lambda_{\min}(\Sigma) = \Omega(1)$, and $\text{tr}(\Sigma) = 2d$.

This requirement could be relaxed to having sub-Gaussian data. We focus on the Gaussian case for simplicity, deferring the discussion on the generalization to Appendix B.2.

**Warm-up: no regularization ($\lambda = 0$).** Our first result concerns the amount of spurious correlations learned in the un-regularized setting.

**Proposition 4.2.** *Let $\lambda = 0$ and $Z^\top Z \in \mathbb{R}^{2d \times 2d}$ be invertible[2]. Let $\mathcal{C}(\hat{\theta}_{\text{LR}}(0))$ be the amount of spurious correlations learned by the model $f_{\text{LR}}(\hat{\theta}_{\text{LR}}(0), \cdot)$. Then, we have that*

$$\mathbb{E}_{\mathcal{E}}[\mathcal{C}(\hat{\theta}_{\text{LR}}(0))] = 0. \qquad (9)$$

*Furthermore, if Assumption 4.1 holds and $n = \omega(d)$,*

$$|\mathcal{C}(\hat{\theta}_{\text{LR}}(0))| = \mathcal{O}(\log d/\sqrt{d}), \qquad (10)$$

*with probability at least $1 - 2\exp(-c\log^2 d)$ over $Z$ and $\mathcal{E}$, where $c$ is an absolute constant.*

---

[2]Under Assumption 4.1, this holds with probability 1 for $n \geq 2d$.

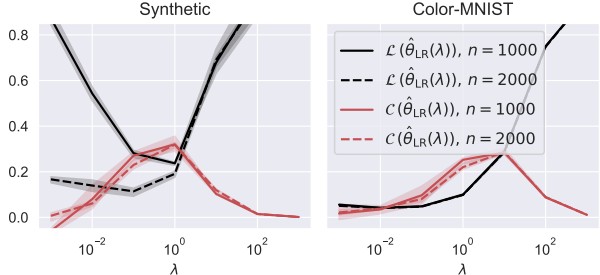

*Figure 2.* Test loss $\mathcal{L}(\hat{\theta}_{\text{LR}}(\lambda))$ (black) and spurious correlations $\mathcal{C}(\hat{\theta}_{\text{LR}}(\lambda))$ (red) as a function of the regularization term $\lambda$ for two values of the number of samples $n$. *Left:* synthetic Gaussian dataset, with $d = 400$ (additional details in Appendix F); *right:* binary Color-MNIST dataset with correlation $\sqrt{1 - \alpha^2} = 0.25$ between color and digit (see Figure 1).

In words, $f_{\text{LR}}(\hat{\theta}_{\text{LR}}(0), \cdot)$ *does not learn* any spurious correlation between the spurious feature $y$ and the label $g$. This is also clear from Figure 2, where we report in red the value of $\mathcal{C}(\hat{\theta}_{\text{LR}}(\lambda))$, which approaches 0 as $\lambda$ becomes small.

The idea of the argument is to write explicitly the solution

$$\hat{\theta}_{\text{LR}}(0) = \left( Z^\top Z \right)^{-1} Z^\top G = \theta^* + \left( Z^\top Z \right)^{-1} Z^\top \mathcal{E}, \quad (11)$$

where in the second step we separate the ground truth $\theta^*$ (which does not capture any dependence on $y$) from a term only depending on the label noise, which is mean-0 and independent from $y$. This directly gives (9). Then, the bound in (10) is obtained via standard concentration results on $\lambda_{\min}(Z^\top Z)$. The details are in Appendix B.

**General case with regularization ($\lambda > 0$).** Setting a regularizer $\lambda > 0$ often reduces the test loss, see the black curve in Figure 2. However, it also leads to non-trivial spurious correlations, and our main result provides a non-asymptotic characterization of this phenomenon.

**Theorem 4.3.** *Let Assumption 4.1 hold, $n = \Theta(d)$ and $\mathcal{C}(\hat{\theta}_{\text{LR}}(\lambda))$ be the amount of spurious correlations learned by the model $f_{\text{LR}}(\hat{\theta}_{\text{LR}}(\lambda), \cdot)$ for $\lambda > 0$. Denote by $P_y \in \mathbb{R}^{2d \times 2d}$ the projector on the last $d$ elements of the canonical basis in $\mathbb{R}^{2d}$, and set*

$$\mathcal{C}^\Sigma(\lambda) := \theta^{*\top} \Sigma \left( \Sigma + \tau(\lambda)I \right)^{-1} P_y \Sigma \theta^*, \qquad (12)$$

*where $\tau := \tau(\lambda)$ is implicitly defined as the unique positive solution of*

$$1 - \frac{\lambda}{\tau} = \frac{1}{n} \text{tr}\left( (\Sigma + \tau I)^{-1} \Sigma \right). \qquad (13)$$

*Then, for every $t \in (0, 1/2)$,*

$$\mathbb{P}_{Z,\mathcal{E}}\left( \left| \mathcal{C}(\hat{\theta}_{\text{LR}}(\lambda)) - \mathcal{C}^\Sigma(\lambda) \right| \geq t \right) \leq Cd\exp\left(-dt^4/C\right),$$

*where $C$ is an absolute constant.*

In words, Theorem 4.3 guarantees that $|\mathcal{C}(\hat{\theta}_{\text{LR}}(\lambda)) - \mathcal{C}^\Sigma(\lambda)| = o(1)$ with high probability (e.g., setting $t = d^{-1/5}$). Thus, for large $d, n$, we can theoretically analyze $\mathcal{C}(\hat{\theta}_{\text{LR}}(\lambda))$ via the deterministic quantity $\mathcal{C}^\Sigma(\lambda)$, which, as highlighted by (12), depends on $\theta^*$, the covariance of the data $\Sigma$, and the regularization $\lambda$ via the parameter $\tau(\lambda)$ introduced in (13). We further analyze the object $\mathcal{C}^\Sigma(\lambda)$ in Section 5 that immediately follows.

Note that, since $\hat{\theta}_{\text{LR}}(\lambda)$ is given by (8), when $\lambda > 0$ it cannot be decomposed as $\theta^* + \left(Z^\top Z\right)^{-1} Z^\top \mathcal{E}$ (as in the proof of Proposition 4.2 for $\lambda = 0$). Thus, we rely on the non-asymptotic characterization of $\hat{\theta}_{\text{LR}}(\lambda)$ recently provided by Han & Xu (2023). In particular, their analysis focuses on the proportional regime $n = \Theta(d)$ (as opposed to the regime $n = \omega(d)$, considered in (10)), and it allows to provide concentration bounds on a certain family of low-dimensional functions of $\hat{\theta}_{\text{LR}}(\lambda)$, which includes $\mathcal{C}$ as defined in (3). The details are in Appendix B.

## 5. Roles of Regularization and Simplicity Bias

We now interpret $\mathcal{C}^\Sigma(\lambda)$, which characterizes the spurious correlations via Theorem 4.3, in terms of the data covariance $\Sigma$ and the regularization $\lambda$. To do so, we express $\Sigma$ as

$$\Sigma = \left( \begin{array}{c|c} \Sigma_{xx} & \Sigma_{xy} \\ \hline \Sigma_{yx} & \Sigma_{yy} \end{array} \right), \tag{14}$$

where the block $\Sigma_{xx} = \mathbb{E}_{x \sim \mathcal{P}_X}\left[xx^\top\right] \in \mathbb{R}^{d \times d}$ ($\Sigma_{yy} = \mathbb{E}_{y \sim \mathcal{P}_Y}\left[yy^\top\right] \in \mathbb{R}^{d \times d}$) denotes the covariance of the core (spurious) feature sampled from its marginal distribution. The off-diagonal blocks are $\Sigma_{xy} = \Sigma_{yx}^\top = \mathbb{E}_{[x^\top, y^\top]^\top \sim \mathcal{P}_{XY}}\left[xy^\top\right] \in \mathbb{R}^{d \times d}$. Let us denote by

$$S_x^\Sigma := \Sigma_{yy} - \Sigma_{yx}\Sigma_{xx}^{-1}\Sigma_{xy} \tag{15}$$

the Schur complement of $\Sigma$ with respect to the top-left $d \times d$ block $\Sigma_{xx}$. In our setting, $S_x^\Sigma$ offers a helpful statistical interpretation. In fact, for multivariate Gaussian data, it corresponds to the conditional covariance of $y$ given $x$, *i.e.*,

$$\begin{aligned} S_x^\Sigma &= \text{Cov}\left(y|x = \bar{x}\right) \\ &= \mathbb{E}_{y|x=\bar{x}}\left[\left(y - \mathbb{E}_{y|x=\bar{x}}[y]\right)\left(y - \mathbb{E}_{y|x=\bar{x}}[y]\right)^\top\right]. \end{aligned} \tag{16}$$

Therefore, the spectrum of $S_x^\Sigma$ describes the degree of dependence between $y$ and $x$: on the one hand, if its eigenvalues are small, the feature $y$ is close to be determined by the knowledge of the feature $x$ (*i.e.*, $y$ is highly correlated with $x$); on the other hand, if its eigenvalues are large, the two features tend to be independent.

As an intuitive example, let us interpret classification for the binary Color-MNIST dataset (see Figure 1) as a 2-dimensional problem: the core feature is $x = +1 \, (-1)$

if the digit on the image is 1 (0); and the spurious feature is $y = +1 \, (-1)$ if the color is red (blue). If the digits and the colors have a correlation of $\alpha$, then $\Sigma_{xx} = \Sigma_{yy} = 1$ and $\Sigma_{xy} = \Sigma_{yx} = \alpha$. Thus, $S_x^\Sigma = 1 - \alpha^2$, which becomes 0 if $|\alpha| = 1$ (full correlation), and it is maximized in the setting of independence between colors and parity ($\alpha = 0$).

At this point, leveraging the decomposition of $\Sigma$ in (14) and the Schur complement in (15), we provide the following bounds on $\mathcal{C}^\Sigma(\lambda)$, which are proved in Appendix B.

**Proposition 5.1.** *Let $\mathcal{C}^\Sigma(\lambda)$ and $S_x^\Sigma$ be defined in (12) and (15), respectively. Then,*

$$\begin{aligned} |\mathcal{C}^\Sigma(\lambda)| \leq \min \Bigg( \|\Sigma_{yx}\|_{\text{op}}, \frac{\lambda_{\max}(\Sigma)^2}{\tau(\lambda)}, \\ \tau(\lambda)\sqrt{\text{Var}(g) - \sigma^2} \frac{\lambda_{\max}(\Sigma_{yy}) - \lambda_{\min}(S_x^\Sigma)}{\lambda_{\min}(S_x^\Sigma)\sqrt{\lambda_{\min}(\Sigma_{xx})}} \Bigg). \end{aligned} \tag{17}$$

We discuss the three upper bounds in (17) below.

*(i)*: $|\mathcal{C}^\Sigma(\lambda)| \leq \|\Sigma_{yx}\|_{\text{op}}$. The off-diagonal blocks $\Sigma_{yx} = \mathbb{E}[yx^\top]$ and $\Sigma_{xy} = \Sigma_{yx}^\top$ describe the correlation between $y$ and $x$. In the limit case $\|\Sigma_{yx}\|_{\text{op}} = 0$, we have that $x$ and $y$ are uncorrelated and, therefore, $\mathcal{C}^\Sigma(\lambda) = 0$, as there is no spurious correlation that the model can learn.

*(ii)*: $|\mathcal{C}^\Sigma(\lambda)| \leq \lambda_{\max}(\Sigma)^2/\tau(\lambda)$. From (13), one obtains that $\tau(\lambda) \to \infty$ as $\lambda \to \infty$. Thus, the bound implies that $\mathcal{C}^\Sigma(\lambda)$ approaches 0 as $\lambda$ grows large. This captures the intuition that, when the regularization $\lambda$ is large, the minimization in (7) is biased towards solutions with small norm and, therefore, the output of the model is small, which drives to 0 the spurious correlations as defined in (3). The behavior is confirmed by Figure 2: $|\mathcal{C}(\hat{\theta}_{\text{LR}}(\lambda))|$ is decreasing for large values of $\lambda$ and it eventually vanishes; at the same time, large values of $\lambda$ make the output of the model small, which in turn increases the test loss $\mathcal{L}(\hat{\theta}_{\text{LR}}(\lambda))$.

*(iii)*: The third bound in (17) can be rewritten as

$$\frac{|\mathcal{C}^\Sigma(\lambda)|\sqrt{\lambda_{\min}(\Sigma_{xx})}}{\sqrt{\text{Var}(g) - \sigma^2}} \leq \tau(\lambda)\frac{\lambda_{\max}(\Sigma_{yy}) - \lambda_{\min}(S_x^\Sigma)}{\lambda_{\min}(S_x^\Sigma)}, \tag{18}$$

where we have isolated on the LHS the terms depending on the covariance of the core feature $x$ ($\sqrt{\lambda_{\min}(\Sigma_{xx})}$) and on the scaling of the labels ($\sqrt{\text{Var}(g) - \sigma^2}$). We now discuss the dependence of the RHS of (18) w.r.t. *(a)* $\tau(\lambda)$, *(b)* $\lambda_{\min}(S_x^\Sigma)$, and *(c)* $\lambda_{\max}(\Sigma_{yy})$. As for *(a)*, we note that $\mathcal{C}^\Sigma(\lambda)$ approaches 0 for small values of $\lambda$. In fact, the RHS of (13) is smaller or equal to $2d/n$; thus, if we consider $2d < n$, we also get $\tau \leq \lambda\left(1 - 2d/n\right)^{-1}$, which implies $\tau(\lambda) \to 0$ as $\lambda \to 0$. This is in agreement with Proposition 4.2, which handles the case without regularization, and also with the numerical experiments of Figure 2. As for

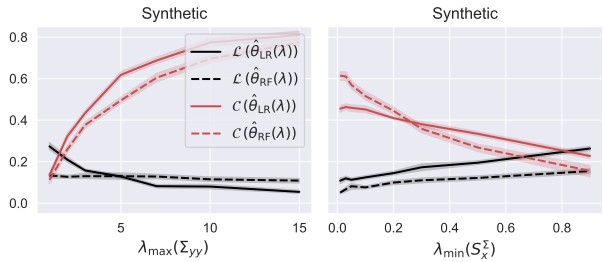

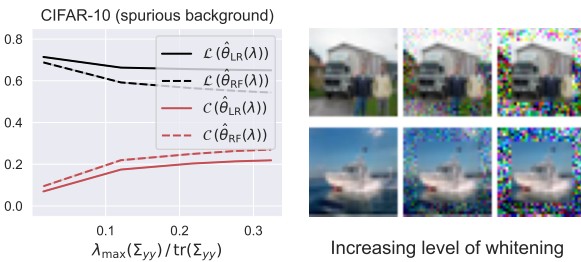

*Figure 3.* Test loss $\mathcal{L}(\hat{\theta}_{\text{LR/RF}}(\lambda))$ (black) and spurious correlations $\mathcal{C}(\hat{\theta}_{\text{LR/RF}}(\lambda))$ (red) as a function of $\lambda_{\max}(\Sigma_{yy})$ (left) and $\lambda_{\min}(S_x^\Sigma)$ (right) on a synthetic Gaussian dataset, for both linear regression and random features, with $\lambda = 1$ (additional details in Appendix F).

*Figure 4.* Test loss $\mathcal{L}(\hat{\theta}_{\text{LR/RF}}(\lambda))$ (black) and spurious correlations $\mathcal{C}(\hat{\theta}_{\text{LR/RF}}(\lambda))$ (red) as a function of $\lambda_{\max}(\Sigma_{yy}) / \text{tr}(\Sigma_{yy})$ on a CIFAR-10 dataset for different levels of whitening (details on the whitening process in Appendix F). We restrict to the classes "boat" and "truck" ($n = 10000$) and consider both linear regression and random features, with $\lambda = 1$.

*(b)*, we note that the bound is decreasing with $\lambda_{\min}(S_x^\Sigma)$. This is in agreement with the earlier discussion on how the spectrum of the Schur complement $S_x^\Sigma$ measures the degree of independence between the spurious feature $y$ and the core feature $x$. Finally, as for *(c)*, we note that the bound is increasing with $\lambda_{\max}(\Sigma_{yy})$, which is connected below to the *simplicity* of the spurious feature $y$. The increasing (decreasing) trend of $\mathcal{C}^\Sigma(\lambda)$ w.r.t. $\lambda_{\max}(\Sigma_{yy})$ $(\lambda_{\min}(S_x^\Sigma))$ is clearly displayed in Figure 3 for Gaussian data.

The connection between $\lambda_{\max}(\Sigma_{yy})$ and the *simplicity bias* of ERM can be illustrated via our initial image recognition example. The (spurious) background feature is intuitively an easy pattern to learn from the model: the pixels corresponding to the spurious feature behave consistently across the training data. This in turn skews the spectrum of $\Sigma_{yy}$, which has few dominant directions with eigenvalues much larger than the others. Note that this interpretation is similar to the model-dependent definition of simplicity in (Morwani et al., 2023). An empirical verification is provided in Figure 4, where we consider the CIFAR-10 dataset, restricted to the "boat" and "truck" classes. Before training a regression model, we whiten up to some level the background feature (as defined in Figure 1) to make it harder to learn, see the right side of Figure 4. Then, for different levels of whitening, we report $\mathcal{C}(\hat{\theta}_{\text{LR}}(\lambda))$ as a function of $\lambda_{\max}(\Sigma_{yy})$. We normalize $\lambda_{\max}(\Sigma_{yy})$ by the trace $\text{tr}(\Sigma_{yy})$ to exclude the size of the pattern from our experiment[3]. The red curve shows an increasing trend analogous to that displayed in Figure 3 for Gaussian data: small values of $\lambda_{\max}(\Sigma_{yy})$ correspond to significant whitening and, hence, to small spurious correlations, as predicted by Proposition 5.1.

We remark that our results concern the parameter $\hat{\theta}$ as defined in (1), which can be interpreted as the convergence point of an optimization algorithm such as gradient descent. This perspective differs from the prior work of Pezeshki

et al. (2021); Qiu et al. (2024), which focus on how spurious correlations evolve during training. On the other hand, in linear regression, solving the gradient flow equation $d\theta = -\nabla_\theta \mathcal{L}_\lambda(\theta) dt$, where $\mathcal{L}_\lambda(\theta)$ is defined as the argument of the $\arg\min$ in (1), gives

$$\theta(t) = \left(1 - \exp\left(-2\left(X^\top X/n + \lambda I\right) t\right)\right)\hat{\theta}. \quad (19)$$

Thus, the components of $\hat{\theta}$ aligned with the top eigen-spaces of $X^\top X$ converge earlier than the others. Hence, from a dynamical point of view, if $X^\top X \sim n\Sigma$, our results suggest that spurious features are learned faster the easier they are.

**Trade-off between $\mathcal{L}(\hat{\theta}_{\textbf{LR}}(\lambda))$ and $\mathcal{C}(\hat{\theta}_{\textbf{LR}}(\lambda))$.** Figure 2 shows that there is an interval of values for the regularization ($\lambda \sim 10^{-1}$) where the test loss $\mathcal{L}(\hat{\theta}_{\text{LR}}(\lambda))$ is decreasing in $\lambda$, while the spurious correlations $\mathcal{C}(\hat{\theta}_{\text{LR}}(\lambda))$ are increasing. This evidence suggests a natural trade-off between these two quantities, mediated by $\lambda$. To theoretically capture such trade-off, we first provide a non-asymptotic concentration bound for $\mathcal{L}(\hat{\theta}_{\text{LR}}(\lambda))$.

**Proposition 5.2.** *Let Assumption 4.1 hold, $n = \Theta(d)$ and $\mathcal{L}(\hat{\theta}_{\text{LR}}(\lambda))$ be the in-distribution test loss of the model $f_{\text{LR}}(\hat{\theta}_{\text{LR}}(\lambda), \cdot)$ for $\lambda > 0$. Set*

$$\mathcal{L}^\Sigma(\lambda) := \frac{\sigma^2 + \tau(\lambda)^2 \left\|(\Sigma + \tau(\lambda)I)^{-1} \Sigma^{1/2}\theta^*\right\|_2^2}{1 - \frac{\text{tr}\left((\Sigma + \tau(\lambda)I)^{-2}\Sigma^2\right)}{n}}, \quad (20)$$

*where $\tau(\lambda)$ is defined via (13). Then, for every $t \in (0, 1/2)$,*

$$\mathbb{P}_{Z,\mathcal{E}}\left(\left|\mathcal{L}(\hat{\theta}_{\text{LR}}(\lambda)) - \mathcal{L}^\Sigma(\lambda)\right| \geq t\right) \leq Cd\exp\left(-dt^4/C\right),$$

*where $C$ is an absolute constant.*

In words, Proposition 5.2 guarantees that $|\mathcal{L}(\hat{\theta}_{\text{LR}}(\lambda)) - \mathcal{L}^\Sigma(\lambda)| = o(1)$ with high probability. Its proof is an adaptation of Theorem 3.1 in (Han & Xu, 2023), and the details are in Appendix B.

---

[3]If $y$ has 0-mean, then $\mathbb{E}\|y\|_2^2 = \text{tr}(\Sigma_{yy})$, i.e., the trace captures the size of the pattern.

Armed with the non-asymptotic bounds of Theorem 4.3 and Proposition 5.2, we characterize the trade-off between $\mathcal{L}(\hat{\theta}_{LR}(\lambda))$ and $\mathcal{C}(\hat{\theta}_{LR}(\lambda))$ by studying the monotonicity of $\mathcal{L}^{\Sigma}(\lambda)$ and $\mathcal{C}^{\Sigma}(\lambda)$.

**Proposition 5.3.** *Let $\mathcal{C}^{\Sigma}(\lambda)$ and $\mathcal{L}^{\Sigma}(\lambda)$ be defined as in (12) and (20). Then, if $2d < n$, we have that $\mathcal{L}^{\Sigma}(\lambda)$ is monotonically decreasing in a right neighborhood of $\lambda = 0$, and there exists $\lambda_{\mathcal{L}} > 0$ such that $\mathcal{L}^{\Sigma}(\lambda)$ is monotonically increasing for $\lambda \geq \lambda_{\mathcal{L}}$. Furthermore, if $\Sigma_{xx} = I$, then $\mathcal{C}^{\Sigma}(\lambda)$ is non-negative and there exists $\lambda_{\mathcal{C}}$ such that $\mathcal{C}^{\Sigma}(\lambda)$ is monotonically increasing for $\lambda \leq \lambda_{\mathcal{C}}$. Finally, as long as*

$$\frac{2d}{n} \leq \frac{\lambda_{\min}(\Sigma)}{4} \min \left( 1, \frac{2\lambda_{\max}(\Sigma)/\sigma^2}{(\lambda_{\max}(\Sigma)/\lambda_{\min}(\Sigma) + 1)^2} \right), \tag{21}$$

*we have that $\lambda_{\mathcal{C}} \geq \lambda_{\mathcal{L}}$.*

In words, Proposition 5.3 shows that $\mathcal{C}^{\Sigma}(\lambda)$ grows with $\lambda$ at least until the regularization equals a value $\lambda_{\mathcal{C}}$. For example, in Figure 2, $\lambda_{\mathcal{C}} \sim 1$ for a Gaussian data and $\lambda_{\mathcal{C}} \sim 10$ for Color-MNIST. Furthermore, in this interval, $\mathcal{L}^{\Sigma}(\lambda)$ is initially decreasing and then increasing as $\lambda \geq \lambda_{\mathcal{L}}$. These trends in turn imply that the optimal value $\lambda_{\mathcal{L}}^*$ that minimizes the test loss is s.t. $\lambda_{\mathcal{L}}^* \in (0, \lambda_{\mathcal{C}}]$ – an interval where the spurious correlations are strictly positive and increasing.

The proof of Proposition 5.3 (whose details are in Appendix B) relies on the monotonicity of $\tau(\lambda)$ in $\lambda$, and the last statement follows from showing that $\tau(\lambda_{\mathcal{C}}) \geq \lambda_{\min}(S_x^{\Sigma}) \geq \lambda_{\min}(\Sigma) \geq \tau(\lambda_{\mathcal{L}})$. The upper bound on $2d/n$ in (21) is required to prove that $\lambda_{\min}(\Sigma) \geq \tau(\lambda_{\mathcal{L}})$ and, due to Assumption 4.1, it is implied by taking $n = \omega(d)$. We note that the latter scaling holds in standard datasets, e.g., MNIST ($n = 6 \cdot 10^4$, $2d \approx 2 \cdot 10^3$ when considering the 3 color channels) and CIFAR-10 ($n = 5 \cdot 10^4$, $2d \approx 3 \cdot 10^3$).

We conclude the section by noting that learning spurious correlations can be beneficial to minimize the (in-distribution) test loss. In fact, the spurious features in $y$ are effectively correlated with the labels, due to their correlation with the core feature $x$, and hence they can be helpful at prediction time. This phenomenon is numerically supported by Figures 3 and 4, where for a fixed value of $\lambda$, easier spurious features (or higher correlations) generate both higher values of $\mathcal{C}(\hat{\theta}_{LR}(\lambda))$ and lower values of $\mathcal{L}(\hat{\theta}_{LR}(\lambda))$. In words, while a blue background cannot strictly predict the label "boat", it is a useful feature in prediction as long as the boats in the test data tend to have a blue background.

The same conclusion does not hold for the out-of-distribution test loss, where the features $x$ and $y$ are sampled independently: the lower bound in (4) increases with $\mathcal{C}$. Figure 6 in Appendix E provides an additional numerical validation of this statement.

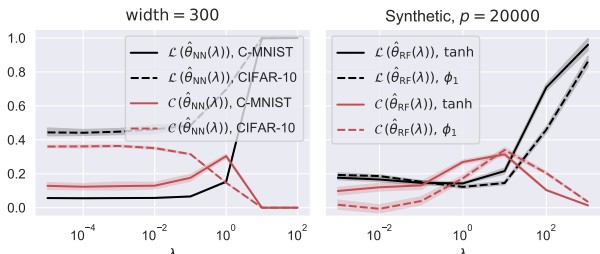

*Figure 5.* Test loss $\mathcal{L}(\hat{\theta}_{NN/RF}(\lambda))$ (black) and spurious correlations $\mathcal{C}(\hat{\theta}_{NN/RF}(\lambda))$ (red) as a function of $\lambda$. *Left:* 2-layer fully connected ReLU network, trained on the binary color(C)-MNIST and CIFAR-10 (boats and trucks). *Right:* RF model with $\tanh$ and $\phi_1 = h_1 + 0.1\, h_3$ activation. Implementation details are in Appendix F.

# 6. Role of Over-parameterization

Our analysis has so far focused on linear regression, highlighting the role of data covariance and regularization. However, moving to complex predictive models, such as neural networks, may lead to differences in the degree to which spurious correlations are learned. As an example, in the left panel of Figure 5, we train an over-parameterized two-layer neural network on the binary Color-MNIST and CIFAR-10 datasets, for different values of the regularizer $\lambda$. While for high values of $\lambda$ the results are qualitatively similar to the ones in Figure 2, a striking difference is that spurious correlations remain significant even when there is little to no regularization (i.e., $\lambda \approx 0$), in sharp contrast with Proposition 4.2. We also note that the phenomenon is in line with previous empirical work (Sagawa et al., 2020a).

We bridge the gap between linear regression and over-parameterized models by focusing on *random features*:

$$f_{RF}(\theta, z) = \phi(Vz)^{\top}\theta, \tag{22}$$

where $V$ is a $p \times 2d$ matrix s.t. $V_{i,j} \sim_{\text{i.i.d.}} \mathcal{N}(0, 1/(2d))$, and $\phi$ is an activation applied component-wise. The number of parameters of this model is $p$, as $V$ is a fixed random matrix and $\theta \in \mathbb{R}^p$ contains trainable parameters. The scaling of input data ($\text{tr}(\Sigma) = 2d$) and the variance of the entries of $V$ guarantee that the pre-activations of the model (*i.e.*, the entries of the vector $Vz \in \mathbb{R}^p$) are of constant order.

We consider the ERM in (1) with a quadratic loss

$$\hat{\theta}_{RF}(\lambda) = \arg\min_{\theta} \left( \frac{1}{n} \|\Phi\theta - G\|_2^2 + \lambda \|\theta\|_2^2 \right), \tag{23}$$

where we set $\Phi := [\phi(Vz_1), \ldots, \phi(Vz_n)]^{\top} \in \mathbb{R}^{n \times p}$. When $\lambda = 0$, if $\Phi\Phi^{\top}$ is invertible, the minimization above does not necessarily have a unique solution. In that case, we set $\hat{\theta}_{RF}(0)$ to be the solution obtained via gradient descent

with 0 initialization, which corresponds to the min-norm interpolator (see equation (33) in (Bartlett et al., 2021)). Then[4], we can write, for $\lambda \geq 0$,

$$\hat{\theta}_{\text{RF}}(\lambda) = \Phi^\top \left( \Phi \Phi^\top + n\lambda I \right)^{-1} G. \qquad (24)$$

**Assumption 6.1** (Activation function). The activation $\phi : \mathbb{R} \to \mathbb{R}$ is a non-linear, odd, Lipschitz function, such that its first Hermite coefficient $\mu_1 \neq 0$.

This choice is motivated by theoretical convenience and is similar to the one considered in (Hu & Lu, 2023). We believe that our result can be extended to a more general setting, as the ones in (Mei & Montanari, 2022; Mei et al., 2022), with a more involved analysis. We refer to (O'Donnell, 2014) for background on Hermite coefficients.

**Assumption 6.2** (Over-parameterization). We let $p$ grow s.t. $p = \omega \left( n \log^4 n \right)$ and $\log p = \Theta(\log n)$.

This requires the width of the model (and, hence, its number of parameters) to grow faster (by at least a poly-log factor) than the number of training samples.

Finally, our requirements on the data are less restrictive than those coming from Assumption 4.1.

**Assumption 6.3** (Data distribution, less restrictive). $\{z_i\}_{i=1}^n$ are $n$ i.i.d. samples from a mean-0, Lipschitz concentrated distribution $\mathcal{P}_{XY}$, with covariance $\Sigma$ s.t. $\operatorname{tr}(\Sigma) = 2d$. Furthermore, the labels $g_i$ are i.i.d. sub-Gaussian random variables.

Note that the labels $g_i$ are not required to follow a linear model $g_i = z_i^\top \theta^* + \epsilon_i$. The Lipschitz concentration property (see Appendix A for details) corresponds to data having well-behaved tails, it includes the distributions considered in Assumption 4.1, as well as the uniform distribution on the sphere or the hypercube (Vershynin, 2018), and it is a common requirement in the related literature (Nguyen et al., 2021; Bubeck & Sellke, 2021; Bombari et al., 2022).

**Theorem 6.4.** *Let Assumptions 6.1, 6.2, and 6.3 hold, $n = \Theta(d)$, and $z \in \mathbb{R}^{2d}$ be sampled from a distribution satisfying Assumption 6.3, not necessarily with the same covariance as $\mathcal{P}_{XY}$, independent from everything else. Let $f_{\text{RF}}(\hat{\theta}_{\text{RF}}(\lambda), z)$ be the RF model defined in (22) with $\hat{\theta}_{\text{RF}}(\lambda)$ given by (24), and $f_{\text{LR}}(\hat{\theta}_{\text{LR}}(\tilde{\lambda}), z)$ be the linear regression model defined in (5) with $\hat{\theta}_{\text{LR}}(\tilde{\lambda})$ given by (8). Then, for $\lambda \geq 0$,*

$$\left| f_{\text{RF}}(\hat{\theta}_{\text{RF}}(\lambda), z) - f_{\text{LR}}(\hat{\theta}_{\text{LR}}(\tilde{\lambda}), z) \right|$$

$$= \mathcal{O} \left( \frac{d^{1/4} \log d}{p^{1/4}} + \frac{\log^{3/2} d}{d^{1/8}} \right) = o(1), \qquad (25)$$

---

[4] $\Phi \Phi^\top$ is proved to be invertible with high probability in Lemma C.3.

*with probability at least $1 - C\sqrt{d}\log^2 d/\sqrt{p} - C\log^3 d/d^{1/4}$, where the* effective regularization *$\tilde{\lambda}$ is given by*

$$\tilde{\lambda} = \frac{2\tilde{\mu}^2 d}{\mu_1^2 n} + \frac{2d}{\mu_1^2 p}\lambda, \qquad (26)$$

*and $\tilde{\mu}^2 = \sum_{k \geq 2} \mu_k^2$, with $\mu_k$ denoting the $k$-th Hermite coefficient of $\phi$.*

In words, Theorem 6.4 shows that the over-parameterized RF model, when evaluated on a new test sample (not necessarily from the same distribution as the input data), is asymptotically equivalent to linear regression with regularization $\tilde{\lambda}$ given by (26). In particular, even in the ridgeless case ($\lambda = 0$), the RF model is equivalent to linear regression with strictly positive regularization. Thus, we expect the presence of spurious correlations, just like in Figure 5, since $\mathcal{C}(\hat{\theta}_{\text{RF}}(0))$ approaches $\mathcal{C}^\Sigma(\tilde{\lambda})$ with $\tilde{\lambda} > 0$. Notably, the effective regularization $\tilde{\lambda}$ depends on the activation $\phi$ via its Hermite coefficients, and it increases with the ratio $\tilde{\mu}^2/\mu_1^2$.

We point out some differences between Theorem 6.4 and earlier work on the equivalence of random features with regularized linear regression (Goldt et al., 2020; 2022; Hu & Lu, 2023; Montanari & Saeed, 2022). First, as mentioned in Section 2, existing results prove equivalence of training and test loss, which does not imply either the equivalence of the covariance in (3) nor the point-wise guarantee of (25). Second, existing results focus on the regime where $p$ and $n$ are proportional, while Assumption 6.2 requires $p = \omega \left( n \log^4 n \right)$. This reflects on the third difference which is in the proof technique: existing results use Lindeberg method, while our strategy (summarized below) relies on concentration tools.

**Proof sketch:** The proof builds on 3 core steps.

*Step 1:* We show that, with high probability,

$$\left\| \frac{\Phi \Phi^\top}{p} - \mu_1^2 \frac{ZZ^\top}{2d} - \tilde{\mu}^2 I \right\|_{\text{op}} = o\left( \frac{\lambda_{\min}\left(\Phi\Phi^\top\right)}{p} \right),$$

which is a consequence of the concentration of $\Phi\Phi^\top$ to its expectation with respect to $V$, then expressed in terms of the Hermite coefficients of $\phi$ (Lemmas C.2 and C.3).

*Step 2:* In Lemma C.5 we upper bound the term $\|\mathbb{E}_z[\tilde{\phi}(Vz)\tilde{\phi}(Vz)^\top]\|_{\text{op}}$ (where we set $\tilde{\phi}(\cdot) := \phi(\cdot) - \mu_1(\cdot)$), which is then used to show that

$$\left| \phi(Vz)^\top \hat{\theta}_{\text{RF}} - \mu_1 z^\top V^\top \hat{\theta}_{\text{RF}} \right| = o(1),$$

with high probability, due to Markov inequality. This means that the non-linear component of $\phi(Vz)$ has a negligible effect on the output (see (149) in Lemma C.7).

*Step 3:* Using a similar intuition, we use matrix Bernstein inequality (Lemma C.6) to show that $\|(\Phi - \mu_1 Z V^\top)V\|_{\text{op}}$ is small with high probability, so that

$$\left| \mu_1 z^\top V^\top \left( \Phi^\top - \mu_1 V Z^\top \right) \left( \Phi \Phi^\top + n\lambda I \right)^{-1} G \right| = o(1),$$

i.e., the non-linear component of $\Phi^\top$ in (24) is also negligible (see (145)).

Finally, by combining *Step 2* and *Step 3* with standard concentration arguments, we conclude that

$$\left| \phi(Vz)^\top \hat{\theta}_{\text{RF}} - \mu_1^2 p \frac{z^\top Z^\top}{2d} (\Phi \Phi^\top + n\lambda I)^{-1} G \right| = o(1),$$

and the thesis follows from *Step 1* and Woodbury matrix identity for the inverse. $\square$

The right panel of Figure 5 presents the test loss $\mathcal{L}(\hat{\theta}_{\text{RF}}(\lambda))$ (in black) and the spurious correlations $\mathcal{C}(\hat{\theta}_{\text{RF}}(\lambda))$ (in red) for two activation functions: $\tanh$ and $\phi_1 = h_1 + 0.1 h_3$, where $h_1$ and $h_3$ denote the first and third Hermite polynomials, respectively. Notice that this gives $\tilde{\mu}^2/\mu_1^2 \sim 0.1$ for $\tanh$, $\tilde{\mu}^2/\mu_1^2 \sim 0.01$ for $\phi_1$, and we take $d = 400$ and $n = 2000$. As expected, $\mathcal{C}(\hat{\theta}_{\text{RF}}(0)) > 0$ for the $\tanh$ activation function, since $\tilde{\lambda} \sim 0.05$ (which matches the corresponding value in Figure 2). On the other hand, $\mathcal{C}(\hat{\theta}_{\text{RF}}(0)) \sim 0$ for the activation $\phi_1$, since $\tilde{\lambda} \sim 0.005$. As $\lambda$ grows, $\mathcal{C}(\hat{\theta}_{\text{RF}}(\lambda))$ goes to 0 faster for the $\tanh$ activation function (which has higher $\tilde{\lambda}$), as predicted by the second upper bound in Proposition 5.1. These results match the qualitative behavior of the 2-layer neural network trained on the Color-MNIST dataset (left panel).

## 7. Conclusions

Our work provides a rigorous study of spurious correlations in high-dimensional regression models, with the purpose of connecting the statistical foundation of this phenomenon with its practical intuition. Specifically, we translate the problem into characterizing the deterministic object $\mathcal{C}^\Sigma(\lambda)$, which allows us to quantitatively capture the roles of ridge regularization, data covariance and over-parameterization.

An interesting future direction is to employ our principled approach to design algorithms that go beyond ERM to mitigate spurious correlations. In that regard, let us mention the usage of multiple models with different ridge regularization or early stopping (connected to ridge penalties, see (Raskutti et al., 2014)), in order to generate additional supervision. Here, having a quantitative control on which features (core or spurious) are learned by which model would allow to optimally use the extra labels (e.g., for up-weighting minority groups, as in (Liu et al., 2021)).

## Acknowledgements

Marco Mondelli is funded by the European Union (ERC, INF², project number 101161364). Views and opinions expressed are however those of the author(s) only and do not necessarily reflect those of the European Union or the European Research Council Executive Agency. Neither the European Union nor the granting authority can be held responsible for them. Simone Bombari is supported by a Google PhD fellowship. The authors would like to thank GuanWen Qiu for helpful discussions.

## Impact Statement

This paper presents work whose goal is to advance the field of machine learning. There are many potential societal consequences of our work, none which we feel must be specifically highlighted here.

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

# A. Additional Notation

We define a sub-Gaussian random variable according to Proposition 2.5.2 in (Vershynin, 2018), and $\|X\|_{\psi_2} := \inf\{t > 0 :$ $\mathbb{E}\left[\exp(X^2/t^2)\right] \le 2\}$. If $X \in \mathbb{R}^n$ is a random vector, then $\|X\|_{\psi_2} := \sup_{\|u\|_2=1} \left\|u^\top X\right\|_{\psi_2}$. When we state that a random variable or vector $X$ is sub-Gaussian, we implicitly mean $\|X\|_{\psi_2} = \mathcal{O}(1)$, i.e. its sub-Gaussian norm does not increase with the scalings of the problem.

We say that $X$ respects the Lipschitz concentration property if, for all 1-Lipschitz continuous functions $\varphi$, we have $\|\varphi(X) - \mathbb{E}\left[\varphi(X)\right]\|_{\psi_2} = \mathcal{O}(1)$. Notice that then, if $X$ is Lipschitz concentrated, then $X - \mathbb{E}[X]$ is sub-Gaussian.

Given two symmetric matrices $A, B$, we use the notation $A \succeq B$ if $A - B$ is p.s.d. Notice that if $A \succeq B \succ 0$, then we also have $B^{-1} \succeq A^{-1}$. We denote with $\|A\|_F$ the Frobenius norm of $A$, and with $\ker(A)$ its kernel space. If $A$ is a square matrix, we use the notation $\mathrm{diag}(A)$ to denote a matrix identical to $A$ on the diagonal, and 0 everywhere else. We let $A \circ B$ denote the Hadamard (component-wise) product between matrices, and $A^{\circ k}$ denote $A \circ A \circ ... \circ A$, where $A$ appears $k$ times.

# B. Proofs for Linear Regression

**Proof of Proposition 4.2.** Note that

$$\hat{\theta}_{\mathrm{LR}}(0) = \left(Z^\top Z\right)^{-1} Z^\top G. \tag{27}$$

Since we have $g_i = z_i^\top \theta^* + \epsilon_i$, (27) reads

$$\hat{\theta}_{\mathrm{LR}}(0) = \left(Z^\top Z\right)^{-1} Z^\top \left(Z\theta^* + \mathcal{E}\right) = \theta^* + \left(Z^\top Z\right)^{-1} Z^\top \mathcal{E}. \tag{28}$$

Then, we can plug this result in the definition of $\mathcal{C}(\hat{\theta})$ in (3) to obtain

$$
\begin{aligned}
\mathbb{E}_{\mathcal{E}}\left[\mathcal{C}(\hat{\theta}_{\mathrm{LR}}(0))\right] &= \mathbb{E}_{\mathcal{E}}\left[\mathrm{Cov}_{[x^\top, y^\top]^\top \sim P_{XY}, g=f_x^*(x), \tilde{x} \sim P_X}\left(f_{\mathrm{LR}}\left(\hat{\theta}_{\mathrm{LR}}(0), [\tilde{x}^\top, y^\top]^\top\right), g\right)\right] \\
&= \mathbb{E}_{\mathcal{E}}\left[\mathrm{Cov}_{[x^\top, y^\top]^\top \sim P_{XY}, \tilde{x} \sim P_X}\left([\tilde{x}^\top, y^\top]\hat{\theta}_{\mathrm{LR}}(0), x^\top \theta_x^*\right)\right] \\
&= \mathbb{E}_{\mathcal{E}}\left[\mathrm{Cov}_{[x,y] \sim P_{XY}, \tilde{x} \sim P_X}\left(\tilde{x}^\top \theta_x^* + [\tilde{x}^\top, y^\top]\left(Z^\top Z\right)^{-1} Z^\top \mathcal{E}, x^\top \theta_x^*\right)\right] \\
&= \mathrm{Cov}_{[x,y] \sim P_{XY}, \tilde{x} \sim P_X}\left(\tilde{x}^\top \theta_x^*, x^\top \theta_x^*\right) \\
&= 0,
\end{aligned}
\tag{29}
$$

where in the second line we used that $\mathcal{E}$ is independent from everything else, in fourth line we used $\mathbb{E}\left[\mathcal{E}\right] = 0$, and that $\mathcal{E}$ is independent from all the other random variables, and the last step holds since $\tilde{x}$ is independent from $x$.

For the second part of the statement we have that

$$
\begin{aligned}
\mathcal{C}(\hat{\theta}_{\mathrm{LR}}(0)) &= \mathrm{Cov}_{[x^\top, y^\top]^\top \sim P_{XY}, \tilde{x} \sim P_X}\left([\tilde{x}^\top, y^\top]\left(Z^\top Z\right)^{-1} Z^\top \mathcal{E}, x^\top \theta_x^*\right) \\
&= \mathrm{Cov}_{[x^\top, y^\top]^\top \sim P_{XY}, \tilde{x} \sim P_X}\left(\mathcal{E}^\top Z \left(Z^\top Z\right)^{-1} P_y[x^\top, y^\top]^\top, [x^\top, y^\top]\theta^*\right) \\
&= \mathcal{E}^\top Z \left(Z^\top Z\right)^{-1} P_y \Sigma \theta^*,
\end{aligned}
\tag{30}
$$

where in the second line we introduced $P_y \in \mathbb{R}^{2d \times 2d}$, defined as the projector on the last $d$ elements of the canonical basis in $\mathbb{R}^{2d}$. Then, since $\mathcal{E}$ is a sub-Gaussian vector (the entries are mean-0, i.i.d. sub-Gaussian) independent from everything else, we have that, with probability at least $1 - 2\exp\left(-c_1 \log^2 d\right)$,

$$\left|\mathcal{C}(\hat{\theta}_{\mathrm{LR}}(0))\right| \le \log d \left\|Z \left(Z^\top Z\right)^{-1} P_y \Sigma \theta^*\right\|_2 \le \log d \left\|Z \left(Z^\top Z\right)^{-1}\right\|_{\mathrm{op}} \|P_y\|_{\mathrm{op}} \|\Sigma\|_{\mathrm{op}} \|\theta^*\|_2 \le \frac{\log d \|\Sigma\|_{\mathrm{op}}}{\sqrt{\lambda_{\min}\left(Z^\top Z\right)}}, \tag{31}$$

where we used $\|P_y\|_{\mathrm{op}} = 1$ and $\|\theta^*\|_2 = 1$. Since $Z$ is a $n \times 2d$ matrix with independent rows having second moment $\Sigma$, by Theorem 5.39 in (Vershynin, 2012) (see Remark 5.40), we have that

$$\left\|\frac{Z^\top Z}{n} - \Sigma\right\|_{\mathrm{op}} = \mathcal{O}\left(\sqrt{\frac{d}{n}}\right) = o(1), \tag{32}$$

with probability at least $1 - 2\exp(-c_2 d)$. Hence, with this probability, by Weyl's inequality, we also have

$$\lambda_{\min}\left(Z^\top Z\right) \geq n\lambda_{\min}(\Sigma) - \left\|Z^\top Z - n\Sigma\right\|_{\mathrm{op}} = \Theta(n), \tag{33}$$

where the last step holds because of Assumption 4.1. Thus, we have that (31) reads

$$\left|\mathcal{C}(\hat{\theta}_{\mathrm{LR}}(0))\right| \leq \frac{\log d \, \|\Sigma\|_{\mathrm{op}}}{\sqrt{\lambda_{\min}\left(Z^\top Z\right)}} = \mathcal{O}\left(\frac{\log d}{\sqrt{n}}\right), \tag{34}$$

with probability at least $1 - 2\exp\left(-c_3 \log^2 d\right)$ over $Z$ and $\mathcal{E}$, which gives the desired result. $\qquad\square$

**Proof of Theorem 4.3.** As in (Han & Xu, 2023), we define the Gaussian sequence model $\hat{\theta}^\rho \in \mathbb{R}^{2d}$ as

$$\hat{\theta}^\rho = (\Sigma + \tau(\lambda)I)^{-1} \Sigma^{1/2}\left(\Sigma^{1/2}\theta^* + \frac{\gamma\rho}{\sqrt{2d}}\right), \tag{35}$$

where $\rho$ is a standard Gaussian vector in $\mathbb{R}^{2d}$ (the Gaussian sequence model is defined in Equation (1.5) in (Han & Xu, 2023), via the different notation $\hat{\mu}^{\mathrm{seq}}_{(\Sigma,\mu_0)}$, and our Gaussian vector $\rho$ is denoted as $g$ in the same equation). In the equation above, $\gamma > 0$ is implicitly defined via

$$\frac{n\gamma^2}{2d} = \sigma^2 + \mathbb{E}_\rho\left[\left\|\Sigma^{1/2}\left(\hat{\theta}^\rho - \theta^*\right)\right\|_2^2\right]. \tag{36}$$

On the other hand, following a similar argument as the one in (29), we have that, for every $\theta \in \mathbb{R}^{2d}$,

$$\begin{aligned}
\mathcal{C}(\theta) &= \mathrm{Cov}_{[x^\top, y^\top]^\top \sim P_{XY}, \tilde{x} \sim P_X}\left([\tilde{x}^\top, y^\top]\theta, x^\top\theta_x^*\right) \\
&= \theta^\top \mathbb{E}_{[x^\top, y^\top]^\top \sim P_{XY}, \tilde{x} \sim P_X}\left[[\tilde{x}^\top, y^\top]^\top x^\top\right]\theta_x^* \\
&= \theta^\top \mathbb{E}_{[x^\top, y^\top]^\top \sim P_{XY}}\left[[\mathbf{0}^\top, y^\top]^\top[x^\top, \mathbf{0}^\top]\right]\theta^* \\
&= \theta^\top P_y \mathbb{E}_{[x^\top, y^\top]^\top \sim P_{XY}}\left[[x^\top, y^\top]^\top[x^\top, y^\top]\right]\theta^* \\
&= \theta^\top P_y \Sigma \theta^*,
\end{aligned} \tag{37}$$

where the third line holds since $\tilde{x}$ has 0 mean and is independent from $x$ and $y$, and by definition of $\theta_x^*$, and the fourth line holds because $P_y[x^\top, y^\top]^\top = [\mathbf{0}^\top, y^\top]^\top$ and because the last $d$ entries of $\theta^*$ are 0 (*i.e.*, $P_y\theta^* = 0$). Thus, since we have that $\|P_y\Sigma\theta^*\|_2 \leq \|P_y\|_{\mathrm{op}}\|\Sigma\|_{\mathrm{op}}\|\theta^*\|_2 \leq \|\Sigma\|_{\mathrm{op}}$ because of Assumption 4.1 (and since $\|\theta^*\| \leq 1$), we have that $\mathcal{C}(\cdot): \mathbb{R}^{2d} \to \mathbb{R}$ is a $\|\Sigma\|_{\mathrm{op}}$-Lipschitz function.

Now, since $\mathcal{P}_{XY}$ is multivariate Gaussian, Theorem 2.3 of (Han & Xu, 2023) gives that, for any 1-Lipschitz function $\varphi: \mathbb{R}^{2d} \to \mathbb{R}$ (denoted as $g$ in (Han & Xu, 2023)), and any $t \in (0, 1/2)$,

$$\mathbb{P}_{Z,G}\left(\left|\varphi(\hat{\theta}_{\mathrm{LR}}(\lambda)) - \mathbb{E}_\rho\left[\varphi\left(\hat{\theta}^\rho\right)\right]\right| \geq t\right) \leq C_1 d \exp\left(-dt^4/C_1\right), \tag{38}$$

where $C_1$ is a constant depending on $\lambda_{\min}(\Sigma)$, $\|\Sigma\|_{\mathrm{op}}$, $\sigma^2$, and $n/d = \Theta(1)$. Since $\mathcal{C}(\cdot)$ is linear, notice that we have

$$\mathbb{E}_\rho\left[\mathcal{C}\left(\hat{\theta}^\rho\right)\right] = \mathcal{C}\left(\mathbb{E}_\rho\left[\hat{\theta}^\rho\right]\right) = \mathcal{C}\left((\Sigma + \tau(\lambda)I)^{-1}\Sigma\theta^*\right) = {\theta^*}^\top \Sigma(\Sigma + \tau(\lambda)I)^{-1}P_y\Sigma\theta^* = \mathcal{C}^\Sigma(\lambda), \tag{39}$$

where we used (37) in the third step, and the definition of $\mathcal{C}^\Sigma(\lambda)$ in (12) in the last one. Thus, setting $\varphi(\cdot)$ to be $\mathcal{C}(\cdot)/\|\Sigma\|_{\mathrm{op}}$, and plugging (39) in (38) we obtain

$$\mathbb{P}_{Z,G}\left(\frac{\left|\mathcal{C}(\hat{\theta}_{\mathrm{LR}}(\lambda)) - \mathcal{C}^\Sigma(\lambda)\right|}{\|\Sigma\|_{\mathrm{op}}} \geq t\right) \leq C_1 d \exp\left(-dt^4/C_1\right), \tag{40}$$

which gives the thesis after absorbing the constant $\|\Sigma\|_{\mathrm{op}}$ in $t$, and noticing that the bound is still true for $t \in (0, 1/2)$ since $\|\Sigma\|_{\mathrm{op}} \geq \mathrm{tr}(\Sigma)/2d = 1$ by Assumption 4.1. $\qquad\square$

**Proposition B.1.** *Let $\mathcal{C}^{\Sigma}(\lambda)$ be defined in (12), and let $S_x^{\Sigma+\tau(\lambda)I}$ be the Schur complement of $\Sigma + \tau(\lambda)I$ with respect to the top-left $d \times d$ block. Then, we have that*

$$\mathcal{C}^{\Sigma}(\lambda) = \tau(\lambda)\, \theta_x^{*\top} \left( \Sigma_{xx} + \tau(\lambda)I \right)^{-1} \Sigma_{xy} \left( S_x^{\Sigma+\tau(\lambda)I} \right)^{-1} \Sigma_{yx}\theta_x^*. \tag{41}$$

*Proof.* During the proof, to ease the notation, we will often leave implicit the dependence of $\tau$ on $\lambda$. Then, we can write

$$\begin{aligned}
\mathcal{C}^{\Sigma}(\lambda) &= \theta^{*\top} \Sigma \left( \Sigma + \tau I \right)^{-1} P_y \Sigma \theta^* \\
&= \theta^{*\top} \left( \Sigma + \tau I - \tau I \right) \left( \Sigma + \tau I \right)^{-1} P_y \Sigma \theta^* \\
&= -\tau \theta^{*\top} \left( \Sigma + \tau I \right)^{-1} P_y \Sigma \theta^* + \theta^{*\top} P_y \Sigma \theta^* \\
&= -\tau \theta^{*\top} \left( \Sigma + \tau I \right)^{-1} P_y \Sigma \theta^*,
\end{aligned} \tag{42}$$

where the last step holds since $P_y \theta^* = 0$. This expression can be further manipulated using the notation introduced in (14). We also introduce the following notation

$$\left( \Sigma + \tau I \right)^{-1} = \left( \begin{array}{c|c} \left[ \left( \Sigma + \tau I \right)^{-1} \right]_{xx} & \left[ \left( \Sigma + \tau I \right)^{-1} \right]_{xy} \\ \hline \left[ \left( \Sigma + \tau I \right)^{-1} \right]_{yx} & \left[ \left( \Sigma + \tau I \right)^{-1} \right]_{yy} \end{array} \right) \tag{43}$$

where we divided $\left( \Sigma + \tau I \right)^{-1}$ in four $d \times d$ blocks. Notice that, the expression in (42) only depends on $\left[ \left( \Sigma + \tau I \right)^{-1} \right]_{xy}$, i.e.,

$$\mathcal{C}^{\Sigma}(\lambda) = -\tau \theta^{*\top} P_x \left( \Sigma + \tau I \right)^{-1} P_y \Sigma \theta^* = -\tau \theta_x^{*\top} \left[ \left( \Sigma + \tau I \right)^{-1} \right]_{xy} \Sigma_{yx} \theta_x^*, \tag{44}$$

where we denoted the projector on the first $d$ elements of the canonical basis of $\mathbb{R}^{2d}$ as $P_x \in \mathbb{R}^{2d \times 2d}$. Exploiting the Schur complement $S_x^{\Sigma+\tau I}$, it holds that

$$\left[ \left( \Sigma + \tau I \right)^{-1} \right]_{xy} = - \left( \Sigma_{xx} + \tau I \right)^{-1} \Sigma_{xy} \left( S_x^{\Sigma+\tau I} \right)^{-1}, \tag{45}$$

which combined with (44) proves (41). $\qquad\square$

**Proof of Proposition 5.1.** During the proof, to ease the notation, we will often leave implicit the dependence of $\tau$ on $\lambda$. Then, according to (12), we have that

$$\left| \mathcal{C}^{\Sigma}(\lambda) \right| = \left| \theta^{*\top} \Sigma \left( \Sigma + \tau I \right)^{-1} P_y \Sigma P_x \theta^* \right| \leq \|\theta^*\|_2^2 \left\| \Sigma \left( \Sigma + \tau I \right)^{-1} \right\|_{\mathrm{op}} \| P_y \Sigma P_x \|_{\mathrm{op}} \leq \| \Sigma_{yx} \|_{\mathrm{op}}, \tag{46}$$

and

$$\left| \mathcal{C}^{\Sigma}(\lambda) \right| = \left| \theta^{*\top} \Sigma \left( \Sigma + \tau I \right)^{-1} P_y \Sigma \theta^* \right| \leq \|\theta^*\|_2^2 \|\Sigma\|_{\mathrm{op}}^2 \frac{1}{\lambda_{\min}(\Sigma) + \tau} \leq \frac{\lambda_{\max}(\Sigma)^2}{\tau}. \tag{47}$$

Then, using (41), we get

$$
\begin{aligned}
\mathcal{C}^{\Sigma}(\lambda) &= \tau\, \theta_x^{*\top} (\Sigma_{xx} + \tau I)^{-1} \Sigma_{xy} \left(S_x^{\Sigma + \tau I}\right)^{-1} \Sigma_{yx} \theta_x^* \\
&= \tau\, \theta_x^{*\top} (\Sigma_{xx} + \tau I)^{-1/2} (\Sigma_{xx} + \tau I)^{-1/2} \Sigma_{xy} \left(S_x^{\Sigma + \tau I}\right)^{-1} \Sigma_{yx} (\Sigma_{xx} + \tau I)^{-1/2} (\Sigma_{xx} + \tau I)^{1/2} \theta_x^* \\
&\leq \tau \left\| (\Sigma_{xx} + \tau I)^{-1/2} \theta_x^* \right\|_2 \left\| (\Sigma_{xx} + \tau I)^{1/2} \theta_x^* \right\|_2 \left\| (\Sigma_{xx} + \tau I)^{-1/2} \Sigma_{xy} \left(S_x^{\Sigma + \tau I}\right)^{-1} \Sigma_{yx} (\Sigma_{xx} + \tau I)^{-1/2} \right\|_{\mathrm{op}} \\
&\leq \tau\, \frac{1}{\sqrt{\lambda_{\min}(\Sigma_{xx}) + \tau}} \sqrt{\theta_x^{*\top} \Sigma_{xx} \theta_x^* + \tau} \left\| (\Sigma_{xx} + \tau I)^{-1/2} \Sigma_{xy} \left(S_x^{\Sigma + \tau I}\right)^{-1} \Sigma_{yx} (\Sigma_{xx} + \tau I)^{-1/2} \right\|_{\mathrm{op}} \\
&\leq \tau\, \frac{\sqrt{\mathbb{E}_{x \sim P_X}\left[ (x^\top \theta_x^*)^2 \right] + \tau}}{\sqrt{\lambda_{\min}(\Sigma_{xx}) + \tau}} \frac{\left\| (\Sigma_{xx} + \tau I)^{-1/2} \Sigma_{xy} \right\|_{\mathrm{op}}^2}{\lambda_{\min}\left(S_x^{\Sigma + \tau I}\right)} \\
&= \tau\, \frac{\sqrt{\mathbb{E}_{g = x^\top \theta_x^* + \epsilon}[g^2] - \sigma^2 + \tau}}{\sqrt{\lambda_{\min}(\Sigma_{xx}) + \tau}} \frac{\lambda_{\max}\left( \Sigma_{yx} (\Sigma_{xx} + \tau I)^{-1} \Sigma_{xy} \right)}{\lambda_{\min}\left( \Sigma_{yy} + \tau I - \Sigma_{yx}(\Sigma_{xx} + \tau I)^{-1} \Sigma_{xy} \right)} \\
&\leq \tau\, \frac{\sqrt{\mathbb{E}_g[g^2] - \sigma^2 + \tau}}{\sqrt{\lambda_{\min}(\Sigma_{xx}) + \tau}} \frac{\lambda_{\max}\left( \Sigma_{yx} \Sigma_{xx}^{-1} \Sigma_{xy} \right)}{\lambda_{\min}\left( \Sigma_{yy} + \tau I - \Sigma_{yx}\Sigma_{xx}^{-1} \Sigma_{xy} \right)} \\
&= \tau\, \frac{\sqrt{\mathbb{E}_g[g^2] - \sigma^2 + \tau}}{\sqrt{\lambda_{\min}(\Sigma_{xx}) + \tau}} \frac{\lambda_{\max}(\Sigma_{yy}) - \lambda_{\min}\left(S_x^{\Sigma}\right)}{\lambda_{\min}\left(S_x^{\Sigma}\right) + \tau} \\
&\leq \tau \sqrt{\mathrm{Var}(g) - \sigma^2}\, \frac{\lambda_{\max}(\Sigma_{yy}) - \lambda_{\min}\left(S_x^{\Sigma}\right)}{\lambda_{\min}\left(S_x^{\Sigma}\right) \sqrt{\lambda_{\min}(\Sigma_{xx})}},
\end{aligned}
$$

(48)

where in the fifth line we denoted with $\mathcal{P}_X$ the marginal distribution of the core feature $x$, and $\mathbb{E}_g[\cdot]$ from the sixth line on denotes an expectation with respect to $g$ distributed as the labels of the model. The last step simplifies the expression with respect to $\tau$, and it holds since $\mathrm{Var}(g) - \sigma^2 = \theta_x^{*\top} \Sigma_{xx} \theta_x^* \geq \lambda_{\min}(\Sigma_{xx})$. This, together with (46) and (47) gives the desired result. $\qquad\square$

**Proof of Proposition 5.2.** During the proof, to ease the notation, we will often leave implicit the dependence of $\tau$ on $\lambda$. Then, as in (Han & Xu, 2023) and in the proof of Theorem 4.3, we define the Gaussian sequence model $\hat{\theta}^\rho \in \mathbb{R}^{2d}$ as (35) where $\rho$ is a standard Gaussian vector in $\mathbb{R}^{2d}$ and $\gamma > 0$ is implicitly defined via

$$
\frac{n\gamma^2}{2d} = \sigma^2 + \mathbb{E}_\rho \left[ \left\| \Sigma^{1/2} \left( \hat{\theta}^\rho - \theta^* \right) \right\|_2^2 \right] = \sigma^2 + \left\| \Sigma^{1/2} \left( (\Sigma + \tau I)^{-1} \Sigma - I \right) \theta^* \right\|_2^2 + \frac{\gamma^2}{2d} \mathrm{tr}\left( (\Sigma + \tau I)^{-2} \Sigma^2 \right), \quad (49)
$$

which also reads

$$
\frac{n\gamma^2}{2d} = \frac{\sigma^2 + \tau^2 \left\| (\Sigma + \tau I)^{-1} \Sigma^{1/2} \theta^* \right\|_2^2}{1 - \frac{\mathrm{tr}\left( (\Sigma + \tau I)^{-2} \Sigma^2 \right)}{n}}. \quad (50)
$$

Then, due to Theorem 3.1 of (Han & Xu, 2023) on the prediction risk, since $\mathcal{P}_{XY}$ is a multivariate Gaussian due to Assumption 4.1, we have that, for any $t \in (0, 1/2)$,

$$
\mathbb{P}_{Z,G}\left( \left| \mathcal{L}(\hat{\theta}_{\mathrm{LR}}(\lambda)) - \mathcal{L}^{\Sigma}(\lambda) \right| \geq t \right) \leq C d \exp\left( -d t^4 / C \right), \quad (51)
$$

where $C$ is a positive constant depending on $\lambda_{\min}(\Sigma)$, $\|\Sigma\|_{\mathrm{op}}$, $\sigma^2$, and $n/d = \Theta(1)$. $\qquad\square$

**Proof of Proposition 5.3** As $\tau(\lambda)$ is an increasing function of $\lambda$, all the statements on the monotonicity of $\mathcal{L}^{\Sigma}(\lambda)$ and $\mathcal{C}^{\Sigma}(\lambda)$ can be proved by showing monotonicity w.r.t. $\tau$ (whose dependence w.r.t. $\lambda$ is left implicit throughout the argument).

In particular, we have

$$
\frac{\mathrm{d}\mathcal{L}^{\Sigma}(\lambda)}{\mathrm{d}\tau} = \frac{\frac{\mathrm{d}}{\mathrm{d}\tau}\left(\tau^2\left\|(\Sigma+\tau I)^{-1}\Sigma^{1/2}\theta^*\right\|_2^2\right)\left(1-\frac{\mathrm{tr}\left((\Sigma+\tau I)^{-2}\Sigma^2\right)}{n}\right)}{\left(1-\frac{\mathrm{tr}\left((\Sigma+\tau I)^{-2}\Sigma^2\right)}{n}\right)^2}
$$
$$
-\frac{\left(\sigma^2+\tau^2\left\|(\Sigma+\tau I)^{-1}\Sigma^{1/2}\theta^*\right\|_2^2\right)\frac{\mathrm{d}}{\mathrm{d}\tau}\left(1-\frac{\mathrm{tr}\left((\Sigma+\tau I)^{-2}\Sigma^2\right)}{n}\right)}{\left(1-\frac{\mathrm{tr}\left((\Sigma+\tau I)^{-2}\Sigma^2\right)}{n}\right)^2}. \tag{52}
$$

To study the sign of the above expression, it suffices to focus on the numerators, as the denominator is always positive.

Note that the RHS of (13) is smaller or equal to $2d/n$; thus, as $2d < n$, we also get $\tau \leq \lambda\left(1-2d/n\right)^{-1}$, which implies $\tau(\lambda) \to 0$ as $\lambda \to 0$. Hence, to show that $\mathcal{L}^{\Sigma}(\lambda)$ is monotonically decreasing in a right neighborhood of $\lambda = 0$, it suffices to show that (52) evaluated in $\tau = 0$ is strictly negative. For $\tau = 0$, the first factor in the numerator of the first term in (52) is 0, as the following chain of equalities holds:

$$
\frac{\mathrm{d}}{\mathrm{d}\tau}\left(\tau^2\left\|(\Sigma+\tau I)^{-1}\Sigma^{1/2}\theta^*\right\|_2^2\right) = \frac{\mathrm{d}}{\mathrm{d}\tau}\left(\left\|(\Sigma/\tau+I)^{-1}\Sigma^{1/2}\theta^*\right\|_2^2\right)
$$
$$
= \theta^{*\top}\frac{\mathrm{d}}{\mathrm{d}\tau}\left(\Sigma/\tau+I\right)^{-2}\Sigma\theta^*
$$
$$
= -\theta^{*\top}\left(\Sigma/\tau+I\right)^{-2}\left(\frac{\mathrm{d}}{\mathrm{d}\tau}\left(\Sigma/\tau+I\right)^2\right)\left(\Sigma/\tau+I\right)^{-2}\Sigma\theta^* \tag{53}
$$
$$
= -\theta^{*\top}\left(\Sigma/\tau+I\right)^{-2}\left(-\frac{2\Sigma}{\tau^2}\left(\Sigma/\tau+I\right)\right)\left(\Sigma/\tau+I\right)^{-2}\Sigma\theta^*
$$
$$
= 2\tau\theta^{*\top}\left(\Sigma+\tau I\right)^{-3}\Sigma^2\theta^*.
$$

Furthermore, the second term gives

$$
-\sigma^2\frac{\mathrm{d}}{\mathrm{d}\tau}\left(1-\frac{\mathrm{tr}\left((\Sigma+\tau I)^{-2}\Sigma^2\right)}{n}\right) = \frac{\sigma^2}{n}\frac{\mathrm{d}}{\mathrm{d}\tau}\left(\sum_{k=1}^{2d}\frac{\lambda_k^2}{(\lambda_k+\tau)^2}\right) = -\frac{2\sigma^2}{n}\sum_{k=1}^{2d}\frac{\lambda_k^2}{(\lambda_k+\tau)^3} < 0, \tag{54}
$$

where $\lambda_k$ denotes the $k$-th eigenvalue of $\Sigma$. This gives the first claim.

To show that there exists $\lambda_{\mathcal{L}} > 0$ such that $\mathcal{L}^{\Sigma}(\lambda)$ is monotonically increasing for $\lambda \geq \lambda_{\mathcal{L}}$ we will show that the derivative of $\mathcal{L}^{\Sigma}(\lambda)$ with respect to $\tau$ is positive for all $\tau \geq \tau_{\mathcal{L}} := \tau(\lambda_{\mathcal{L}})$. For simplicity, in the rest of the argument we use the notation $\lambda_{\max}$ and $\lambda_{\min}$ to indicate the largest and smallest eigenvalues of $\Sigma$, respectively. Instead, the notation $\lambda_{\min}(\cdot)$ still represents the smallest eigenvalue of its argument. For the first factor of the first term of (52), continuing from (53), we have

$$
\frac{\mathrm{d}}{\mathrm{d}\tau}\left(\tau^2\left\|(\Sigma+\tau I)^{-1}\Sigma^{1/2}\theta^*\right\|_2^2\right) \geq 2\frac{1}{\lambda_{\max}\left(\Sigma/\tau+I\right)^3}\lambda_{\min}\left(\Sigma/\tau\right)^2 = \frac{2\lambda_{\min}^2}{\tau^2\left(\lambda_{\max}/\tau+1\right)^3}. \tag{55}
$$

For the second factor of the first term of (52), we have

$$
1-\frac{\mathrm{tr}\left((\Sigma+\tau I)^{-2}\Sigma^2\right)}{n} = 1-\frac{1}{n}\sum_{k=1}^{2d}\frac{\lambda_k^2}{(\lambda_k+\tau)^2} \geq 1-\frac{2d\lambda_{\max}^2}{n\tau^2}. \tag{56}
$$

For the first factor of the second term of (52), we have

$$
\tau^2\left\|(\Sigma+\tau I)^{-1}\Sigma^{1/2}\theta^*\right\|_2^2 \leq \tau^2\frac{\lambda_{\max}}{(\lambda_{\min}+\tau)^2} \tag{57}
$$

For the second factor of the second term of (52), we have

$$\frac{\mathrm{d}}{\mathrm{d}\tau}\left(1 - \frac{\mathrm{tr}\left((\Sigma + \tau I)^{-2}\Sigma^2\right)}{n}\right) = -\frac{1}{n}\frac{\mathrm{d}}{\mathrm{d}\tau}\left(\sum_{k=1}^{2d}\frac{\lambda_k^2}{(\lambda_k + \tau)^2}\right) = \frac{2}{n}\sum_{k=1}^{2d}\frac{\lambda_k^2}{(\lambda_k + \tau)^3} \leq \frac{4d\lambda_{\max}^2}{n\tau^3}. \tag{58}$$

Thus, putting together (52), (55), (56), (57), and (58), the monotonicity of $\mathcal{L}^\Sigma(\lambda)$ is implied by

$$\frac{2\lambda_{\min}^2}{\tau^2\left(\lambda_{\max}/\tau + 1\right)^3}\left(1 - \frac{2d\lambda_{\max}^2}{n\tau^2}\right) \overset{?}{\geq} \left(\sigma^2 + \tau^2\frac{\lambda_{\max}}{(\lambda_{\min} + \tau)^2}\right)\frac{4d\lambda_{\max}^2}{n\tau^3}. \tag{59}$$

Now, we have that the above inequality holds for sufficiently large $\tau$: the LHS is $\Theta(1/\tau^2)$ (considering fixed the other quantities), while the RHS is $\Theta(1/\tau^3)$; and the desired statement is therefore proved.

Next, we set $\tau_\mathcal{C} := \tau(\lambda_\mathcal{C}) = \sqrt{\lambda_{\min}\left(S_x^\Sigma\right)}$ and show that $\mathcal{C}^\Sigma(\lambda)$ is monotonically increasing for $\tau \in [0, \tau_\mathcal{C}]$. Plugging $\Sigma_{xx} = I$ in (41) we get

$$\mathcal{C}^\Sigma(\lambda) = \tau\,\theta_x^{*\top}\left(\Sigma_x + \tau I\right)^{-1}\Sigma_{xy}\left(S_x^{\Sigma + \tau I}\right)^{-1}\Sigma_{yx}\theta_x^* = \frac{\tau}{1 + \tau}\theta_x^{*\top}\Sigma_{xy}\left(\Sigma_{yy} + \tau I - \frac{\Sigma_{yx}\Sigma_{xy}}{1 + \tau}\right)^{-1}\Sigma_{yx}\theta_x^*. \tag{60}$$

By the product rule, and introducing the shorthand $A(\tau) = \Sigma_{yy} + \tau I - \frac{\Sigma_{xy}^\top \Sigma_{xy}}{1+\tau}$, we have

$$\begin{aligned}
\frac{\mathrm{d}\mathcal{C}^\Sigma(\lambda)}{\mathrm{d}\tau} &= \left(\frac{\mathrm{d}}{\mathrm{d}\tau}\left(\frac{\tau}{1+\tau}\right)\right)\left(\theta_x^{*\top}\Sigma_{xy}A(\tau)^{-1}\Sigma_{xy}^\top\theta_x^*\right) + \left(\frac{\tau}{1+\tau}\right)\left(\frac{\mathrm{d}}{\mathrm{d}\tau}\left(\theta_x^{*\top}\Sigma_{xy}A(\tau)^{-1}\Sigma_{xy}^\top\theta_x^*\right)\right) \\
&= \frac{1}{(1+\tau)^2}\theta_x^{*\top}\Sigma_{xy}A(\tau)^{-1}\Sigma_{xy}^\top\theta_x^* + \left(\frac{\tau}{1+\tau}\right)\left(\theta_x^{*\top}\Sigma_{xy}A(\tau)^{-1}\left(-\frac{\mathrm{d}}{\mathrm{d}\tau}A(\tau)\right)A(\tau)^{-1}\Sigma_{xy}^\top\theta_x^*\right) \\
&= \frac{1}{(1+\tau)^2}\theta_x^{*\top}\Sigma_{xy}A(\tau)^{-1}\Sigma_{xy}^\top\theta_x^* - \frac{\tau}{1+\tau}\left(\theta_x^{*\top}\Sigma_{xy}A(\tau)^{-1}\left(I + \frac{\Sigma_{xy}^\top\Sigma_{xy}}{(1+\tau)^2}\right)A(\tau)^{-1}\Sigma_{xy}^\top\theta_x^*\right) \\
&= \frac{1}{(1+\tau)}\theta_x^{*\top}\Sigma_{xy}A(\tau)^{-1}\left(\frac{A(\tau)}{1+\tau} - \tau\left(I + \frac{\Sigma_{xy}^\top\Sigma_{xy}}{(1+\tau)^2}\right)\right)A(\tau)^{-1}\Sigma_{xy}^\top\theta_x^* \\
&= \frac{1}{(1+\tau)}\theta_x^{*\top}\Sigma_{xy}A(\tau)^{-1}\left(\frac{\Sigma_{yy} + \tau I}{1+\tau} - \frac{\Sigma_{xy}^\top\Sigma_{xy}}{(1+\tau)^2} - \tau I - \tau\frac{\Sigma_{xy}^\top\Sigma_{xy}}{(1+\tau)^2}\right)A(\tau)^{-1}\Sigma_{xy}^\top\theta_x^* \\
&= \frac{1}{(1+\tau)^2}\theta_x^{*\top}\Sigma_{xy}A(\tau)^{-1}\left(\Sigma_{yy} - \Sigma_{xy}^\top\Sigma_{xy} - \tau^2 I\right)A(\tau)^{-1}\Sigma_{xy}^\top\theta_x^* \\
&= \frac{1}{(1+\tau)^2}\theta_x^{*\top}\Sigma_{xy}A(\tau)^{-1}\left(S_x^\Sigma - \tau^2 I\right)A(\tau)^{-1}\Sigma_{xy}^\top\theta_x^*,
\end{aligned} \tag{61}$$

where in the second line we used the identity $\frac{\mathrm{d}}{\mathrm{d}\tau}\left(A(\tau)^{-1}\right) = A(\tau)^{-1}\left(-\frac{\mathrm{d}}{\mathrm{d}\tau}A(\tau)\right)A(\tau)^{-1}$. Then, if $\tau \leq \sqrt{\lambda_{\min}\left(S_x^\Sigma\right)} = \tau_\mathcal{C}$, we have that $\left(S_x^\Sigma - \tau^2 I\right)$ is p.s.d., which in turn implies $\frac{\mathrm{d}\mathcal{C}^\Sigma(\lambda)}{\mathrm{d}\tau} \geq 0$, thus giving the desired claim. The non-negativity of $\mathcal{C}^\Sigma(\lambda)$ readily follows from (60).

For the last statement, setting $\tau_\mathcal{L} = \lambda_{\min}(\Sigma)$ we show that $\mathcal{L}^\Sigma(\lambda)$ is monotonically increasing for all $\tau \in [\tau_\mathcal{L}, +\infty)$ as long as the additional bound on $2d/n$ holds. As $\lambda_{\min}\left(S_x^\Sigma\right) \leq \lambda_{\min}\left(\Sigma_{yy}\right) \leq \mathrm{tr}\left(\Sigma_{yy}\right)/d = \mathrm{tr}\left(\Sigma - \Sigma_{xx}\right)/d = 1$, we also have

$$\tau_\mathcal{C} = \sqrt{\lambda_{\min}\left(S_x^\Sigma\right)} \geq \lambda_{\min}\left(S_x^\Sigma\right) \geq \lambda_{\min}(\Sigma) = \tau_\mathcal{L}, \tag{62}$$

where the second inequality follows from Lemma B.3. Thus, from the monotonicity of $\tau(\lambda)$ in $\lambda$, the final result readily follows.

It remains to prove the monotonicity of $\mathcal{L}^\Sigma(\lambda)$ in $[\lambda_{\min}(\Sigma), +\infty)$. To do so, we again study the sign of (52).

For the first factor of the first term of (52), we have

$$
\begin{aligned}
\frac{\mathrm{d}}{\mathrm{d}\tau}\left(\tau^2 \left\|(\Sigma + \tau I)^{-1} \Sigma^{1/2}\theta^*\right\|_2^2\right) &= 2\theta^{*\top}\left(\Sigma/\tau + I\right)^{-3}\left(\Sigma/\tau\right)^2 \theta^* \\
&= 2\theta^{*\top}\Sigma^{1/2}\left(\Sigma + \tau I\right)^{-1}\left(\Sigma + \tau I\right)^{-1}\tau\Sigma\left(\Sigma + \tau I\right)^{-1}\Sigma^{1/2}\theta^* \\
&\geq \frac{2}{\tau}\lambda_{\min}\left(\Sigma\left(\Sigma + \tau\right)^{-1}\right)\tau^2\left\|(\Sigma + \tau I)^{-1}\Sigma^{1/2}\theta^*\right\|_2^2 \\
&= \frac{2}{\tau}\frac{\lambda_{\min}}{\lambda_{\min} + \tau}\tau^2\left\|(\Sigma + \tau I)^{-1}\Sigma^{1/2}\theta^*\right\|_2^2.
\end{aligned}
\tag{63}
$$

For the second factor of the first term of (52), we have

$$
1 - \frac{\mathrm{tr}\left((\Sigma + \tau I)^{-2}\Sigma^2\right)}{n} = 1 - \frac{1}{n}\sum_{k=1}^{2d}\frac{\lambda_k^2}{(\lambda_k + \tau)^2} \geq 1 - \frac{2d}{n}.
\tag{64}
$$

For the second factor of the second term of (52), we have

$$
\frac{\mathrm{d}}{\mathrm{d}\tau}\left(1 - \frac{\mathrm{tr}\left((\Sigma + \tau I)^{-2}\Sigma^2\right)}{n}\right) = \frac{2}{n}\sum_{k=1}^{2d}\frac{\lambda_k^2}{(\lambda_k + \tau)^3} \leq \frac{2}{n}\frac{1}{\tau(\lambda_{\min} + \tau)}\sum_{k=1}^{2d}\frac{\lambda_k^2}{(\lambda_k + \tau)} \leq \frac{4d}{n\tau(\lambda_{\min} + \tau)}.
\tag{65}
$$

Thus, putting together (52), (63), (64), and (65), the monotonicity of $\mathcal{L}^\Sigma(\lambda)$ is implied by

$$
\frac{2}{\tau}\frac{\lambda_{\min}}{\lambda_{\min} + \tau}\tau^2\left\|(\Sigma + \tau I)^{-1}\Sigma^{1/2}\theta^*\right\|_2^2\left(1 - \frac{2d}{n}\right) \overset{?}{\geq} \left(\sigma^2 + \tau^2\left\|(\Sigma + \tau I)^{-1}\Sigma^{1/2}\theta^*\right\|_2^2\right)\frac{4d}{n\tau(\lambda_{\min} + \tau)}
\tag{66}
$$

Since we assumed that $2d/n \leq \lambda_{\min}/4 \leq 1/4$, we have

$$
\frac{2}{\tau}\frac{\lambda_{\min}}{\lambda_{\min} + \tau}\left(1 - \frac{2d}{n}\right) - \frac{4d}{n\tau(\lambda_{\min} + \tau)} = \frac{2}{\tau}\frac{\lambda_{\min}}{\lambda_{\min} + \tau}\left(1 - \frac{2d}{n} - \frac{2d}{n\lambda_{\min}}\right) \geq \frac{1}{\tau}\frac{\lambda_{\min}}{\lambda_{\min} + \tau},
\tag{67}
$$

and

$$
\begin{aligned}
\tau^2\left\|(\Sigma + \tau I)^{-1}\Sigma^{1/2}\theta^*\right\|_2^2 &\geq \lambda_{\min}\left(\tau^2\Sigma\left(\Sigma + \tau I\right)^{-2}\right) \\
&= \min_k \frac{\tau^2\lambda_k}{(\lambda_k + \tau)^2} \\
&= \min_k \frac{\lambda_k}{(\lambda_k/\tau + 1)^2} \\
&\geq \min_k \frac{\lambda_k}{(\lambda_k/\lambda_{\min} + 1)^2} \\
&= \lambda_{\min}\min_k \frac{\lambda_k/\lambda_{\min}}{(\lambda_k/\lambda_{\min} + 1)^2} \\
&= \frac{\lambda_{\max}}{(\lambda_{\max}/\lambda_{\min} + 1)^2},
\end{aligned}
\tag{68}
$$

where in the fourth line we used that $\tau \geq \lambda_{\min}$, and in the last step we used that $f(x) := x/(x+1)^2$ is decreasing for $x \geq 1$.

Thus, using (67) and (68) gives that (66) is implied by

$$
\frac{\lambda_{\max}}{(\lambda_{\max}/\lambda_{\min} + 1)^2}\frac{1}{\tau}\frac{\lambda_{\min}}{\lambda_{\min} + \tau} \overset{?}{\geq} \sigma^2\frac{4d}{n\tau(\lambda_{\min} + \tau)},
\tag{69}
$$

which holds since we assumed

$$
\frac{2d}{n} \leq \frac{1}{2\sigma^2}\frac{\lambda_{\max}\lambda_{\min}}{(\lambda_{\max}/\lambda_{\min} + 1)^2}.
\tag{70}
$$

$\square$

**B.1. Proofs on $S_x^\Sigma$**

For completeness, in this section we prove two known results about $S_x^\Sigma$.

**Lemma B.2.** *Let $z = \begin{bmatrix} x^\top, y^\top \end{bmatrix}^\top \sim \mathcal{P}_{XY}$ be distributed according to a mean-0, multivariate Gaussian distribution with covariance $\Sigma$, such that $\Sigma$ is invertible. Then, the Schur complement $S_x^\Sigma$ of $\Sigma$ with respect to the top left block $\Sigma_{xx}$ (see (14)) corresponds to the conditional covariance of $y$ given $x$, i.e.,*

$$S_x^\Sigma = \text{Cov}\left(y|x=\bar{x}\right) = \mathbb{E}_{y|x=\bar{x}}\left[\left(y - \mathbb{E}_{y|x=\bar{x}}[y]\right)\left(y - \mathbb{E}_{y|x=\bar{x}}[y]\right)^\top\right]. \tag{71}$$

*Proof.* Consider the expression $z^\top \Sigma^{-1} z$. According to the notation in (14) and in (43), we have

$$z^\top \Sigma^{-1} z = x^\top \left[\Sigma^{-1}\right]_{xx} x + y^\top \left[\Sigma^{-1}\right]_{yy} y + x^\top \left[\Sigma^{-1}\right]_{xy} y + y^\top \left[\Sigma^{-1}\right]_{yx} x. \tag{72}$$

Then, the formulas for the inverse of a block matrix give

$$z^\top \Sigma^{-1} z$$
$$= x^\top \left(\Sigma_{xx}^{-1} + \Sigma_{xx}^{-1}\Sigma_{xy}S_x^{\Sigma^{-1}}\Sigma_{yx}\Sigma_{xx}^{-1}\right) x + y^\top S_x^{\Sigma^{-1}} y + x^\top \left(-\Sigma_{xx}^{-1}\Sigma_{xy}S_x^{\Sigma^{-1}}\right) y + y^\top \left(-S_x^{\Sigma^{-1}}\Sigma_{yx}\Sigma_{xx}^{-1}\right) x \tag{73}$$
$$= x^\top \Sigma_{xx}^{-1} x + \left(y - \Sigma_{yx}\Sigma_{xx}^{-1}x\right)^\top S_x^{\Sigma^{-1}} \left(y - \Sigma_{yx}\Sigma_{xx}^{-1}x\right).$$

Then, denoting with $p(x,y)$ and $p(x)$ the probability density functions of $z = \begin{bmatrix} x^\top, y^\top \end{bmatrix}^\top$ and $x$ respectively, we get that the probability density function of $y$ conditioned on $x$ takes the form

$$\begin{aligned}
p(y|x) &= \frac{p(x,y)}{p(x)} \\
&= \frac{\sqrt{(2\pi)^d \det(\Sigma_{xx})}}{\sqrt{(2\pi)^{2d} \det(\Sigma)}} \frac{\exp\left(-\begin{bmatrix} x^\top, y^\top \end{bmatrix} \Sigma^{-1} \begin{bmatrix} x^\top, y^\top \end{bmatrix}^\top /2\right)}{\exp\left(-x^\top \Sigma_{xx}^{-1} x/2\right)} \\
&= \frac{1}{\sqrt{(2\pi)^d \det(S_x^\Sigma)}} \exp\left(-\begin{bmatrix} x^\top, y^\top \end{bmatrix} \Sigma^{-1} \begin{bmatrix} x^\top, y^\top \end{bmatrix}^\top /2 + x^\top \Sigma_{xx}^{-1} x/2\right) \\
&= \frac{\exp\left(-\left(y - \Sigma_{yx}\Sigma_{xx}^{-1}x\right)^\top S_x^{\Sigma^{-1}} \left(y - \Sigma_{yx}\Sigma_{xx}^{-1}x\right)/2\right)}{\sqrt{(2\pi)^d \det(S_x^\Sigma)}},
\end{aligned} \tag{74}$$

where we used Schur formula for the determinants in the third line, and (73) in the last step. Thus, we have that $p(y|x)$ describes the density of a multivariate Gaussian random variable, with covariance $S_x^\Sigma$. $\square$

**Lemma B.3.** *Let $\Sigma \in \mathbb{R}^{2d \times 2d}$ be a p.s.d., invertible matrix. Then, the Schur complement $S_x^\Sigma \in \mathbb{R}^{d \times d}$ of $\Sigma$ with respect to the top left block $\Sigma_{xx}$ (see (14)) is such that*

$$\lambda_{\min}\left(S_x^\Sigma\right) \geq \lambda_{\min}\left(\Sigma\right). \tag{75}$$

*Proof.* Let $\Gamma \in \mathbb{R}^{2d \times d}$ be the rank-$d$ matrix defined as

$$\Gamma = \begin{pmatrix} \Sigma_{xx}^{1/2} \\ \Sigma_{yx}\Sigma_{xx}^{-1/2} \end{pmatrix}, \tag{76}$$

and $S \in \mathbb{R}^{2d \times 2d}$ as the matrix containing $S_x^\Sigma$ in its bottom-right $d \times d$ block, and 0 everywhere else. Then, we have that

$$\Sigma = S + \Gamma\Gamma^\top, \tag{77}$$

where both $S$ and $\Gamma\Gamma^\top$ are rank-$d$ p.s.d. matrices.

Denoting by $\lambda_k(S)$ the $k$-th largest eigenvalue of $S$, by the Courant–Fischer–Weyl min-max principle, we can write

$$\lambda_k(S) = \max_{W, \dim(W)=k} \min_{u \in W, \|u\|_2=1} \left(u^\top S u\right), \tag{78}$$

where with $W$ we denote a generic $k$-dimensional subspace of $\mathbb{R}^{2d}$. Thus, the desired result follows from

$$
\begin{aligned}
\lambda_{\min}(\Sigma) &= \lambda_{\min}\left(S + \Gamma\Gamma^\top\right) \\
&= \min_{\|u\|_2=1} u^\top \left(S + \Gamma\Gamma^\top\right) u \\
&\leq \min_{u \in \ker(\Gamma\Gamma^\top),\, \|u\|_2=1} u^\top \left(S + \Gamma\Gamma^\top\right) u \\
&= \min_{u \in \ker(\Gamma\Gamma^\top),\, \|u\|_2=1} u^\top S u \\
&\leq \max_{W,\, \dim(W)=d} \min_{u \in W,\, \|u\|_2=1} u^\top S u \\
&= \lambda_d(S) \\
&= \lambda_{\min}\left(S_x^\Sigma\right),
\end{aligned}
\tag{79}
$$

where the last step holds since the $d$ smallest eigenvalues of $S$ are equal to $0$, and the $d$ largest correspond to the ones of $S_x^\Sigma$. $\qquad\square$

## B.2. Remarks on Assumption 4.1

Our results on linear regression rely on Assumption 4.1, and in particular on the training samples to be normally distributed. This assumption is made for technical convenience, as the concentration results in Theorem 4.3 and Proposition 5.2 still hold under the following milder requirement.

**Assumption B.4** (Data distribution). The input samples $\{z_i\}_{i=1}^n$ are $n$ i.i.d. samples from a mean-0, sub-Gaussian distribution $\mathcal{P}_{XY}$, such that

1. its covariance $\Sigma \in \mathbb{R}^{2d \times 2d}$ is invertible, with $\lambda_{\max}(\Sigma) = \mathcal{O}(1)$, $\lambda_{\min}(\Sigma) = \Omega(1)$, and $\mathrm{tr}(\Sigma) = 2d$;

2. for $z \sim P_Z$, the random variable $\Sigma^{-1/2}z$ has independent, mean-0, unit variance, sub-Gaussian entries.

This assumption resembles the requirements A-B in Section 2.2 in (Han & Xu, 2023), where we also included the scaling of the trace. To formally state the equivalent of Theorem 4.3 and Proposition 5.2, one also has to enforce the following technical condition on the true parameter $\theta^*$.

**Assumption B.5.** Let $\delta = 1/72$, then we assume that

$$
\theta^* \text{ s.t. } \left\| \Sigma^{1/2}\tau(\Sigma + \tau I)^{-1}\theta^* \right\|_\infty \leq C d^{\delta - 1/2}.
\tag{80}
$$

In Proposition 10.3 in (Han & Xu, 2023), it is shown that this condition excludes a negligible fraction $(Ce^{-n^{2\delta}/C})$ of the $\theta^*$ on the unit ball. Since we set $\delta = 1/72$, following the same arguments of the proofs of Theorem 4.3 and Proposition 5.2, we have that Theorems 2.4 and 3.1 in (Han & Xu, 2023) imply the results below.

**Theorem B.6.** *Let Assumptions 6.3 and B.5 hold, and let $n = \Theta(d)$. Let $\hat{\theta}_{\mathrm{LR}}(\lambda)$ be defined as in (8), and let $\mathcal{C}(\hat{\theta}_{\mathrm{LR}}(\lambda))$ be the amount of spurious correlations learned by the model $f_{\mathrm{LR}}\left(\hat{\theta}_{\mathrm{LR}}(\lambda), \cdot\right)$ as defined in (3). Then, for any $\lambda > 0$, we have that, for every $t \in (0, 1/2)$,*

$$
\mathbb{P}_{Z,G}\left( \left| \mathcal{C}(\hat{\theta}_{\mathrm{LR}}(\lambda)) - \mathcal{C}^\Sigma(\lambda) \right| \geq t \right) \leq C t^{-13} d^{-1/8},
\tag{81}
$$

*where $\mathcal{C}^\Sigma(\lambda)$ is defined in (12), and $C$ is a an absolute constant.*

**Proposition B.7.** *Let Assumptions 6.3 and B.5 hold, and let $n = \Theta(d)$. Let $\hat{\theta}_{\mathrm{LR}}(\lambda)$ be defined as in (8), and let $\mathcal{L}(\hat{\theta}_{\mathrm{LR}}(\lambda))$ be the in-distribution test loss of the model $f_{\mathrm{LR}}(\hat{\theta}_{\mathrm{LR}}(\lambda), \cdot)$ as defined in (5). Then, , for any $\lambda > 0$, we have that, for every $t \in (0, 1/2)$,*

$$
\mathbb{P}_{Z,G}\left( \left| \mathcal{L}(\hat{\theta}_{\mathrm{LR}}(\lambda)) - \mathcal{L}^\Sigma(\lambda) \right| \geq t \right) \leq C t^{-c} d^{-1/6.5},
\tag{82}
$$

*where $\mathcal{L}^\Sigma(\lambda)$ is defined in (20), and $C$ and $c$ are positive absolute constants.*

## C. Proofs for Random Features

**Lemma C.1.** *We have that*

$$
\|V\|_{\mathrm{op}} = \mathcal{O}\left( \sqrt{\frac{p}{d}} \right),
\tag{83}
$$

$$\|Z\|_{\mathrm{op}} = \mathcal{O}\left(\sqrt{d}\right), \tag{84}$$

*with probability at least $1 - 2\exp\left(-cd\right)$ over $V$ and $Z$, where $c$ is an absolute constant. Furthermore, for every $i \in [n]$, we have*

$$\left\|\|z_i\|_2 - \sqrt{2d}\right\|_{\psi_2} = \mathcal{O}\left(1\right). \tag{85}$$

*Proof.* $V$ has independent, mean-0, unit variance, sub-Gaussian entries. Then, the first statement is a direct consequences of Theorem 4.4.5 of (Vershynin, 2018) and of the scaling $d = o(p)$.

By Assumption 6.3, we have that $Z$ has i.i.d. mean-0, Lipschitz concentrated rows. This property also implies that the rows are i.i.d. sub-Gaussian. Thus, by Remark 5.40 in (Vershynin, 2012), we have that

$$\left\|Z^\top Z - n\Sigma\right\|_{\mathrm{op}} = \mathcal{O}\left(n\frac{d}{n}\right) = \mathcal{O}\left(d\right), \tag{86}$$

with probability at least $1 - 2\exp\left(-c_1 d\right)$. Then, conditioning on this high probability event, by Weyl's inequality, we have

$$\left\|Z^\top Z\right\|_{\mathrm{op}} \le \|n\Sigma\|_{\mathrm{op}} + \left\|Z^\top Z - n\Sigma\right\|_{\mathrm{op}} = \mathcal{O}\left(d\right), \tag{87}$$

where the last step follows from the argument used to prove $\|\Sigma\|_{\mathrm{op}} = \mathcal{O}\left(1\right)$ in Lemma C.1 in (Bombari & Mondelli, 2024).

For the last statement, we have

$$2d = \mathrm{tr}(\Sigma) = \mathrm{tr}\left(\mathbb{E}\left[zz^\top\right]\right) = \mathbb{E}\left[\mathrm{tr}\left(zz^\top\right)\right] = \mathbb{E}\left[\mathrm{tr}\left(z^\top z\right)\right] = \mathbb{E}\left[\|z\|_2^2\right], \tag{88}$$

where we used the cyclic property of the trace. Furthermore, we have

$$\left\|\|z\|_2 - \mathbb{E}\left[\|z\|_2\right]\right\|_{\psi_2} = \mathcal{O}\left(1\right), \tag{89}$$

since $z$ is Lipschitz concentrated. Then,

$$0 \le 2d - \mathbb{E}\left[\|z\|_2\right]^2 = \mathbb{E}\left[\left(\|z\|_2 - \mathbb{E}\left[\|z\|_2\right]\right)^2\right] \le C_1, \tag{90}$$

for some absolute constant $C_1$. Thus, as $\sqrt{1-x} \ge 1 - x$ for $x \in [0, 1]$, we obtain

$$1 - \frac{C_1}{2d} \le \sqrt{1 - \frac{C_1}{2d}} \le \frac{\mathbb{E}\left[\|z\|_2\right]}{\sqrt{2d}} \le 1. \tag{91}$$

Plugging this last result in (89) gives the desired claim. $\qquad\square$

**Lemma C.2.** *We have that, denoting with $\tilde{\mu}^2 = \sum_{k \ge 2} \mu_k^2$, with $\mu_k$ denoting the $k$-th Hermite coefficient of $\phi$,*

$$\left\|\mathbb{E}_V\left[\Phi\Phi^\top\right] - p\left(\mu_1^2 \frac{ZZ^\top}{2d} + \tilde{\mu}^2 I\right)\right\|_{\mathrm{op}} = \mathcal{O}\left(\frac{p\log^3 d}{\sqrt{d}}\right), \tag{92}$$

*with probability at least $1 - 2\exp\left(-c\log^2 d\right)$ over $Z$, where $c$ is an absolute constant.*

*Proof.* For all $i \in [n]$, we define the functions $\phi^{(i)} : \mathbb{R} \to \mathbb{R}$ as $\phi^{(i)}(\cdot) = \phi(\|z_i\|_2 \cdot /\sqrt{2d})$. Note that $\phi^{(i)}$ is odd, since $\phi$ is odd by Assumption 6.1. Thus, denoting with $\mu_k^{(i)}$ the $k$-th Hermite coefficient of $\phi^{(i)}$, for every $i \in [n]$, we have that $\mu_k^{(i)} = 0$ for all even $k$. This implies that, by denoting with $v$ a random vector distributed as the rows of $V$, *i.e.*, $\sqrt{2d}\,v$ is a standard Gaussian vector, we have

$$
\begin{aligned}
\left[\mathbb{E}_V\left[\Phi\Phi^\top\right]\right]_{ij} &= p\,\mathbb{E}_v\left[\phi(z_i^\top v)\phi(z_j^\top v)\right] \\
&= p\,\mathbb{E}_v\left[\phi^{(i)}\left(\frac{z_i^\top}{\|z_i\|_2}\sqrt{2d}v\right)\phi^{(j)}\left(\frac{z_j^\top}{\|z_j\|_2}\sqrt{2d}v\right)\right] \\
&= p\sum_{k=0}^{+\infty} \mu_k^{(i)}\mu_k^{(j)}\left(\frac{z_i^\top z_j}{\|z_i\|_2\|z_j\|_2}\right)^k \\
&= p\mu_1^{(i)}\mu_1^{(j)}\frac{z_i^\top z_j}{\|z_i\|_2\|z_j\|_2} + p\sum_{k\ge3}\mu_k^{(i)}\mu_k^{(j)}\left(\frac{z_i^\top z_j}{\|z_i\|_2\|z_j\|_2}\right)^k.
\end{aligned}
\tag{93}
$$

Then, denoting with $D_k \in \mathbb{R}^{n \times n}$ the diagonal matrix containing $\mu_k^{(i)} / \|z_i\|_2^k$ in its $i$-th entry, we can write

$$\mathbb{E}_V \left[ \Phi \Phi^\top \right] = p D_1 Z Z^\top D_1 + p \sum_{k \geq 3} D_k \left( Z Z^\top \right)^{\circ k} D_k. \tag{94}$$

Notice that, due to the last statement in Lemma C.1, we have that, jointly for all $i \in [n]$,

$$\left| \frac{\|z_i\|_2}{\sqrt{2d}} - 1 \right| = \mathcal{O}\left( \frac{\log d}{\sqrt{d}} \right), \tag{95}$$

with probability at least $1 - 2\exp(-c_1 \log^2 d)$. Then, conditioning on such high probability event and denoting with $\rho$ a standard Gaussian random variable, for all $i \in [n]$ we have

$$\begin{aligned}
\left| \mu_1^{(i)} - \mu_1 \right| &= \left| \mathbb{E}_\rho \left[ \rho \phi^{(i)}(\rho) \right] - \mathbb{E}_\rho \left[ \rho \phi(\rho) \right] \right| \\
&= \left| \mathbb{E}_\rho \left[ \rho \left( \phi \left( \frac{\|z_i\|_2}{\sqrt{2d}} \rho \right) - \phi(\rho) \right) \right] \right| \\
&= \left| \mathbb{E}_\rho \left[ \rho \left( \phi \left( \left( \frac{\|z_i\|_2}{\sqrt{2d}} - 1 \right) \rho + \rho \right) - \phi(\rho) \right) \right] \right| \\
&\leq \mathbb{E}_\rho \left[ |\rho| \left| \phi \left( \left( \frac{\|z_i\|_2}{\sqrt{2d}} - 1 \right) \rho + \rho \right) - \phi(\rho) \right| \right] \\
&\leq L \mathbb{E}_\rho \left[ |\rho| \left| \frac{\|z_i\|_2}{\sqrt{2d}} - 1 \right| |\rho| \right] \\
&= L \left| \frac{\|z_i\|_2}{\sqrt{2d}} - 1 \right| \mathbb{E}_\rho \left[ \rho^2 \right] \\
&= \mathcal{O}\left( \frac{\log d}{\sqrt{d}} \right),
\end{aligned} \tag{96}$$

where we used Jensen's inequality in the fourth line, the $L$-Lipschitzness of $\phi$ in the fifth line, and (95) in the last step. With a similar approach, denoting with $\|\cdot\|_{L^2}$ the $L^2$ norm with respect to the Gaussian measure, we have that for all $i \in [n]$

$$\begin{aligned}
\left| \left\| \phi^{(i)} \right\|_{L^2} - \|\phi\|_{L^2} \right| &\leq \left\| \phi^{(i)} - \phi \right\|_{L^2} \\
&= \mathbb{E}_\rho \left[ \left( \phi^{(i)}(\rho) - \phi(\rho) \right)^2 \right]^{1/2} \\
&= \mathbb{E}_\rho \left[ \left( \phi \left( \left( \frac{\|z_i\|_2}{\sqrt{2d}} - 1 \right) \rho + \rho \right) - \phi(\rho) \right)^2 \right]^{1/2} \\
&\leq L \left| \frac{\|z_i\|_2}{\sqrt{2d}} - 1 \right| \mathbb{E}_\rho \left[ \rho^2 \right]^{1/2} \\
&= \mathcal{O}\left( \frac{\log d}{\sqrt{d}} \right),
\end{aligned} \tag{97}$$

which directly implies that, for all $i \in [n]$, $\left\| \phi^{(i)} \right\|_{L^2} = \sum_{k \geq 0} \left( \mu_k^{(i)} \right)^2 = \Theta(1)$, and that

$$\left| \sum_{k \geq 3} \left( \mu_k^{(i)} \right)^2 - \sum_{k \geq 3} \mu_k^2 \right| \leq \left| \left\| \phi^{(i)} \right\|_{L^2}^2 - \|\phi\|_{L^2}^2 \right| + \left| \left( \mu_1^{(i)} \right)^2 - \mu_1^2 \right| = \mathcal{O}\left( \frac{\log d}{\sqrt{d}} \right). \tag{98}$$

Thus, we are ready to estimate the operator norm of the off-diagonal part of the second term on the RHS of (94), specifically

$$
\begin{aligned}
&\left\| \sum_{k \geq 3} D_k \left(ZZ^\top\right)^{\circ k} D_k - \mathrm{diag}\left( \sum_{k \geq 3} D_k \left(ZZ^\top\right)^{\circ k} D_k \right) \right\|_{\mathrm{op}} \\
&\leq \sum_{k \geq 3} \left\| D_k \left(ZZ^\top\right)^{\circ k} D_k - \mathrm{diag}\left( D_k \left(ZZ^\top\right)^{\circ k} D_k \right) \right\|_F \\
&\leq \sum_{k \geq 3} \max_{i \neq j} \left( \frac{|z_i^\top z_j|}{\|z_i\|_2 \|z_j\|_2} \right)^k \left( \sum_{i \in [n], j \in [n]} \left( \mu_k^{(i)} \mu_k^{(j)} \right)^2 \right)^{1/2} \\
&\leq \max_{i \neq j} \left( \frac{|z_i^\top z_j|}{\|z_i\|_2 \|z_j\|_2} \right)^3 \sum_{i=0}^{n} \sum_{k \geq 3} \left( \mu_k^{(i)} \right)^2 \\
&= \mathcal{O}\left( \frac{1}{d^{3/2}} \log^3 d \, n \right) = \mathcal{O}\left( \frac{\log^3 d}{\sqrt{d}} \right),
\end{aligned}
\tag{99}
$$

where in the first step we replaced the operator norm with the Frobenius norm, and used triangle inequality; in the fifth line we used that $\|z_i\|_2 = \Theta(\sqrt{d})$ for all $i \in [n]$ (true because of (95)), and that jointly for all $i \neq j$ we have $|z_i^\top z_j| / \|z_j\|_2 = \mathcal{O}(\log d)$ with probability at least $1 - 2\exp(-c_2 \log^2 d)$ since the $z_i$-s are independent sub-Gaussian vectors (since they are mean-0 and Lipschitz concentrated). The diagonal part of the second term on the RHS of (94) respects

$$
\left\| \mathrm{diag}\left( \sum_{k \geq 3} D_k \left(ZZ^\top\right)^{\circ k} D_k \right) - \tilde{\mu}^2 I \right\|_{\mathrm{op}} = \max_{i \in [n]} \left| \sum_{k \geq 3} \left( \mu_k^{(i)} \right)^2 - \sum_{k \geq 3} \mu_k^2 \right| = \mathcal{O}\left( \frac{\log d}{\sqrt{d}} \right),
\tag{100}
$$

because of (98). Lastly, notice that,

$$
\begin{aligned}
\left\| D_1 ZZ^\top D_1 - \mu_1^2 \frac{ZZ^\top}{2d} \right\|_{\mathrm{op}} &= \sup_{\|u\|_2 = 1} \left| u^\top D_1 ZZ^\top D_1 u - \mu_1^2 u^\top \frac{ZZ^\top}{2d} u \right| \\
&= \sup_{\|u\|_2 = 1} \left| \left\| Z^\top D_1 u \right\|_2^2 - \mu_1^2 \left\| \frac{Z^\top}{\sqrt{2d}} u \right\|_2^2 \right| \\
&\leq \sup_{\|u\|_2 = 1} \left( \left\| Z^\top D_1 u \right\|_2 + \mu_1 \left\| \frac{Z^\top}{\sqrt{2d}} u \right\|_2 \right) \sup_{\|u\|_2 = 1} \left( \left\| Z^\top D_1 u - \mu_1 \frac{Z^\top}{\sqrt{2d}} u \right\|_2 \right) \\
&\leq \left( \left\| Z^\top D_1 \right\|_{\mathrm{op}} + \mu_1 \left\| \frac{Z^\top}{\sqrt{2d}} \right\|_{\mathrm{op}} \right) \left\| Z^\top D_1 - \mu_1 \frac{Z^\top}{\sqrt{2d}} \right\|_{\mathrm{op}} \\
&\leq \left( \|Z\|_{\mathrm{op}} \|D_1\|_{\mathrm{op}} + \mu_1 \frac{\|Z\|_{\mathrm{op}}}{\sqrt{2d}} \right) \|Z\|_{\mathrm{op}} \left\| D_1 - \frac{\mu_1}{\sqrt{2d}} \right\|_{\mathrm{op}}.
\end{aligned}
\tag{101}
$$

By Lemma C.1, we have that $\|Z\|_{\mathrm{op}} = \mathcal{O}\left(\sqrt{d}\right)$ with probability at least $1 - 2\exp\left(-c_2 d\right)$, and since $\|z_i\|_2 = \Theta(\sqrt{d})$ and $\mu_1^{(i)} = \mathcal{O}(1)$ for all $i \in [n]$ (true because of (95) and (96) respectively), we have that $\|D_1\|_{\mathrm{op}} = \mathcal{O}\left(1/\sqrt{d}\right)$. Furthermore, we have

$$
\begin{aligned}
\left\| D_1 - \frac{\mu_1}{\sqrt{2d}} \right\|_{\mathrm{op}} &= \max_i \left| \frac{\mu_1^{(i)}}{\|z_i\|_2} - \frac{\mu_1}{\sqrt{2d}} \right| \\
&\leq \max_i \frac{1}{\|z_i\|_2} \left( \left| \mu_1^{(i)} - \mu_1 \right| + \mu_1 \left| 1 - \frac{\|z_i\|_2}{\sqrt{2d}} \right| \right) \\
&= \mathcal{O}\left( \frac{\log d}{d} \right),
\end{aligned}
\tag{102}
$$

where the last step is a consequence of (95) and (96). Then, we have that (101) reads

$$\left\| D_1 Z Z^\top D_1 - \mu_1^2 \frac{Z Z^\top}{2d} \right\|_{\text{op}} = \mathcal{O}\left( \frac{\log d}{\sqrt{d}} \right). \tag{103}$$

A standard application of the triangle inequality to (99), (100) and (103) gives

$$\left\| \left( D_1 Z Z^\top D_1 + \sum_{k \geq 3} D_k \left( Z Z^\top \right)^{\circ k} D_k \right) - \left( \mu_1^2 \frac{Z Z^\top}{2d} + \tilde{\mu}^2 I \right) \right\|_{\text{op}} = \mathcal{O}\left( \frac{\log^3 d}{\sqrt{d}} \right), \tag{104}$$

with probability at least $1 - 2 \exp\left( -c_3 \log^2 d \right)$ over $Z$ (where we used $\mu_2 = 0$ since $\phi$ is odd), which readily gives the thesis when plugged in (94). $\qquad \square$

**Lemma C.3.** *We have that*

$$\|\Phi\|_{\text{op}} = \mathcal{O}\left( \sqrt{p} \right), \tag{105}$$

$$\left\| \Phi\Phi^\top - \mathbb{E}_V \left[ \Phi\Phi^\top \right] \right\|_{\text{op}} = \mathcal{O}\left( \sqrt{pd} \right), \tag{106}$$

$$\lambda_{\min}\left( \Phi\Phi^\top \right) = \Omega\left( p \right), \tag{107}$$

*with probability at least $1 - 2 \exp\left( -c \log^2 d \right)$ over $Z$ and $V$, where $c$ is an absolute constant.*

*Proof.* $\Phi^\top$ is a matrix with i.i.d. rows in the probability space of $V$. In particular, its $i$-th row takes the form

$$\left[ \Phi^\top \right]_{i:} = \phi\left( Z V_{i:} \right) = \phi\left( Z V_{i:} \right) - \mathbb{E}_V \left[ \phi\left( Z V_{i:} \right) \right], \tag{108}$$

where the last step holds since the (Gaussian) distribution of $V_{i:}$ is symmetric and $\phi$ is an odd function by Assumption 6.1. Then, since $\sqrt{2d} V_{i:}$ is a standard Gaussian (and hence Lipschitz concentrated) random vector, and $\phi$ is a Lipschitz continuous function, we have that

$$\left\| \left[ \Phi^\top \right]_{i:} \right\|_{\psi_2} = \mathcal{O}\left( \frac{\|Z\|_{\text{op}}}{\sqrt{d}} \right) = \mathcal{O}\left( 1 \right), \tag{109}$$

where the $\|\cdot\|_{\psi_2}$ is meant on the probability space of $V$, and the second step holds with probability at least $1 - 2 \exp\left( -c_1 d \right)$ over $Z$ due to Lemma C.1. Conditioning on this high probability event, $\Phi^\top$ is a $p \times n$ matrix whose rows are i.i.d. mean-0 sub-Gaussian random vectors in $\mathbb{R}^n$. Then, by Lemma B.7 in (Bombari et al., 2022), we have

$$\left\| \Phi^\top \right\|_{\text{op}} = \mathcal{O}\left( \sqrt{n} + \sqrt{p} \right) = \mathcal{O}\left( \sqrt{p} \right), \tag{110}$$

with probability at least $1 - 2 \exp(-c_2 n)$ over $V$, where the second step holds because $n = o(p)$.

For the second part of the proof, we again follow the argument in Lemma B.7 in (Bombari et al., 2022), which in turn exploits the discussion in Remark 5.40 in (Vershynin, 2012), and conclude that

$$\left\| \Phi\Phi^\top - \mathbb{E}_V \left[ \Phi\Phi^\top \right] \right\|_{\text{op}} = \mathcal{O}\left( p\sqrt{\frac{n}{p}} \right) = \mathcal{O}\left( \sqrt{pn} \right) = \mathcal{O}\left( \sqrt{pd} \right), \tag{111}$$

with probability at least $1 - 2 \exp(-c_3 n)$ over $Z$ and $V$.

For the last statement, Lemma C.2 and Weyl's inequality imply that, with probability at least $1 - 2 \exp\left( -c_2 \log^2 d \right)$ over $Z$ we have

$$\begin{aligned}
\lambda_{\min}\left( \Phi\Phi^\top \right) &\geq p\, \lambda_{\min}\left( \mu_1^2 \frac{Z Z^\top}{2d} + \tilde{\mu}^2 I \right) - \left\| \mathbb{E}_V \left[ \Phi\Phi^\top \right] - p \left( \mu_1^2 \frac{Z Z^\top}{2d} + \tilde{\mu}^2 I \right) \right\|_{\text{op}} - \left\| \Phi\Phi^\top - \mathbb{E}_V \left[ \Phi\Phi^\top \right] \right\|_{\text{op}} \\
&\geq p\tilde{\mu}^2 - \mathcal{O}\left( \frac{p \log^3 d}{\sqrt{d}} \right) - \mathcal{O}\left( \sqrt{pd} \right) = \Omega(p),
\end{aligned} \tag{112}$$

where the last step is true since $\tilde{\mu} \neq 0$, as $\phi$ is non-linear by Assumption 6.1. $\qquad \square$

**Lemma C.4.** *Let $\tilde{\phi} : \mathbb{R} \to \mathbb{R}$ be defined as $\tilde{\phi}(\cdot) := \phi(\cdot) - \mu_1(\cdot)$, and set*

$$n' = \min\left(\left\lfloor \frac{p}{\log^4 p} \right\rfloor, \left\lfloor \frac{d^{3/2}}{\log^3 d} \right\rfloor\right). \tag{113}$$

*Let $\{\hat{z}_i\}_{i=1}^{n'}$ be $n'$ i.i.d. random variables sampled from a distribution respecting Assumption 6.3, not necessarily with the same covariance as $\mathcal{P}_{XY}$, and independent from $V$. Then, if $\tilde{\Phi}_{n'} \in \mathbb{R}^{n' \times p}$ is defined as the matrix containing $\tilde{\phi}(V\hat{z}_i)$ in its i-th row, we have that*

$$\left\| \tilde{\Phi}_{n'} \right\|_{op} = \mathcal{O}\left(\sqrt{p}\right), \tag{114}$$

*with probability at least $1 - 2\exp\left(-c\log^2 d\right)$ over $\{\hat{z}_i\}_{i=1}^{n'}$ and $V$, where $c$ is an absolute constant.*

*Proof.* The proof follows the same strategy as Lemma C.8 in (Bombari & Mondelli, 2024), with the only difference that they work under their Assumption 1.2, *i.e.* that the data is normalized as $\|\hat{z}_i\|_2 = \sqrt{2d}$ (in our notation). This difference, however, does not affect the result. We can in fact condition on the high probability event that all $\hat{z}_i$ are such that $\|\hat{z}_i\|_2 = \Theta(\sqrt{d})$, which holds with probability at least $1 - 2\exp(-cd)$ by Lemma C.1, and proceed in the same way (as their Equation (C.78) now holds) until their Equation (C.81), which requires their Lemma C.7, *i.e.* that

$$\left\| \mathbb{E}_V \left[ \tilde{\Phi}_{n'} \tilde{\Phi}_{n'}^\top \right] \right\|_{op} = \mathcal{O}\left(p\right). \tag{115}$$

This holds also in our case, as it can be proven following the argument in (99) and (100), where now the $n$ in the last line of (99) has to be replaced with $n'$, making the RHS there being $\mathcal{O}(1)$, as $n' = \mathcal{O}\left(\frac{d^{3/2}}{\log^3 d}\right)$ by definition. Lastly, the normalization of the data is used one more time in their Equation (C.91), but it is not critical to obtain the result, as $\|\hat{z}_i\|_2 = \Theta(\sqrt{d})$ is sufficient. We remark that Assumption 1.2 in (Bombari & Mondelli, 2024) also requires the covariance of the distribution to be well-conditioned, which however is not required for the purposes of the above mentioned lemmas. $\square$

**Lemma C.5.** *Let $\tilde{\phi} : \mathbb{R} \to \mathbb{R}$ be defined as $\tilde{\phi}(\cdot) := \phi(\cdot) - \mu_1(\cdot)$, and let $z \in \mathbb{R}^{2d}$ be sampled from a distribution respecting Assumption 6.3, not necessarily with the same covariance as $\mathcal{P}_{XY}$, and independent from $V$. Then we have that*

$$\left\| \mathbb{E}_z \left[ \tilde{\phi}(Vz)\tilde{\phi}(Vz)^\top \right] \right\|_{op} = \mathcal{O}\left( \log^4 d + \frac{p\log^3 d}{d^{3/2}} \right), \tag{116}$$

*with probability at least $1 - 2p^2\exp\left(-c\log^2 d\right)$ over $V$, where $c$ is an absolute constant.*

*Proof.* The proof follows a similar path as the one in Lemma C.15 in (Bombari & Mondelli, 2024). In particular, set

$$n' = \min\left(\left\lfloor \frac{p}{\log^4 p} \right\rfloor, \left\lfloor \frac{d^{3/2}}{\log^3 d} \right\rfloor\right), \qquad N = p^2 n', \tag{117}$$

and let $\tilde{\Phi}_N \in \mathbb{R}^{N \times p}$ be a matrix containing $\tilde{\phi}(V\hat{z}_i)$ in its i-th row, where every $\{\hat{z}_i\}_{i=1}^{N}$ is sampled independently from the same distribution of $z$. Thus, $\tilde{\Phi}_N$ can be seen as the vertical stacking of $p^2$ matrices with size $n' \times p$. All these matrices respect the hypotheses of Lemma C.4, and hence have their operator norm bounded by $\mathcal{O}\left(\sqrt{p}\right)$ with probability at least $1 - 2\exp\left(-c_1\log^2 d\right)$. Thus, performing a union bound over these $p^2$ matrices, we get

$$\left\| \tilde{\Phi}_N^\top \tilde{\Phi}_N \right\|_{op} = \mathcal{O}\left(p^2 p\right) = \mathcal{O}\left(\frac{Np}{n'}\right) = \mathcal{O}\left(N\log^4 p + \frac{Np\log^3 d}{d^{3/2}}\right), \tag{118}$$

with probability at least $1 - 2p^2\exp\left(-c_1\log^2 d\right)$ over $V$ and $\{\hat{z}_i\}_{i=1}^{N}$.

Via the same argument used for the last statement of Lemma C.1, denoting with $v_k \in \mathbb{R}^{2d}$ the k-th row of $V$, we have that $\|v_k\|_2 = \mathcal{O}(1)$ uniformly for every $k$ with probability at least $1 - 2p\exp(-c_2 d)$. Conditioning on such event, we have that each entry of $\tilde{\phi}(V\hat{z}_1)$ is sub-Gaussian (with uniformly bounded sub-Gaussian norm), since $\hat{z}_1$ is sub-Gaussian (as it is mean-0 and Lipschitz concentrated) and $\tilde{\phi}$ is a Lipschitz function. Thus, we have that each entry of $\mathbb{E}_{\hat{z}_1}\left[\tilde{\phi}(V\hat{z}_1)\right]$ is $\mathcal{O}(1)$

(see Proposition 2.5.2 in (Vershynin, 2018)), and therefore that $\left\|\mathbb{E}_{z_1}\left[\tilde{\phi}(V\hat{z}_1)\right]\right\|_{\psi_2} = \mathcal{O}\left(\left\|\mathbb{E}_{z_1}\left[\tilde{\phi}(V\hat{z}_1)\right]\right\|_2\right) = \mathcal{O}\left(\sqrt{p}\right)$.
Then, conditioning on the high probability event $\|V\|_{\mathrm{op}} = \mathcal{O}\left(\sqrt{p/d}\right)$ given by Lemma C.1, we have

$$\left\|\tilde{\phi}(V\hat{z}_1)\right\|_{\psi_2} \leq \left\|\tilde{\phi}(V\hat{z}_1) - \mathbb{E}_{z_1}\left[\tilde{\phi}(V\hat{z}_1)\right]\right\|_{\psi_2} + \left\|\mathbb{E}_{z_1}\left[\tilde{\phi}(V\hat{z}_1)\right]\right\|_{\psi_2} = \mathcal{O}\left(\sqrt{\frac{p}{d}} + \sqrt{p}\right) = \mathcal{O}\left(\sqrt{p}\right), \tag{119}$$

where the second step holds because $\hat{z}_1$ is Lipschitz concentrated and $\tilde{\phi}$ is Lipschitz. Since the rows of $\tilde{\Phi}_N$ are identically distributed, this also holds jointly for all other $\hat{z}_i$-s, for $i \in [N]$. Then, $\tilde{\Phi}_N/\sqrt{p}$ is a matrix with independent sub-Gaussian rows, and by Theorem 5.39 in (Vershynin, 2012) (see their Remark 5.40 and Equation (5.25)), we have that

$$\frac{1}{p}\left\|\frac{\tilde{\Phi}_N^\top \tilde{\Phi}_N}{N} - \mathbb{E}_z\left[\tilde{\phi}(Vz)\tilde{\phi}(Vz)^\top\right]\right\|_{\mathrm{op}} = \mathcal{O}\left(\sqrt{\frac{p}{N}}\right), \tag{120}$$

with probability at least $1 - 2\exp\left(-c_3 p\right)$ over $\{\hat{z}_i\}_{i=1}^N$. Then, we have

$$\begin{aligned}
\left\|\mathbb{E}_z\left[\tilde{\phi}(Vz)\tilde{\phi}(Vz)^\top\right]\right\|_{\mathrm{op}} &\leq \left\|\frac{\tilde{\Phi}_N^\top \tilde{\Phi}_N}{N} - \mathbb{E}_z\left[\tilde{\phi}(Vz)\tilde{\phi}(Vz)^\top\right]\right\|_{\mathrm{op}} + \frac{\left\|\tilde{\Phi}_N^\top \tilde{\Phi}_N\right\|_{\mathrm{op}}}{N} \\
&= \mathcal{O}\left(p\sqrt{\frac{p}{N}}\right) + \mathcal{O}\left(\log^4 p + \frac{p\log^3 d}{d^{3/2}}\right) \\
&= \mathcal{O}\left(\sqrt{p}\sqrt{\frac{\log^4 p}{p}} + \sqrt{p}\sqrt{\frac{\log^3 d}{d^{3/2}}}\right) + \mathcal{O}\left(\log^4 p + \frac{p\log^3 d}{d^{3/2}}\right) \\
&= \mathcal{O}\left(\log^4 p + \frac{p\log^3 d}{d^{3/2}}\right),
\end{aligned} \tag{121}$$

where the first step follows from the triangle inequality, the second step is a consequence of (120) and (118), and the third step follows from the definition of $N$.

Taking the intersection between the high probability events in (118), (119) and (120), the previous equation then holds with probability at least $1 - 2p^2\exp\left(-c_4 \log^2 d\right)$ over $V$ and $\{\hat{z}_i\}_{i=1}^N$. Also note that its LHS does not depend on $\{\hat{z}_i\}_{i=1}^N$, which were introduced as auxiliary random variables. Thus, the high probability bound holds restricted to the probability space of $V$, and the desired result follows. $\qquad\square$

**Lemma C.6.** *Let $\tilde{\phi} : \mathbb{R} \to \mathbb{R}$ be defined as $\tilde{\phi}(\cdot) := \phi(\cdot) - \mu_1(\cdot)$ and $\tilde{\Phi} \in \mathbb{R}^{n \times p}$ as the matrix containing $\tilde{\phi}(Vz_i)$ in its $i$-th row. Then, we have*

$$\left\|\tilde{\Phi}V\right\|_{\mathrm{op}} = \mathcal{O}\left(\sqrt{p}\log d + \frac{p\log d}{d}\right), \tag{122}$$

*with probability at least $1 - 2\exp\left(-c\log^2 d\right)$ over $Z$ and $V$, where $c$ is an absolute constant.*

*Proof.* Note that $\tilde{\phi}$ is Lipschitz (since $\phi$ is Lipschitz by Assumption 6.1). During all the proof, we condition on the event $\|Z\|_{\mathrm{op}} = \mathcal{O}\left(\sqrt{d}\right)$ and $\|z_i\|_2 = \Theta\left(\sqrt{d}\right)$ for all $i \in [n]$, which holds with probability at least $1 - 2\exp\left(-c_1 d\right)$ by Lemma C.1. During the proof we also use the shorthand $v \in \mathbb{R}^{2d}$ to denote a random vector such that $\sqrt{2d}\,v$ is a standard Gaussian vector, *i.e.*, it has the same distribution as the rows of $V$. This implies

$$\mathbb{E}_v\left[\left\|\tilde{\phi}(Zv)\right\|_2\right] = \mathcal{O}\left(\sqrt{n}\right), \qquad \left\|\left\|\tilde{\phi}(Zv)\right\|_2 - \mathbb{E}_v\left[\left\|\tilde{\phi}(Zv)\right\|_2\right]\right\|_{\psi_2} = \mathcal{O}\left(1\right), \tag{123}$$

and

$$\mathbb{E}_v\left[\|v\|_2\right] = \mathcal{O}\left(1\right), \qquad \left\|\|v\|_2 - \mathbb{E}_v\left[\|v\|_2\right]\right\|_{\psi_2} = \mathcal{O}\left(\frac{1}{\sqrt{d}}\right), \tag{124}$$

where both sub-Gaussian norms are meant on the probability space of $v$, and where the very first equation follows from the discussion in Lemma C.3 in (Bombari et al., 2022). Then, there exists an absolute constant $C_1$ such that we jointly have

$$\left\|\tilde{\phi}\left(Zv\right)\right\|_2 \le C_1\sqrt{d}, \qquad \|v\|_2 \le C_1 \tag{125}$$

with probability at least $1 - 2\exp\left(-c_2 d\right)$ over $v$.

Let $E_k$ be the indicator defined on the high probability event above with respect to the random variable $v_k := V_{:k}$ (the $k$-th row of $V$), *i.e.*

$$E_k := \mathbf{1}\left(\|v_k\|_2 \le C_1 \text{ and } \left\|\tilde{\phi}\left(Zv_k\right)\right\|_2 \le C_1\sqrt{d}\right), \tag{126}$$

and we define $E \in \mathbb{R}^{p \times p}$ as the diagonal matrix containing $E_k$ in its $k$-th entry. Notice that we have $\|I - E\|_{\mathrm{op}} = 0$ with probability at least $1 - 2p\exp\left(-c_2 d\right)$, and $\mathbb{E}_V\left[\|I - E\|_{\mathrm{op}}\right] \le 2p\exp\left(-c_2 d\right)$.

Thus, we have

$$
\begin{aligned}
\left\|\mathbb{E}_V\left[\tilde{\Phi}\left(I - E\right)V\right]\right\|_{\mathrm{op}} &\le \mathbb{E}_V\left[\left\|\tilde{\Phi}\right\|_{\mathrm{op}}\|1 - E\|_{\mathrm{op}}\|V\|_{\mathrm{op}}\right] \\
&\le \mathbb{E}_V\left[\left\|\tilde{\Phi}\right\|_{\mathrm{op}}^2\|V\|_{\mathrm{op}}^2\right]^{1/2}\mathbb{E}_V\left[\|I - E\|_{\mathrm{op}}^2\right]^{1/2} \\
&\le \mathbb{E}_V\left[\left\|\tilde{\Phi}\right\|_{\mathrm{op}}^4\right]^{1/4}\mathbb{E}_V\left[\|V\|_{\mathrm{op}}^4\right]^{1/4}\left(2p\exp\left(-c_2 d\right)\right)^{1/2} \\
&\le \mathbb{E}_V\left[\left\|\tilde{\Phi}\right\|_F^4\right]^{1/4}\mathbb{E}_V\left[\|V\|_F^4\right]^{1/4}\left(2p\exp\left(-c_2 d\right)\right)^{1/2} \\
&= o(1),
\end{aligned}
\tag{127}
$$

where the last step holds because of our initial conditioning on $Z$: the first two terms are the sum of finite powers of sub-Gaussian random variables (the entries of $\tilde{\Phi}$ and $V$), and thus (see Proposition 2.5.2 in (Vershynin, 2018)) the first two factors in the third line of the previous equation will be $\mathcal{O}\left(p^\alpha\right)$ for some finite $\alpha$, which gives the last line due to Assumption 6.2.

As in Lemma C.2, we introduce the notation (for all $i \in [n]$) $\tilde{\phi}^{(i)} : \mathbb{R} \to \mathbb{R}$ such that $\tilde{\phi}^{(i)}(\cdot) = \tilde{\phi}(\|z_i\|_2 \cdot /\sqrt{2d})$. Thus, denoting with $v \in \mathbb{R}^{2d}$ a random vector such that $\sqrt{2d}v$ is standard Gaussian (*i.e.*, distributed as the rows of $V$), we can write

$$\left[\mathbb{E}_V\left[\tilde{\Phi}V\right]\right]_{ij} = p\left[\mathbb{E}_v\left[\tilde{\phi}(Zv)v^\top\right]\right]_{ij} = \frac{p}{\sqrt{2d}}\mathbb{E}_v\left[\tilde{\phi}^{(i)}\left(\frac{z_i^\top}{\|z_i\|_2}\sqrt{2d}v\right)\left(e_j^\top\left(\sqrt{2d}v\right)\right)\right] = \frac{p}{\sqrt{2d}}\frac{\tilde{\mu}_1^{(i)}z_i^\top e_j}{\|z_i\|_2}, \tag{128}$$

where $\tilde{\mu}_1^{(i)}$ is the first Hermite coefficient of $\tilde{\phi}^{(i)}$. Then, denoting with $\tilde{D} \in \mathbb{R}^{n \times n}$ the diagonal matrix containing $\tilde{\mu}_1^{(i)}/\|z_i\|_2$ in its $i$-th entry, we can write

$$\left\|\mathbb{E}_V\left[\tilde{\Phi}V\right]\right\|_{\mathrm{op}} = \frac{p}{\sqrt{2d}}\left\|\tilde{D}Z\right\|_{\mathrm{op}} \le \frac{p}{\sqrt{2d}}\left\|\tilde{D}\right\|_{\mathrm{op}}\|Z\|_{\mathrm{op}}. \tag{129}$$

Then, since we conditioned on $\|z_i\|_2 = \Theta\left(\sqrt{d}\right)$ for all $i \in [n]$, following the same argument as in (96), and since the first Hermite coefficient of $\tilde{\phi}$ is 0 by definition, we have

$$\left\|\mathbb{E}_V\left[\tilde{\Phi}V\right]\right\|_{\mathrm{op}} = \mathcal{O}\left(\frac{p}{\sqrt{d}}\frac{\log d}{d}\sqrt{d}\right) = \mathcal{O}\left(\frac{p\log d}{d}\right), \tag{130}$$

with probability at least $1 - 2\exp\left(-c_3\log^2 d\right)$ over $Z$. A standard application of the triangle inequality to this last equation and (127) then gives

$$\left\|\mathbb{E}_V\left[\tilde{\Phi}EV\right]\right\|_{\mathrm{op}} \le \left\|\mathbb{E}_V\left[\tilde{\Phi}V\right]\right\|_{\mathrm{op}} + \left\|\mathbb{E}_V\left[\tilde{\Phi}\left(I - E\right)V\right]\right\|_{\mathrm{op}} = \mathcal{O}\left(\frac{p\log d}{d}\right), \tag{131}$$

with probability at least $1 - 2\exp\left(-c_4 \log^2 d\right)$ over $Z$.

Let's now look at

$$\tilde{\Phi}EV - \mathbb{E}_V\left[\tilde{\Phi}EV\right] = \sum_{k=1}^{p} \tilde{\phi}(Zv_k)E_kv_k^\top - \mathbb{E}_{v_k}\left[\tilde{\phi}(Zv_k)E_kv_k^\top\right] =: \sum_{k=1}^{p} W_k, \tag{132}$$

where we defined the shorthand $W_k = \tilde{\phi}(Zv_k)E_kv_k^\top - \mathbb{E}_{v_k}\left[\tilde{\phi}(Zv_k)E_kv_k^\top\right]$. (132) is the sum of $p$ i.i.d. mean-0 random matrices $W_k$ (in the probability space of $V$), such that

$$\begin{aligned}
\sup_{v_k}\left\|\tilde{\phi}(Zv_k)E_kv_k^\top - \mathbb{E}_{v_k}\left[\tilde{\phi}(Zv_k)E_kv_k^\top\right]\right\|_{\mathrm{op}} &\leq 2\sup_{v_k}\left\|\tilde{\phi}(Zv_k)E_kv_k^\top\right\|_{\mathrm{op}} \\
&= 2\sup_{v_k}\left\|\tilde{\phi}(Zv_k)\right\|_2 \|v_k\|_2 E_k \\
&\leq 2C_1^2\sqrt{d},
\end{aligned} \tag{133}$$

because of (126). Then, by matrix Bernstein's inequality for rectangular matrices (see Exercise 5.4.15 in (Vershynin, 2018)), we have that

$$\mathbb{P}_V\left(\left\|\tilde{\Phi}EV - \mathbb{E}_V\left[\tilde{\Phi}EV\right]\right\|_{\mathrm{op}} \geq t\right) \leq (n+d)\exp\left(-\frac{t^2/2}{\sigma^2 + 2C_1^2\sqrt{d}t/3}\right), \tag{134}$$

where $\sigma^2$ is defined as

$$\sigma^2 = p\max\left(\left\|\mathbb{E}_{v_k}\left[W_kW_k^\top\right]\right\|_{\mathrm{op}}, \left\|\mathbb{E}_{v_k}\left[W_k^\top W_k\right]\right\|_{\mathrm{op}}\right). \tag{135}$$

For every matrix $A$, we have $\mathbb{E}\left[(A - \mathbb{E}[A])(A - \mathbb{E}[A])^\top\right] = \mathbb{E}\left[AA^\top\right] - \mathbb{E}[A]\mathbb{E}[A]^\top \preceq \mathbb{E}\left[AA^\top\right]$. Thus,

$$\begin{aligned}
\left\|\mathbb{E}_{v_k}\left[W_kW_k^\top\right]\right\|_{\mathrm{op}} &\leq \left\|\mathbb{E}_{v_k}\left[\tilde{\phi}(Zv_k)E_kv_k^\top v_kE_k\tilde{\phi}(Zv_k)^\top\right]\right\|_{\mathrm{op}} \\
&\leq \left\|\mathbb{E}_{v_k}\left[\tilde{\phi}(Zv_k)\tilde{\phi}(Zv_k)^\top\right]\right\|_{\mathrm{op}}\sup_{v_k}\left(E_k\|v_k\|_2^2\right) \\
&\leq C_1^2\left\|\mathbb{E}_{v_k}\left[\tilde{\phi}(Zv_k)\tilde{\phi}(Zv_k)^\top\right]\right\|_{\mathrm{op}} \\
&= \mathcal{O}(1),
\end{aligned} \tag{136}$$

where the last step is a direct consequence of Lemma C.2, applied to $\tilde{\Phi}$ instead of to $\Phi$, and holds with probability at least $1 - 2\exp\left(-c_5\log^2 d\right)$ over $Z$. For the other argument in the $\max$ in (135) we similarly have

$$\begin{aligned}
\left\|\mathbb{E}\left[W_k^\top W_k\right]\right\|_{\mathrm{op}} &\leq \left\|\mathbb{E}_{v_k}\left[v_kE_k\tilde{\phi}(Zv_k)^\top\tilde{\phi}(Zv_k)E_kv_k^\top\right]\right\|_{\mathrm{op}} \\
&\leq \left\|\mathbb{E}_{v_k}\left[v_kv_k^\top\right]\right\|_{\mathrm{op}}\sup_{v_k}\left(E_k\left\|\tilde{\phi}(Zv_k)\right\|_2^2\right) \\
&\leq \frac{1}{d}C_1^2d \\
&= \mathcal{O}(1).
\end{aligned} \tag{137}$$

Then, plugging these last two equations in (134) we get

$$\mathbb{P}_V\left(\left\|\tilde{\Phi}EV - \mathbb{E}_V\left[\tilde{\Phi}EV\right]\right\|_{\mathrm{op}} \geq \sqrt{p}\log d\right) \leq (n+d)\exp\left(-\frac{p\log^2 d/2}{C_2p + 2C_1^2\sqrt{d}\sqrt{p}\log d/3}\right) \leq 2\exp\left(-c_6\log^2 d\right), \tag{138}$$

where we used Assumption 6.2. Then, applying a triangle inequality and using (131) and (138), we get

$$\left\|\tilde{\Phi}EV\right\|_{\mathrm{op}} = \mathcal{O}\left(\sqrt{p}\log d + \frac{p\log d}{d}\right), \tag{139}$$

with probability at least $1 - 2\exp\left(-c_7 \log^2 d\right)$ over $Z, V$. Then, since $E = I$ with probability at least $1 - 2p\exp\left(-c_3 d\right)$, using Assumption 6.2 we get the desired result.

$\square$

**Lemma C.7.** *Let $z \in \mathbb{R}^{2d}$ be sampled from a distribution respecting Assumption 6.3, not necessarily with the same covariance as $\mathcal{P}_{XY}$, independent from everything else, and let $f_{\mathrm{RF}}(\hat{\theta}_{\mathrm{RF}}(\lambda), z)$ be the RF model defined in (22) with $\hat{\theta}_{\mathrm{RF}}(\lambda)$ defined in (24), with $\lambda \geq 0$. Then, we have that*

$$\left| f_{\mathrm{RF}}(\hat{\theta}_{\mathrm{RF}}(\lambda), z) - \mu_1^2 p \frac{z^\top Z^\top}{2d} \left(\Phi\Phi^\top + n\lambda I\right)^{-1} G \right| = \mathcal{O}\left(\frac{d^{1/4}\log d}{p^{1/4}} + \frac{\log^{3/2} d}{d^{1/8}}\right) = o(1), \tag{140}$$

*with probability $1 - C\sqrt{d}\log^2 d/\sqrt{p} - C\log^3 d/d^{1/4}$ over $Z$, $G$, $V$ and $z$, where $C$ is an absolute constant*

*Proof.* Let $\tilde{\phi} : \mathbb{R} \to \mathbb{R}$ be defined as $\tilde{\phi}(\cdot) := \phi(\cdot) - \mu_1(\cdot)$, and let $\tilde{\Phi} \in \mathbb{R}^{n\times p}$ be defined as the matrix containing $\tilde{\phi}(Vz_i)$ in its $i$-th row. Then, introducing the shorthand $\hat{G} = \left(\Phi\Phi^\top + n\lambda I\right)^{-1} G$, we can write

$$\begin{aligned}
f_{\mathrm{RF}}(\hat{\theta}_{\mathrm{RF}}(\lambda), z) &= \left(\mu_1 Vz + \tilde{\phi}(Vz)\right)^\top \left(\mu_1 VZ^\top + \tilde{\Phi}^\top\right)\hat{G} \\
&= \mu_1^2 p \frac{z^\top Z^\top}{2d}\hat{G} + \mu_1^2 z^\top \left(V^\top V - \frac{p}{2d}I\right)Z^\top\hat{G} + \mu_1 z^\top V^\top \tilde{\Phi}^\top\hat{G} + \tilde{\phi}(Vz)^\top \Phi\hat{G}.
\end{aligned} \tag{141}$$

Notice that since every entry of $G$ is sub-Gaussian and independent by Assumption 6.3, Theorem 3.1.1 in (Vershynin, 2018) readily gives $\|G\|_2 = \mathcal{O}(\sqrt{n}) = \mathcal{O}\left(\sqrt{d}\right)$ with probability at least $1 - 2\exp(-c_1 d)$ over $G$. Then, conditioning on the high probability event described by Lemma C.3, we get

$$\left\|\hat{G}\right\|_2 \leq \left(\lambda_{\min}\left(\Phi\Phi^\top + n\lambda I\right)\right)^{-1}\|G\|_2 \leq \left(\lambda_{\min}\left(\Phi\Phi^\top\right)\right)^{-1}\|G\|_2 = \mathcal{O}\left(\frac{\sqrt{d}}{p}\right), \tag{142}$$

with probability at least $1 - 2\exp\left(-c_2 \log^2 d\right)$ over $V$, $Z$ and $G$. We will condition on this high probability event until the end of the proof. Let's then investigate the last 3 terms on the RHS of (141) separately:

(i) A direct application of Theorem 5.39 of (Vershynin, 2012) (see their Equation 5.23) gives

$$\left\|\frac{2d}{p}V^\top V - I\right\|_{\mathrm{op}} = \mathcal{O}\left(\sqrt{\frac{d}{p}}\right), \tag{143}$$

with probability at least $1 - 2\exp\left(-c_3 d\right)$ over $V$. Then, since $z$ is sub-Gaussian and independent from everything else, with probability $1 - 2\exp\left(-c_4 \log^2 d\right)$ over itself we have

$$\begin{aligned}
\left|\mu_1^2 z^\top \left(V^\top V - \frac{p}{2d}I\right)Z^\top\hat{G}\right| &\leq \log d \left\|\left(V^\top V - \frac{p}{2d}I\right)Z^\top\hat{G}\right\|_2 \\
&\leq \log d \left\|V^\top V - \frac{p}{2d}I\right\|_{\mathrm{op}}\|Z\|_{\mathrm{op}}\left\|\hat{G}\right\|_2 \\
&= \mathcal{O}\left(\log d \sqrt{\frac{p}{d}}\sqrt{d}\frac{\sqrt{d}}{p}\right) = \mathcal{O}\left(\frac{\sqrt{d}\log d}{\sqrt{p}}\right),
\end{aligned} \tag{144}$$

where the third step holds with probability at least $1 - 2\exp\left(-c_5 d\right)$ over $Z$ due to Lemma C.1.

(ii) As before, since $z$ is sub-Gaussian and independent from everything else, with probability $1 - 2\exp\left(-c_4 \log^2 d\right)$ we have

$$\begin{aligned}
\left|\mu_1 z^\top V^\top \tilde{\Phi}^\top\hat{G}\right| &\leq \log d \left\|V^\top \tilde{\Phi}^\top\right\|_{\mathrm{op}}\left\|\hat{G}\right\|_2 \\
&= \mathcal{O}\left(\log d \left(\sqrt{p}\log d + \frac{p\log d}{d}\right)\frac{\sqrt{d}}{p}\right) = \mathcal{O}\left(\log^2 d \left(\sqrt{\frac{d}{p}} + \frac{1}{\sqrt{d}}\right)\right),
\end{aligned} \tag{145}$$

where the second step holds because of Lemma C.6, and holds with probability at least $1 - 2\exp\left(-c_6 \log^2 d\right)$ over $Z$ and $V$.

(iii) For the last term of the RHS of (141), its second moment in the probability space of $z$ reads

$$
\mathbb{E}_z \left[ \hat{G}^\top \Phi^\top \tilde{\phi} \left( Vz \right) \tilde{\phi} \left( Vz \right)^\top \Phi \hat{G} \right] \leq \left\| \mathbb{E}_z \left[ \tilde{\phi} \left( Vz \right) \tilde{\phi} \left( Vz \right)^\top \right] \right\|_{\mathrm{op}} \| \Phi \|_{\mathrm{op}}^2 \left\| \hat{G} \right\|_2^2
$$

$$
= \mathcal{O} \left( \left( \log^4 d + \frac{p \log^3 d}{d^{3/2}} \right) \sqrt{d} \frac{\sqrt{d}}{p} \right) = \mathcal{O} \left( \frac{d \log^4 d}{p} + \frac{\log^3 d}{\sqrt{d}} \right), \tag{146}
$$

where the second step follows from Lemmas C.5 and C.3, and holds with probability at least $1 - 2 \exp \left( -c_7 \log^2 d \right)$ over $Z$ and $V$. Then, by Markov inequality, we have that there exists a constant $C_1$ such that

$$
\left( \tilde{\phi} \left( Vz \right)^\top \Phi \hat{G} \right)^2 < C_1 \left( \frac{d \log^4 d}{p} + \frac{\log^3 d}{\sqrt{d}} \right) t, \tag{147}
$$

with probability at least $1 - 1/t$ over $z$. Setting

$$
t = \min \left( \frac{\sqrt{p}}{\sqrt{d} \log^2 d}, d^{1/4} \right) = \omega(1), \tag{148}
$$

since $p = \omega \left( d \log^4 d \right)$ by Assumption 6.2, we have

$$
\left| \tilde{\phi} \left( Vz \right)^\top \Phi \hat{G} \right| = \mathcal{O} \left( \frac{d^{1/4} \log d}{p^{1/4}} + \frac{\log^{3/2} d}{d^{1/8}} \right), \tag{149}
$$

with probability at least $1 - \sqrt{d} \log^2 d / \sqrt{p} - \log^3 d / d^{1/4}$.

Then, plugging (i), (ii) and (iii) in (141) gives

$$
\left| f_{\mathrm{RF}}(\hat{\theta}_{\mathrm{RF}}(\lambda), z) - \mu_1^2 p \frac{z^\top Z^\top}{2d} \hat{G} \right| = \mathcal{O} \left( \frac{d^{1/4} \log d}{p^{1/4}} + \frac{\log^{3/2} d}{d^{1/8}} \right), \tag{150}
$$

with probability at least $1 - c_8 \frac{\sqrt{d} \log^2 d}{\sqrt{p}} - c_8 \frac{\log^3 d}{d^{1/4}}$, which gives the desired result. $\qquad \square$

**Proof of Theorem 6.4**  Let $E \in \mathbb{R}^{n \times n}$ be the matrix defined as

$$
E = \Phi \Phi^\top - p \left( \mu_1^2 \frac{ZZ^\top}{2d} + \tilde{\mu}^2 I \right). \tag{151}
$$

Note that

$$
\| E \|_{\mathrm{op}} \leq \left\| \Phi \Phi^\top - \mathbb{E}_V \left[ \Phi \Phi^\top \right] \right\|_{\mathrm{op}} + \left\| \mathbb{E}_V \left[ \Phi \Phi^\top \right] - p \left( \mu_1^2 \frac{ZZ^\top}{2d} + \tilde{\mu}^2 I \right) \right\|_{\mathrm{op}} = \mathcal{O} \left( p \left( \sqrt{\frac{d}{p}} + \frac{\log^3 d}{\sqrt{d}} \right) \right), \tag{152}
$$

with probability at least $1 - 2 \exp \left( -c_1 \log^2 d \right)$ over $Z, V$ due to Lemmas C.2 and C.3. By the Woodbury matrix identity (or Hua's identity), we have

$$
\left( \Phi \Phi^\top + n\lambda I \right)^{-1} = \left( p \left( \mu_1^2 \frac{ZZ^\top}{2d} + \tilde{\mu}^2 I \right) + E + n\lambda I \right)^{-1}
$$

$$
= \left( \mu_1^2 p \frac{ZZ^\top}{2d} + \left( \tilde{\mu}^2 p + n\lambda \right) I + E \right)^{-1}
$$

$$
= \left( \mu_1^2 p \frac{ZZ^\top}{2d} + \left( \tilde{\mu}^2 p + n\lambda \right) I \right)^{-1}
$$

$$
- \left( \mu_1^2 p \frac{ZZ^\top}{2d} + \left( \tilde{\mu}^2 p + n\lambda \right) I \right)^{-1} E \left( \mu_1^2 p \frac{ZZ^\top}{2d} + \left( \tilde{\mu}^2 p + n\lambda \right) I + E \right)^{-1}, \tag{153}
$$

which gives

$$
\begin{aligned}
&\left| \mu_1^2 p \frac{z^\top Z^\top}{2d} \left( \Phi\Phi^\top + n\lambda I \right)^{-1} G - \mu_1^2 p \frac{z^\top Z^\top}{2d} \left( \mu_1^2 p \frac{ZZ^\top}{2d} + (\tilde\mu^2 p + n\lambda) I \right)^{-1} G \right| \\
&\qquad \leq \left| \mu_1^2 p \frac{z^\top Z^\top}{2d} \left( \mu_1^2 p \frac{ZZ^\top}{2d} + (\tilde\mu^2 p + n\lambda) I \right)^{-1} E \left( \mu_1^2 p \frac{ZZ^\top}{2d} + (\tilde\mu^2 p + n\lambda) I + E \right)^{-1} G \right| \\
&\qquad \leq \log d \left\| \frac{p}{2d} Z^\top \left( \mu_1^2 p \frac{ZZ^\top}{2d} + (\tilde\mu^2 p + n\lambda) I \right)^{-1} E \left( \mu_1^2 p \frac{ZZ^\top}{2d} + (\tilde\mu^2 p + n\lambda) I + E \right)^{-1} G \right\|_2 \\
&\qquad \leq \log d \frac{p}{2d} \| Z \|_{\mathrm{op}} \frac{1}{\tilde\mu^2 p} \| E \|_{\mathrm{op}} \frac{1}{\lambda_{\min}(\Phi\Phi^\top)} \| G \|_2 \\
&\qquad = \mathcal{O} \left( \log d \frac{p}{d} \sqrt{d} \frac{1}{p} p \left( \sqrt{\frac{d}{p}} + \frac{\log^3 d}{\sqrt{d}} \right) \frac{1}{p} \sqrt{d} \right) \\
&\qquad = \mathcal{O} \left( \log d \sqrt{\frac{d}{p}} + \frac{\log^4 d}{\sqrt{d}} \right).
\end{aligned}
\tag{154}
$$

Here, the second step holds with probability at least $1 - 2\exp\left(-c_2 \log^2 d\right)$ since $z$ is sub-Gaussian and independent from everything else; the fourth step is a consequence of Lemma C.1, (152), Lemma C.3, and $\|G\|_2 = \mathcal{O}\left(\sqrt{d}\right)$ (see the argument prior to (142)), and as a whole holds with probability $1 - 2\exp\left(-c_3 \log^2 d\right)$ over $Z$, $G$, and $V$.

Note that the second term in the LHS of (154) can be written as

$$
\begin{aligned}
\mu_1^2 p \frac{z^\top Z^\top}{2d} \left( \mu_1^2 p \frac{ZZ^\top}{2d} + (\tilde\mu^2 p + n\lambda) I \right)^{-1} G &= z^\top Z^\top \left( ZZ^\top + n \left( \frac{2\tilde\mu^2 d}{\mu_1^2 n} + \frac{2d}{\mu_1^2 p} \lambda \right) I \right)^{-1} G \\
&= z^\top \left( Z^\top Z + n \left( \frac{2\tilde\mu^2 d}{\mu_1^2 n} + \frac{2d}{\mu_1^2 p} \lambda \right) I \right)^{-1} Z^\top G \\
&= f_{\mathrm{LR}}(\hat\theta_{\mathrm{LR}}(\tilde\lambda), z),
\end{aligned}
\tag{155}
$$

where the second line is due to the classical identity $A^\top \left( AA^\top + \kappa I \right)^{-1} = \left( A^\top A + \kappa I \right)^{-1} A^\top$, and the third line uses the definition in (5), with $\hat\theta_{\mathrm{LR}}(\tilde\lambda)$ defined in (8) and

$$
\tilde\lambda = \frac{2\tilde\mu^2 d}{\mu_1^2 n} + \frac{2d}{\mu_1^2 p} \lambda.
\tag{156}
$$

Furthermore, the first term of the LHS of (154) satisfies

$$
\left| \mu_1^2 p \frac{z^\top Z^\top}{2d} \left( \Phi\Phi^\top + n\lambda I \right)^{-1} G - f_{\mathrm{RF}}(\hat\theta_{\mathrm{RF}}(\lambda), z) \right| = \mathcal{O} \left( \frac{d^{1/4} \log d}{p^{1/4}} + \frac{\log^{3/2} d}{d^{1/8}} \right),
\tag{157}
$$

with probability $1 - C_1 \sqrt{d} \log^2 d / \sqrt{p} - C_1 \log^3 d / d^{1/4}$ over $Z$, $G$, $V$ and $z$, due to Lemma C.7. Thus, an application of the triangle inequality together with (154), (155) and (157) gives the desired result. $\qquad\square$

## D. Features Composed as $z = x + y$

In this work, we consider the data to be composed as $z = [x^\top, y^\top]^\top$. However, some of the high level results we obtain can be recovered for settings where the features are composed differently. As an example, let us consider the model

$$
z = x + y, \qquad g = x^\top \theta^* + \varepsilon.
\tag{158}
$$

Then, according to (3), we have

$$
\mathcal{C}(\hat\theta) = \mathrm{Cov}\left( (\tilde x + y)^\top \hat\theta, x^\top \theta^* \right) = \hat\theta^\top \Sigma_{yx} \theta^*,
\tag{159}
$$

*Figure 6.* Out-of-distribution test loss $\mathcal{L}(\hat{\theta}_{\text{LR/RF}}(\lambda))$ (black) and spurious correlations $\mathcal{C}(\hat{\theta}_{\text{LR/RF}}(\lambda)$ (red) as a function of $\lambda_{\max}(\Sigma_{yy})$ (first panel) and $\lambda_{\min}(S_x^\Sigma)$ (second panel) on a Gaussian synthetic dataset, and for the CIFAR-10 experiment (third panel). We consider the same set-up as Figures 3 and 4, for Gaussian and CIFAR-10 data, respectively.

which provides a quantity that could be studied again via the analysis in (Han & Xu, 2023). In this new setup, we remark that the full data covariance takes the form

$$\Sigma_{zz} = \Sigma_{xx} + \Sigma_{yy} + \Sigma_{xy} + \Sigma_{yx}. \tag{160}$$

In a nutshell, we expect the analysis for this setting to provide a qualitative behavior similar to that unveiled by the case $z = [x^\top, y^\top]^\top$. In fact, the experiments on Color-MNIST (which does not strictly follow the model $z = [x^\top, y^\top]^\top$, as the color overlaps with the core feature pattern as in the model $z = x + y$) suggest that our conclusions hold beyond the setting of "orthogonal features". We further remark that in the setting $z = [x^\top, y^\top]^\top$ the optimal solution $\hat{\theta} = \theta^*$ gives $\mathcal{C} = 0$, while this is not necessarily the case in the setting $z = x + y$.

## E. Proof of (4): Connection with Out-of-distribution Loss

Let $\tilde{x}$ and $[x^\top, y^\top]^\top$ be sampled independently from $\mathcal{P}_X$ and $\mathcal{P}_{XY}$ respectively. For simplicity, assume that $\mathbb{E}\left[f(\hat{\theta}, [\tilde{x}^\top, y])^2\right] = \mathbb{E}\left[f_x^*(\tilde{x})^2\right] = 1$ and $\mathbb{E}\left[f(\hat{\theta}, [\tilde{x}^\top, y])\right] = \mathbb{E}[f_x^*(\tilde{x})] = 0$. Thus, for the quadratic loss, we readily get

$$\mathbb{E}_{\tilde{x}, y}\left[\left(f(\hat{\theta}, [\tilde{x}^\top, y]) - f_x^*(\tilde{x})\right)^2\right] = 2 - 2\mathbb{E}_{\tilde{x}, y}\left[f(\hat{\theta}, [\tilde{x}^\top, y])f_x^*(\tilde{x})\right] = 2 - 2\text{Cov}\left(f(\hat{\theta}, [\tilde{x}^\top, y]), f_x^*(\tilde{x})\right). \tag{161}$$

Denoting with $S$ the covariance matrix of the three random variables $f(\hat{\theta}, [\tilde{x}^\top, y])$, $f_x^*(\tilde{x})$, and $f_x^*(x)$, we have

$$S = \begin{pmatrix} 1 & \rho & \mathcal{C} \\ \rho & 1 & 0 \\ \mathcal{C} & 0 & 1 \end{pmatrix}, \tag{162}$$

where we introduced the shorthands $\mathcal{C} = \text{Cov}\left(f(\hat{\theta}, [\tilde{x}^\top, y]), f_x^*(x)\right)$ and $\rho = \text{Cov}\left(f(\hat{\theta}, [\tilde{x}^\top, y]), f_x^*(\tilde{x})\right)$. Since $S$ is p.s.d., its determinant has to be non-negative, hence

$$1 - \rho^2 - \mathcal{C}^2 \geq 0, \tag{163}$$

which, when plugged in (161), gives (4).

This bound suggests the close connection between $\mathcal{C}$ and the out-of-distribution test loss. In Figure 6, we repeat the same experiments of Figures 3-4 and report in black the out-of-distribution test loss. The plots clearly show that $\mathcal{C}$ and the out-of-distribution test loss follow a similar trend, for both linear regression and random features.

## F. Experimental Details

All the plots in the figures report the average over 10 independent trials, with a shaded area describing a confidence interval of 1 standard deviation. For the Gaussian and Color-MNIST datasets, every iteration involves re-generating (or re-coloring) the data, while for the CIFAR-10 dataset the randomness comes from the model and the training algorithm.

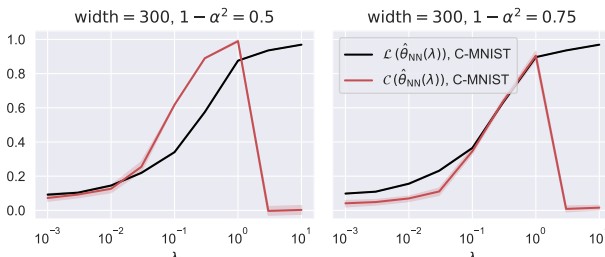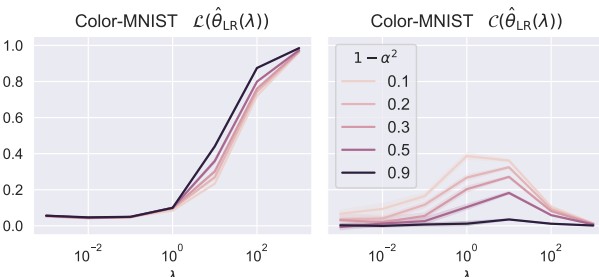

*Figure 7.* Test loss $\mathcal{L}(\hat{\theta}_{\text{NN/LR}}(\lambda))$ (black) and spurious correlations $\mathcal{C}(\hat{\theta}_{\text{NN/LR}}(\lambda)$ (red) as a function of $\lambda$. *First and second panel:* 2-layer fully connected ReLU network, trained on the multi-class color(C)-MNIST, for two different values of $\alpha$. *Third and fourth panel:* Same setup as in Figure 2, for the binary C-MNIST dataset, with multiple values of $\alpha$.

**Synthetic Gaussian data generation.** This follows the same model across all the numerical experiments presented in the paper. In particular, we fix $d = 400$ and set $\Sigma_{xx} = I$. $\Sigma_{yy}$ is a diagonal matrix, such that its first entry equals $\lambda_{\max}(\Sigma_{yy})$ and all the other entries equal $(d - \lambda_{\max}(\Sigma_{yy}))/(d-1)$. In this way, $\text{tr}(\Sigma) = 2d$. Then, we set the off-diagonal blocks $\Sigma_{xy}$ and $\Sigma_{yx}$ to the same diagonal matrix, so that

$$\Sigma_{xy} = \Sigma_{yx} = (\Sigma_{yy} - \beta I)^{1/2}, \tag{164}$$

which implies that the Schur complement $S_x^{\Sigma} = \beta I$ and, therefore, $\lambda_{\min}(S_x^{\Sigma}) = \beta$. To conclude, we set the ground truth $\theta_x^* = e_1$, *i.e.*, the first element of the canonical basis in $\mathbb{R}^d$. This design choice is motivated by our interest in capturing the role of $\lambda_{\max}(\Sigma_{yy})$ and to have an easy control on the Schur complement $S_x^{\Sigma}$ (which is therefore chosen to be proportional to the identity).

Unless differently stated in the figure, $n = 2000$, $\lambda_{\max}(\Sigma_{yy}) = 2$, $\beta = 0.5$, and $\lambda = 1$. Furthermore, to generate the labels, we add an independent noise with variance $\sigma^2 = 0.25$, and we subtract this quantity from the test loss, so that the optimal predictor $\theta^*$ has a test loss equal to 0.

When we use an RF model, on every dataset, unless differently stated in the figure, we use $\tanh$ as activation function, with $p = 20000$ neurons.

**Binary color MNIST.** This dataset is graphically shown in Figure 1. To generate it, we take a subset of the MNIST training dataset ($n = 1000$ samples as default, unless differently specified) made only of zeros and ones. Then, for every training image, we color the white portion in blue (red) with probability $(1 + \alpha)/2$ if the digit is a zero (one), and red (blue) otherwise. For the test set, we proceed in the same way, but setting $\alpha = 0$, to make the core feature (the digit) effectively independent from the spurious one (the color). For all the experiments, we set $\beta = 1 - \alpha^2 = 0.25$.

**CIFAR-10.** For the experiments on CIFAR-10, we implicitly suppose that the middle $22 \times 22$ square contains the core, predictive feature $x$. Thus, we sum to all the channels of the outer region white noise with increasing variance, and we later clamp the pixels to ensure their value is between 0 and 1. Increasing the variance of the noise, this progressively makes the outer portion being dominated by random noise, thus reducing its value of $\lambda_{\max}(\Sigma_{yy})/\text{tr}(\Sigma_{yy})$ when estimating the covariance on the perturbed training set. At test time, we take the images from the CIFAR-10 test set, and we add the same level of noise. To compute $\mathcal{C}$, we create an out-of-distribution dataset where the core features are randomly permuted across different backgrounds. We always consider the subset of boats and trucks, which contains $n = 10000$ images.

**2-layer neural network.** In the experiments shown in Figure 5, we consider a 2-layer neural network trained with gradient descent and quadratic loss on the Color-MNIST and CIFAR-10 datasets. For both datasets, we train for 1000 epochs, with learning rate 0.003, and batch size 1000.

### F.1. Additional Experiments

In the left two panels of Figure 7 we consider Color-MNIST including all 10 digits. We train a 2-layer network on all classes with one-hot encoding and MSE loss. Odd (even) digits are red (blue) with probability $(1 + \alpha)/2$, and blue (red) with

probability $(1 - \alpha)/2$. To compute $\mathcal{C}$, at test time we consider the parity of the logit with the highest value, with respect to the color of the image. The first and the second panel correspond to two values of $\alpha$-s and follow a similar profile as Figure 5 (left), showing the same qualitative behavior of $\mathcal{L}$ and $\mathcal{C}$ with respect to $\lambda$ for the full Color-MNIST dataset (i.e., in the multi-class setting). In the third and fourth panel of Figure 7 we repeat the experiment in Figure 2 (right) for multiple values of $\alpha$, reporting $\mathcal{L}$ and $\mathcal{C}$ with respect to $\lambda$ for linear regression. The curves behave as expected: for any value of $\lambda$, as $\alpha$ decreases, $\mathcal{C}$ decreases. Furthermore, the (in-distribution) test loss decreases as $\alpha$ increases, in agreement with our discussion at the end of Section 5.

In Figure 8 we train a 2-layer network and a random feature model on the classes "trucks" and "boats" from the Corrupted CIFAR-10 dataset (used in the context of spurious correlations in (Liu et al., 2023)), enforcing a correlation in the training set with respectively the textures "brightness" and "glass blur", with a correlation $\alpha = 0.95$. In both panels, we see a mild increase in $\mathcal{C}$ as $\lambda$ initially increases, until the later decrease predicted by Proposition 5.1. The profiles are also qualitatively similar to the ones of Figure 5 (left).

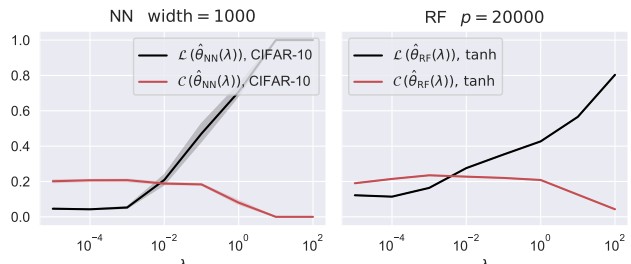

*Figure 8.* Test loss $\mathcal{L}(\hat{\theta}_{\mathrm{NN/RF}}(\lambda))$ (black) and spurious correlations $\mathcal{C}(\hat{\theta}_{\mathrm{NN/RF}}(\lambda))$ (red) as a function of $\lambda$. *Left:* 2-layer fully connected ReLU network, trained on the Corrupted CIFAR-10 dataset (boats and trucks, with added textures "brightness" and "glass blur"). *Right:* RF model with $\tanh$.

