# OpenReview forum: "Spurious Correlations in High Dimensional Regression: The Roles of Regularization, Simplicity Bias and Over-Parameterization"
_ICML.cc/2025/Conference — ICML 2025 poster_

### Official Review · Reviewer_mVaN · 2025-03-05

**Overall Recommendation:** 4

**Summary:**

This submission investigates the extent to which spurious correlations are used for learning in two models, linear ridge regression and random feature models, under the setting that the input dimension grows proportionally with the sample size. The key definition is $\mathcal{C}(\hat{\theta})$ in (3), and for $\hat{\theta}$ with and without regularisation, its non-asymptotic concentration results are proven (Theorems 4.1, 4.2). Section 5 is dedicated to investigating the role of regularisation in the behaviour of the above quantity and the test loss, and Section 6 is dedicated to showing an asymptotic similarity of random feature models with regularised linear ridge regression with a specific regularisation parameter.

**Claims And Evidence:**

The claims are supported by proofs in the Appendix and experiments. The experiments are well-aligned with the claims, and even though I couldn't go through all of the proofs, I couldn't find any errors. I had a question in one of the proofs, and it is written in the "Questions for Authors" section, and I would be grateful if the authors could answer it.

**Essential References Not Discussed:**

Not anything that I know of!

**Experimental Designs Or Analyses:**

The experiments look good.

**Methods And Evaluation Criteria:**

Yes.

**Other Comments Or Suggestions:**

102R: "is independently" -> "is independent" or "is chosen independently"

146L: "conditional to" -> "conditional on"

136R: "it can connected" -> "it can be connected"

242L: In (16), I don't think there is any need for bars, I think $S^\Sigma_x=\text{Cov}(y\mid x)=E_{y|x}[(y-E_{y|x}[y])(y-E_{y|x}[y])^\top]$ suffices. If you insist on using $\bar{x}$ to denote a particular value of the variable $x$, then I think it should also be $S^\Sigma_{\bar{x}}$ on the left-hand side.

In Section 3, in $f(\theta,z)$, the parameter $\theta$ comes before the input $z$, but in Section 6, this is reversed. Moreover, in Section 4, $f(\theta)$ is used without the input argument. I think it would be good to make this consistent, and even when the input argument is not explicitly present, write $f(\theta,\cdot)$.

625: $\epsilon$ should be $\mathcal{E}$.

188R: The projection operator $P_y$ is introduced here and again on 647. Perhaps redundant?

677: Full stop missing in (35).

683: In (36), in going from the second line to the third, $x^\top\theta^*_x$ turns into $[x^\top,\mathbf{0}^\top]^\top\theta^*$, but the second transpose shouldn't be there, it should be $[x^\top,\mathbf{0}^\top]\theta^*$.

688: "independent with" -> "independent from"

690: $\lVert\theta^*\rVert_2\leq1$ is not in Assumption 4.1, but on 166L. Also, you probably don't mean $\lVert\Sigma\rVert_\text{op}$-Lipschitz, but that you have $\lVert\mathcal{C}(\cdot)\rVert_\text{op}\leq\lVert\Sigma\rVert_\text{op}$? Slightly strange to talk about Lipschitz continuity of linear maps, as they are always Lipschitz continuous.

[Hastie et al., 2019] should be [Hastie et al., 2022].

**Other Strengths And Weaknesses:**

The paper was a pleasure to read - very well written (except some minor comments listed in the next section). I loved the fact that after every result, there was a paragraph starting with "In words", explaining clearly the significance of the result.

**Questions For Authors:**

146L: Covariance being zero does not imply independence either. Perhaps better to replace "as the covariance between $y_i$ and $x_i$ is in general non-zero" to "as $y_i$ and $x_i$ are in general not independent"?

662: In (32), could you please cite which exact form of Weyl's inequality you used, and how? I tried to derive (32) from the basic form of Weyl's inequality on wikipedia but I couldn't immediately get there.

206R: This sentence sounds strange, as it says it will "estimate" the empirical value by the "true" value, although I think I see what the authors mean, as the quantity of interest is actually $\mathcal{C}(\hat{\theta}_\text{LR}(\lambda))$, the amount of spurious correlation learned by the trained model $\hat{\theta}$. Still, I don't think it's the right choice of words, especially because we do not have access to the true value $\mathcal{C}^\Sigma(\lambda)$.

**Relation To Broader Scientific Literature:**

The related literature is discussed. This paper takes a slightly different aim at the problem of spurious correlations, in that the previous works seem to have focused on how to mitigate this problem, but this paper aims to characterise the extent of learning from spurious correlations and what role the regularisation parameter plays. Compared to what I imagine approaches like data augmentation does, which should not be model-specific, the results in this paper are specific for linear regression.

**Theoretical Claims:**

I checked the proofs of Theorems in Section 4, and some in Section 5. I couldn't find any errors. The only problem was that some proofs were very hard to follow, as they refer to (Han and Xu, 2023) without citing any of their notations and results, and I had to spend a lot of time looking into that paper, because the results are in that paper are not how it is used in the proofs here. I think it would be better if the authors cited the results precisely, stating precisely what simplifications were made.

Some minor comments are listed in the "Other Comments or Suggestions" section.

---

> ### Author Rebuttal · Authors · 2025-03-31
>
> We thank the reviewer for the remarkable care in reviewing our work and for the positive evaluation. We address concerns below.
>
> ---
>
> **Lack of clarity when citing (Han&Xu2023)**:
>
> To address this and ease the comparison of our claims with the results in Han&Xu2023, we will add to the appendix a discussion about the notation in the proof of Theorem 4.3. Specifically, we will connect our definition of the Gaussian sequence model $\hat\theta^\rho$ with the definition in Equation (1.5) in Han&Xu2023, and our definition for the test function $\varphi$ (see our line 693) with their notation ($\textup{g}$) in their Theorem 2.3.
>
> ---
>
> **Other Comments**:
>
> - 102R, 146L, 136R: Thanks for pointing these typos out, we will fix them in the revision.
>
> - 242L: While, in general, the conditional covariance depends on the particular value of $\bar x$, the definition of the Schur complement of the matrix $\Sigma$ does not rely on any specific instance of the random variable $x$. In the multivariate Gaussian case, it turns out that the conditional covariance is also independent of the particular instance $\bar x$, but we opted for leaving this notation at first as we did not consider it a trivial fact. If the reviewer finds this confusing, we are happy to remove this notation in the revision.
>
> - The arguments of $f(\theta, x)$ are sometimes swapped: Thanks for spotting this, we will fix it.
>
> - 625, 677, 683, 688: Thanks for pointing out these typos, we will fix them in the revision.
>
> - 647: Thanks for noticing this. While it is true that the notation of $P_y$ also appears in the body in line 188R, in line 647 we are providing the proof for a statement prior to that part, and we opted for redundancy to avoid confusion.
>
> - 690: Thanks for spotting this typo. Also, when we write that $\mathcal C(\cdot)$ is $\|| \Sigma \|| _{op}$-Lipschitz, we mean that the Lipschitz constant of $\mathcal C(\cdot)$ is upper bounded by the value of $\|| \Sigma \|| _{op}$. If the reviewer finds this notation confusing, we can elaborate more on this statement in the revision of the work.
>
> - Hastie&al.2022: thanks for noticing the typo, we will fix it in the revision.
>
> ---
>
> **Questions for Authors**:
>
> - 146L: We thank the reviewer for pointing this out. While the implication holds for Gaussian distributions, in 146L we still did not state Assumption 4.1 and the wording at this point could be misleading. We will fix it in the revision.
>
> - 662: We thank the reviewer for the question. Let us consider the inequality taken from the wikipedia page on Weyl’s inequality:
> $$ \lambda_{i+j-1}(A+B) \leq \lambda_{i}(A) + \lambda_{j}(B),$$
> where $A$ and $B$ are two $d \times d$ symmetric matrices and $\lambda_{j}(\cdot)$ denotes the $j$-th largest eigenvalue. Then, one could set $A = n \Sigma - Z^\top Z$ and $B = Z^\top Z$. Taking $i = 1$ and $j = d$ gives our Equation (32).
>
> - 206R: We thank the reviewer for pointing this out. Indeed the sentence might lead to confusing interpretations. We propose to rephrase it as: “Thus, for large $d, n$, we can theoretically analyze $\mathcal C(\hat \theta_{\textup{LR}}(\lambda))$ via the deterministic quantity $\mathcal C^\Sigma(\lambda)$, which, as highlighted by Equation (12), depends on $\theta^*$, the covariance of the data $\Sigma$, and the regularization $\lambda$ via the parameter $\tau(\lambda)$ introduced in (13).”

---

> > ### Comment · Reviewer_mVaN · 2025-04-03
> >
> > Dear authors,
> >
> > Thank you for your comments, I found them very adequate and thank you for promising to make the corrections I suggested. I maintain my (very) positive evaluation of this submission.
> >
> > Best,
> > reviewer

---

### Official Review · Reviewer_BGwm · 2025-03-13

**Overall Recommendation:** 4

**Summary:**

This paper quantifies the notion of spurious correlations -- where the feature of an image which determines it's classification correlates with another feature which does not determine the label -- and sudies the effect of spurious correlations for linear ridge regression and random-feature ridge regression.  Precisely, they define spurious correlations as the covariance between the spurious feature and the label when the informative feature is sampled independently.  They first demonstrate that with enough training data and zero ridge, the spurious correlations learned by a linear regression model will become small.  Increasing the ridge parameter, they observe that spurious correlations can improve test error when sampling in-distribution, and a proof is provided in the special case of isotropic covariance for the informative feature.  They then study spurious correlations in an over-parameterized random feature model.  By proving that the random feature model converges to a linear model with an increased ridge parameter, they show that spurious correlations are larger, but similar in their behavior, for random-feature models.

**Claims And Evidence:**

The claims made in this paper regarding spurious correlations in linear regression are well-supported by theoretical proofs and numerical experiments.  The claim I take issue with is the claim that over-parameterization increases spurious correlations.  This is because the scaling limit used to study the random feature model here is very limited.  The joint requirements that $p = \omega (n \log^4(n))$ and $\log(p) = \Theta (\log n)$ restrict the scaling to a very narrow regime where p grows only slightly faster than linearly with n.  This can also be recovered effectively by taking the proportional limit $p, n  \to \Infinity$ with $p/n = \gamma$ and then taking the limit $\gamma \to \infty$,  This second limit will destroy the variance induced by the random projection to a set of random features, which might have interesting effects on the spurious correlations learned.  In this limit, the higher-order Hermite coefficients of the nonlinearity (captured by $\tilde{\mu}$) have an identical effect as random i.i.d. noise applied to the features would have.  This additional contribution doesn't really interact with the structure of the data in a meaningful way

Also, in other scaling limits, such as $p \sim n^q$ as studied by Lu et. al in (https://arxiv.org/abs/2403.08160), the learning curves no longer reduce to the linear case with a renormalized ridge parameter.  To fully answer the question of how overparameterization affects spurious correlations, these faster scaling limits would need to be examined.

**Essential References Not Discussed:**

Random feature models beyond the proportional regime: https://arxiv.org/abs/2403.08160

These results also relate to work on linear regression with (possibly) noisy feature maps.  See, for example,
https://arxiv.org/abs/2102.08127

**Experimental Designs Or Analyses:**

N/A

**Methods And Evaluation Criteria:**

yes

**Other Comments Or Suggestions:**

none

**Other Strengths And Weaknesses:**

Strengths: Rigor, clarity of exposition, addresses an important, universal problem in the

Weakness: The main weakness is the novelty of the results. All follow from known estimates of the in-distribution and out-of-distribution prediction error for the proportional limit of linear and random feature models, except for some additional pointwise concentration guarantees in a narrow scaling regime.

**Questions For Authors:**

Is it necessarily correct to assume that the informative feature and spurious feature live in orthogonal subspaces of the input space?  How would your results change if instead of setting z = [x, y], you set z = x + y?

**Relation To Broader Scientific Literature:**

Spurious correlations are a universal problem in machine learning.  Thus, this relates broadly to the scientific literature on machine learning.  The technical contributions in this paper are also closely related to recent work on deterministic equivalents and random matrix methods for linear and random feature regression models.

**Theoretical Claims:**

I did not check the proofs, but the results are reasonable and consistent with the existing literature.

---

> ### Author Rebuttal · Authors · 2025-03-31
>
> We thank the reviewer for recognizing the rigor, clarity and importance of our results. We address concerns below.
>
> ---
>
> **Joint requirements $p = \omega(n)$ and $\log p = \Theta(\log n)$**:
>
> We note that the regime $p = \omega(n)$ and $\log p = \Theta(\log n)$ formally includes all scalings where $p \sim n^q$ for q > 1, as the latter is equivalent to $\log p = q \log n$. Thus, our assumptions include the faster scalings mentioned in the **claims and evidence** section of the review.
>
> In case the reviewer referred to a polynomial regime also between $d$ and $n$ (if $n = \Omega(d^l)$, with $l \geq 2$, the RF model learns more than the linear component of the target as indicated in Hu&al.2024 – the paper referenced by the reviewer), it is indeed true that the higher order component of the features would behave qualitatively differently, making the RF model qualitatively different from a regularized linear regression. We have opted to focus on the proportional regime $n = \Theta(d)$ due to its popularity in the literature (see e.g. Mei&Montanari2020, Hastie&al.2020) and due to its closeness to standard datasets in deep learning.
>
> ---
>
> **Relation to work on linear regression with (possibly) noisy feature maps**:
>
> We thank the reviewer for bringing to our attention the paper by Loureiro et al. ("Learning curves of generic features maps for realistic datasets with a teacher-student model"), which has indeed a similar setting to ours. Using the notation $z = [x, y]$ as in our paper, they consider a teacher-student setting where the labels are defined as a function of the feature $x$ (see their Eq. (1.2)), while the estimator $\hat \theta$ is obtained via ERM using only the (correlated) features $y$ (see their Eq. (1.3)). Then, their work is focused on studying the training and generalization error of the model that has access only to the partial information. Our setting, instead, looks at the ERM on both features, where the model has direct access also to the core features $x$. Due to the similarity with their setting, we will mention this related work and remark the differences in the revision of the paper.
>
> ---
>
> **Setting where $z = x + y$ instead of $z = [x, y]$**:
>
> We thank the reviewer for the insightful comment. Let us consider the model
>
> $$z = x + y, g = x^\top \theta^* + \varepsilon.$$
>
> Then, we have
>
> $$\mathcal C = \text{Cov} \left( (\tilde x + y)^\top \hat \theta, x^\top \theta^* \right) = \hat \theta^\top \Sigma_{yx} \theta^*,$$
>
> and this quantity could be studied via the analysis in Han&Xu2023 as in our current setting, considering that the covariance of the data will take the form
>
> $$\Sigma_{zz} = \Sigma_{xx} + \Sigma_{yy} + \Sigma_{xy} + \Sigma_{yx}.$$
>
> In a nutshell, we expect the analysis for this setting to provide a qualitative behaviour similar to that unveiled in the current version of our work. In fact, the experiments on Color-MNIST (which does not strictly follow the model $z = [x, y]$, as the color overlaps with the core feature pattern as in the model $z = x + y$) suggest that our conclusions hold beyond the setting of orthogonal features. We remark that in the setting $z = [x, y]$ the optimal solution $\hat \theta = \theta^*$ gives $\mathcal C = 0$, while this is not necessarily the case in the setting $z = x + y$.
>
> We will add a discussion on this point in the revision.

---

> > ### Comment · Reviewer_BGwm · 2025-04-04
> >
> > Thank you for addressing my questions.  I will maintain my (already high) score of 4.

---

### Official Review · Reviewer_8UVc · 2025-03-14

**Overall Recommendation:** 3

**Summary:**

This paper investigates spurious correlations in high-dimensional regression, focusing on the effects of regularization, simplicity bias, and over-parameterization. Using linear regression, the study quantifies how regularization influences the reliance on spurious correlations, revealing a trade-off where increasing regularization reduces test loss but strengthens spurious dependencies. It also demonstrates that models exhibit simplicity bias, favoring spurious features with dominant eigenvalues in their covariance structure, as these features offer an easier shortcut for prediction. The analysis introduces a formal measure of spurious correlations and links it to data covariance properties, particularly through the Schur complement, which captures the statistical dependence between core and spurious features.

To examine over-parameterization, the paper extends its analysis to random feature regression, showing that such models behave like regularized linear regression, even in the absence of explicit regularization. This result explains why spurious correlations persist in over-parameterized models, as the implicit regularization effect does not eliminate them. Theoretical results are complemented by numerical experiments on Gaussian synthetic data, Color-MNIST, and CIFAR-10, validating the key claims. The findings provide a rigorous statistical foundation for understanding spurious correlations and their interaction with model complexity, offering insights that can inform mitigation strategies for improving robustness and fairness in machine learning.

## Update after rebuttal
Thank you for the detailed rebuttal and thoughtful clarifications. I acknowledge the authors' responses and appreciate the effort in addressing the distinctions with related work, the discussion of applicability to deep networks, and the consideration of potential extensions. After reviewing the rebuttal, I will keep my original score.

**Claims And Evidence:**

The claims in the submission are largely supported by rigorous theoretical analysis and numerical experiments, making the evidence clear and convincing in most cases. The authors derive precise mathematical characterizations of spurious correlations, leveraging results from high-dimensional statistics, regularized linear regression, and random feature models. Their theoretical findings, such as the trade-off between regularization and spurious correlations and the equivalence between over-parameterized models and regularized regression, are well-grounded in established techniques. Additionally, the numerical experiments on Gaussian synthetic data, Color-MNIST, and CIFAR-10 align with the theoretical results, further strengthening their validity.

**Essential References Not Discussed:**

Yes, the following work is not cited and discussed while it also studies a similar setting. Furthermore, the paper significantly relates to other papers of Bombari et al. Even many of the proofs (e.g., see Lemma C.3-C.5) rely on the mentioned papers. There should be a more apparent discussion of how the current work is distinguished from the mentioned work.

> Bombari et al., 2024: "How Spurious Features are Memorized: Precise Analysis for Random and NTK Features"

—This work also investigates the role of spurious correlations in over-parameterized models, particularly focusing on random features and Neural Tangent Kernel (NTK) models. Given the conceptual overlap and shared proof techniques, the current paper should clarify how its contributions extend or differ from Bombari et al.'s findings.

**Experimental Designs Or Analyses:**

They look sound at a high level. The experiments on Gaussian synthetic data, Color-MNIST, and CIFAR-10 align well with the theoretical claims, providing empirical validation for key results. The analyses appear thorough, with appropriate comparisons and visualizations. While the study focuses on relatively simple models, the chosen datasets and methodologies effectively illustrate the impact of regularization, simplicity bias, and over-parameterization on spurious correlations.

**Methods And Evaluation Criteria:**

Yes, they are reasonable. The paper employs linear regression and random feature models, which are well-suited for studying spurious correlations in high-dimensional settings. The evaluation is based on both theoretical analysis and numerical experiments, using Gaussian synthetic data, Color-MNIST, and CIFAR-10, which are appropriate for validating the claims. While the analysis focuses on simplified models, the chosen methods effectively capture the core statistical phenomena under investigation.

**Other Comments Or Suggestions:**

-

**Other Strengths And Weaknesses:**

Strengths
1. The paper is well-written and easy to follow, presenting complex statistical concepts in a clear and structured manner.
2. By characterizing a deterministic object $\( \mathcal{C}^\Sigma(\lambda) \)$ to quantify spurious correlations, the paper provides a rigorous analysis of how regularization strength $\( \lambda \)$, data covariance $\( \Sigma \)$, and over-parameterization influence the learning of spurious features.

Weaknesses
1. The analysis primarily focuses on linear regression and random feature regression, making the setting simplistic and potentially limited in capturing the behavior of more complex models like deep neural networks.
2. The paper lacks a detailed comparison with prior work by Bombari et al., making it difficult to fully assess its technical contributions and novel challenges addressed. A clearer distinction from existing literature would strengthen the paper’s positioning.

**Questions For Authors:**

- Is it possible to extend the current findings to settings with feature learning (e.g., two-layer neural networks with both layers trained) rather than using fixed random features? Would the implicit bias of gradient-based optimization affect the spurious correlation analysis?

- Can the analysis be extended beyond the given data assumptions? Many real-world datasets exhibit heavy-tailed or structured dependencies—how would this impact the theoretical guarantees?

- How does this work differ fundamentally from Bombari et al. (2024)? Several proofs (e.g., Lemmas C.3–C.5) rely on techniques from Bombari et al., but a direct comparison is missing. Could you clarify the key distinctions and novel contributions?

- Does the identified trade-off between regularization and spurious correlations hold across different training objectives?

**Relation To Broader Scientific Literature:**

The key contribution is quantifying the amount of spurious correlations learned in high-dimensional regression with respect to regularization, simplicity bias, and over-parameterization. This builds on prior work in machine learning robustness, generalization in over-parameterized models, and implicit bias in deep learning, extending these ideas with a rigorous statistical characterization. The study connects to research on shortcut learning, generalization in empirical risk minimization (ERM), and random feature models, providing a more precise understanding of how spurious correlations emerge and persist. By linking these phenomena to covariance structures and regularization effects, the paper contributes valuable insights to ongoing discussions on fairness, bias mitigation, and model interpretability in modern machine learning.

**Theoretical Claims:**

Skimmed through it; they look sound at a high level. The proofs follow standard techniques in high-dimensional statistics, leveraging tools like Schur complements, concentration inequalities, and random matrix theory. Key results, such as the trade-off between regularization and spurious correlations and the equivalence between random feature models and regularized regression, appear well-structured and logically derived. A more detailed verification would be needed to confirm full correctness, but no obvious issues stand out.

---

> ### Author Rebuttal · Authors · 2025-03-31
>
> We thank the reviewer for the positive comments and the several interesting suggestions for extensions. We answer questions and address concerns below. We will incorporate the discussions in the revision.
>
> ---
>
> **Comparison with (Bombari et al., 2024):**
>
> Our work concerns the problem of spurious correlations, where a trained model is tested on a **newly sampled** data-point independent from training data, and we crucially use the independence between $\tilde x$, $x$, $y$ and $\hat \theta$ (see Proposition 4.2 and Theorem 4.3). In contrast, **(Bombari et al., 2024) do not consider the problem of spurious correlations**, but rather the setting where spurious features in the training set are memorized by an over-parameterized model. This is discussed in the first paragraph of their introduction, and it is quantitatively apparent in their definition of memorization in Equation (3.7), where the covariance is computed comparing the trained model evaluated on a spurious feature **contained in the training set** $y_i$ and the corresponding label $g_i$.
>
> In other words, the setting of our work is related to **robustness to distribution shift**, while (Bombari et al., 2024) focus on a setting where the individual training data are **memorized**, raising potential **privacy concerns**. Thus, the two works look at qualitatively very different problems.
>
> This difference is reflected in the proof strategies. While our work shares with (Bombari et al., 2024) an approach based on concentration of measure (and, consequently, also technical lemmas), the proof techniques are fundamentally different. Our work relies on the characterization of the ridge estimator $\hat \theta$ provided by Han&Xu2023 for linear regression, and it transfers the insights to random features via a point-wise equivalence principle. In contrast, the argument of (Bombari et al., 2024) is based on showing concentration of the auxiliary quantity $\mathcal F(z_i^s, z_i)$ that serves as a proxy to characterize the amount of memorization for an individual sample.
>
> ---
>
> **Setting might not capture the behaviour of deep neural networks**:
>
> While it is true that our analysis covers only high-dimensional regression, Figure 5 (left) shows a degree of similarity for shallow networks. For more complex and deep models, we point to the empirical results in Sagawa&al.2020, where higher penalty terms are shown to decrease test accuracy on ResNet50 and Bert. While it is hard to provide an exact predictive theory for deep models, we do believe our approach to capture important statistical aspects of the phenomenon of spurious correlations in more general settings than the one we precisely study.
>
> ---
>
> **Extension to feature learning**:
>
> Following Ba&al.2022 and Moniri&al.2023, one could extend our results to the setting where the target is not a linear function of the inputs and one step of gradient descent on the feature map improves the representation. We also note that our experiments with neural networks show concordance in the qualitative behavior of high-dimensional regression and 2-layer networks with both layers being trained.
>
> ---
>
> **Heavy tailed data**:
>
> The recent work by Adomaityte&al.2024 (“High-dimensional robust regression under heavy-tailed data: asymptotics and universality”) considers heavy tailed data in high-dimensional regression: the covariates are isotropic Gaussian with variance sampled from a distribution with heavy tails. Note that our problem setup requires a non-isotropic covariance, so one would have to first generalize their analysis accordingly. Then, a possible direction would be to investigate how different tail weights (between core and spurious features) favor learning of spurious correlations.
>
> ---
>
> **Different training objectives**:
>
> Empirically, the identified trade-off between regularization and spurious correlations has been verified in prior work (Sagawa&al.2020) looking at models trained on classification tasks.
>
> Theoretically, work by Montanari&al.2023 (“The generalization error of max-margin linear classifiers”) and Deng&al.2020 (“A model of double descent for high-dimensional binary linear classification”) provides the asymptotics for the generalization error of max-margin linear classifiers, also in the setting where classification is performed on a set of random features. For classification, we could define $\mathcal C$ as in our Equation (3), taking the $\text{sign}$ of the output of the model (which for classification represents the prediction of the model at test time).
>
> Equation (5.6) in Montanari&al.2023 provides a set of fixed point equations giving the limit deterministic value of the maximum margin and the prediction error (see their Theorem 3), also for non-isotropic covariates. Then, to extend our results to this setting, one approach could be to follow the strategy as in part c) of the proof technique in their Section 5.3, with the difference of computing $\mathcal C$ instead of the generalization error.

---

### Official Review · Reviewer_gx6Q · 2025-03-15

**Overall Recommendation:** 3

**Summary:**

The paper characterizes the learning of spurious features in linear regression as function of $\ell_2$ regularization strength and spurious feature simplicity. They also show that under overparametrization incurred by random features the effect of regularization is modified in a way that explains empirical results obtained from neural networks. The paper tests these hypotheses on synthetic and semi-synthetic datasets.

## Update after rebuttal
I thank the authors for their response and maintain my recommendation for acceptance.

**Claims And Evidence:**

The authors support their claim of characterizing learning in linear regression under spurious correlations theoretically in a convincing manner, with some additional empirical support.

**Essential References Not Discussed:**

I am not aware of any major, relevant papers that the current paper fails to cite.

**Experimental Designs Or Analyses:**

In addition to not using a original high-dimensional regression task, the authors' use of Colored MNIST and CIFAR-10 tasks diverge from the way they are commonly used, without explicit justification (and previous uses are not cited, . For binary Colored MNIST dataset the authors only work with a subset of the dataset, and experiment only with a single value of correlation between core and spurious features. They also create a spurious correlation dataset out of CIFAR-10, without referencing or reusing a very common variant of CIFAR-10 that's frequently used in the literature called Corrupted CIFAR-10.

**Methods And Evaluation Criteria:**

This is the weakest part of the paper, since the paper does not evaluate its predictions on any real bona fide regression task, but rather repurposes two classification tasks for this.

**Other Comments Or Suggestions:**

- Given the fact that "test loss" can often refer to loss under an unbiased test distribution (i.e. OOD risk) when studying spurious correlations, the authors should take care to remind the reader that their test loss is ID. The difference between the two can be reinforced by assigning a designated notation for OOD test loss.
- Overloading of $\lambda$ for eigenvalues and regularization coefficient is somewhat confusing, please change the notation for one if possible.
- Use of $y$ to denote spurious features and $g$ to denote labels creates an unnecessary cognitive load since it directly contradicts common usage in the previous work. I would recommend using $s$ and $y$ respectively, but ultimately it's in authors' discretion.
- 034L: "Gaussian dataset" -> "synthetic Gaussian dataset"?
- Theorem 4.3: Please alert the reader _beforehand_ that they are not supposed to have an intuition re. $\mathcal{C}^{\Sigma}(\lambda)$, and that this property will be studied in the following section. Otherwise Theorem 4.3 is needlessly confusing to digest in the first read.
- Please explicitly mention and discuss the implications of the fact that Proposition 4.2 and Theorem 4.3 have different assumptions regarding the data and sample dimensionality relationship.

**Other Strengths And Weaknesses:**

The paper sets out a clear motivation and proceeds to systematically demonstrate its claims, supported by the fact that the paper is well-written, making the authors' arguments easier to follow.

**Questions For Authors:**

Not a question per se, but to summarize my points above: Improving experimentation (see above) and a more in-depth discussion of the results in light of literature would improve the paper the most.

**Relation To Broader Scientific Literature:**

The paper is positioned appropriately within the relevant literature, and the paper's motivations are clearly presented. However, I would appreciate a more involved discussion of their results in relation the existing results in the literature, especially the implications of their work regarding feature learning order and interference between features based on difficulty and spurious correlation strength (c.f. Pezeshki et al. 2021, Qiu et al. 2024)

**Theoretical Claims:**

I have checked the proofs of Proposition 4.2 and Theorem 4.3, and did not see any problems.

---

> ### Author Rebuttal · Authors · 2025-03-31
>
> We thank the reviewer for the positive evaluation and helpful comments. We address concerns below.
>
> ---
>
> **Improving experimentation**:
>
> Following the reviewer’s suggestion, we will add the following experiments to the revision.
>
> In https://ibb.co/21W8qbLJ, we consider Color-MNIST including all digits (rather than a subset). We train a 2-layer network on all classes with one-hot encoding and MSE loss. Odd (even) digits are red (blue) with probability $(1+\alpha)/2$, and blue (red) with probability
> $(1-\alpha)/2$. To compute $\mathcal C$, at test time we consider the parity of the logit with the highest value, with respect to the color of the image. The two figures correspond to two values of $\alpha$-s and follow a similar profile as Figure 5 (left), showing the same qualitative behaviour of $\mathcal L$ and $\mathcal C$ with respect to $\lambda$ for the full Color-MNIST dataset (i.e., in the multi-class setting).
>
> In https://ibb.co/zWKBHk5k, we repeat the experiment in Figure 2 (right) for multiple values of $\alpha$, reporting $\mathcal L$ and $\mathcal C$ with respect to $\lambda$ for linear regression. The curves behave as expected: for any value of $\lambda$, as $\alpha$ decreases, $\mathcal C$ decreases. Furthermore, the (in-distribution) test loss decreases as $\alpha$ increases, in agreement with our discussion at lines 347-350 (left).
>
>
> We note that the choice of considering a data split in predictive and spurious features is conceptually similar to Section 5.1 of “Invariant risk minimization” by Arjovsky et al. As for our CIFAR-10 implementation, our experiment is designed to verify our claims on the simplicity bias in a controllable setting. In fact, introducing a tunable amount of noise allows us to modify $\lambda_{\max}(\Sigma_{yy})$ and use it as an independent variable in Figure 4. Nevertheless, considering image backgrounds as spurious features was done in the seminal cited work by Xiao&al.2020. Besides, the theoretical results suggesting that higher values of the regularizer $\lambda$ can be associated with higher values of $\mathcal C$ are also supported by the numerical evidence in Table 1 of Sagawa2020a.
>
>
> We thank the reviewer for mentioning the Corrupted CIFAR-10 dataset considered e.g. in “Avoiding spurious correlations via logit correction” by Liu et al. In https://ibb.co/TBg7Khwn, we train a 2-layer network and a random feature model on the classes ‘trucks’ and 'boats’, enforcing a correlation in the training set with respectively the textures ‘brightness’ and ‘glass_blur’, as in the available data for C-CIFAR-10 (here we use correlation $\alpha = 0.95$). In both figures, we see a mild increase in $\mathcal C$ as $\lambda$ initially increases, until the later decrease predicted by Proposition 5.1. The profiles are also qualitatively similar to the ones of Figure 5 (left).
>
> ---
>
> **More discussion in light of literature**:
>
> The main difference with (Qiu et al., 2024; Pezeshki et al., 2011) is that these works mainly focus on how spurious correlations evolve during training, while our work studies spurious correlations at convergence.
>
> To further connect with (Qiu et al., 2024; Pezeshki et al., 2011), we briefly discuss the intuition coming from our results from a dynamical perspective. In Section 5, we argue that $\lambda_{\max}(\Sigma_{yy})$ is related to $\mathcal C$, suggesting a measure of the simplicity of the feature $y$. In linear regression, solving gradient flow ($d \theta = - \nabla_\theta \mathcal L(\theta)dt$) gives
>
>
> $$\theta(t) = (1 - e^{ - (X^\top X + n \lambda I ) t}) \hat \theta.$$
>
>
> Thus, the components of $\hat \theta$ aligned with the top eigenspaces of $X^\top X$ converge earlier than the others. Hence, if $X^\top X \sim n \Sigma$, it is natural to expect that spurious features are learned faster the easier they are, and that they would prevail with respect to the core features (according to our bound in Proposition 5.1).
>
>
> ---
>
> **Other comments/suggestions**:
>
> We thank the reviewer for the detailed comments, which will improve the clarity of the revision:
>
> - We will explicitly note that the test loss is in distribution after Equation (2).
> - We will clarify the usage of the notation $\lambda$, as well as of $y$ to denote spurious features and $g$ to denote labels.
> - We will fix the typo in line 34L.
> - We will add the suggested remark before Theorem 4.3.
> - We will discuss the difference between the two different scaling regimes considered in Theorem 4.3 and Proposition 4.2.

---

### Decision · Program_Chairs · 2025-05-01

**Decision:**

Accept (poster)

**Comment:**

This work characterizes the effect of spurious features in high-dimensional ridge-regularized linear regression, evaluated by the covariance between the target value and the predicted score on an altered feature vector with the core features replaced with independent copies. This spurious covariance attains zero in the ridge-less case, and has a non-trivial value with regularization. The analysis reveals an interesting regime where the spurious covariance increases and the test error decreases, as the regularization strength varies. An asymptotic equivalence is established between ridge-regularized linear regression and overparametrized random feature models, allowing for an alternative interpretation of the analysis with respect to overparametrization.

The analysis is well-motivated and clearly presented, revealing interesting theoretical consequences supported by empirical results. The authors’ rebuttal effectively addressed the reviewers’ comments concerning the discussion with respect to related work, the implication of the theoretical result, and the experimentation, which is strengthened with added experiments. A weak point raised in the reviews is the technical novelty, as the analysis build upon the existing sharp results of linear regression, but with a different focus on the quantification of spurious correlation.